# Stability and Generalization for Bellman Residuals

## Abstract

Offline reinforcement learning and offline inverse reinforcement learning aim to recover near–optimal value functions or reward models from a fixed batch of logged trajectories, yet current practice still struggles to enforce Bellman consistency. *Bellman residual minimization* (BRM) has emerged as an attractive remedy, as a globally convergent stochastic gradient descent–ascent based method for BRM has been recently discovered. However, its statistical behavior in the offline setting remains largely unexplored. In this paper, we close this statistical gap. Our analysis introduces a single Lyapunov potential that couples SGDA runs on neighbouring datasets and yields an $\mathcal{O}(1/n)$ *on-average argument-stability* bound—doubling the best known sample-complexity exponent for convex–concave saddle problems. The same stability constant translates into the $\mathcal{O}(1/n)$ excess risk bound for BRM, without variance reduction, extra regularization, or restrictive independence assumptions on minibatch sampling. The results hold for standard neural-network parameterizations and minibatch SGD.

## 1 Introduction

Modern decision-making systems—from sepsis treatment strategies in intensive-care units to route planning for autonomous vehicles—must reason about sequences of actions whose consequences only unfold over time. Reinforcement learning (RL) provides a principled framework for such dynamic problems, formalizing them as Markov decision processes (MDPs) and prescribing policies that optimize long-horizon rewards. Yet in many high-stakes domains on-line interaction is either unethical, dangerous, or simply too expensive Jiang & Xie (2024). Practitioners therefore turn to offline RL, where the learner is handed a fixed batch of interaction data collected by some behaviour policy, and to its sister field inverse RL (IRL), which infers underlying preferences (i.e., reward functions) from logged expert trajectories.

In both offline RL and IRL, the crux of learning remains the same: given the fixed offline data, estimate an action-value function (Q function) that satisfies the *Bellman optimality equation.*, which characterizes the optimal action-value function $Q^\star$ as the fixed point of the equation. We say a $Q$-function is *Bellman consistent* if it is the fixed point of the corresponding Bellman operator (optimal or policy-specific), i.e., it satisfies the Bellman optimality equation for all state–action pairs.

Unfortunately, finding $Q$ that satisfies Bellman consistency offline is notoriously difficult. Policy-gradient objectives (Mei et al., 2020; Cen et al., 2022) assume online sampling and misalign with a static dataset. Fitted fixed-point methods, exemplified by Fitted Q-Iteration (Ernst et al., 2005) or Fitted Value Iteration (Munos & Szepesvári, 2008), can diverge catastrophically once the "deadly triad" of function approximation, bootstrapping, and off-policy data is present (Tsitsiklis & Van Roy, 1996; Van Hasselt et al., 2018; Chen et al., 2023). Importance-weighting approaches mitigate distribution shift but leave no guarantee that the learned $Q$ satisfies the Bellman equations uniformly over all $(s, a)$ pairs (Jiang & Xie, 2024). Bellman residual minimisation (BRM), directly fitting $Q$ by driving the squared Bellman error to zero, has long been viewed as conceptually appealing and practically effective, yet with limited theory to back global optimality convergence (Jiang & Xie, 2024).

A recent breakthrough by Kang et al. (2025) revisited BRM through a modern optimization lens. They showed that, after a classical bi-conjugate transformation, for popular $Q$-function parametrization choices such as linear functions and neural networks, minimizing the mean-squared Bellman error (MSBE) can be cast as a Polyak–Łojasiewicz (PL)–strongly-concave minimax optimization problem

[1]. This geometry immediately implies that plain stochastic gradient descent–ascent (SGDA) enjoys global convergence—sidestepping deadly-triad instability without intricate algorithmic tricks. As the optimization picture is now clear, what remains open is the statistical picture:

*How many offline samples we need for BRM to recover a near-optimal value function, under SGDA?*

**Contributions.** In this paper, we close this statistical gap popular function classes such as neural networks and linear function. Building on the PL structure identified by Kang et al. (2025), we develop a Lyapunov potential tailored to PL–strongly-concave optimization and blend it with a modern on-average argument-stability analysis. Our main theorem shows that, for a dataset of size $n$, SGDA attains an $\mathcal{O}(1/n)$ excess MSBE loss, a rate that doubles the exponent enjoyed in convex–concave optimization with Markov-sampled data, where the best known rate is $\mathcal{O}(1/\sqrt{n})$ (Wang et al., 2022). In particular:

1. We prove $\mathcal{O}(1/n)$ on-average argument-stability bound for SGDA under PL–strong concavity, avoiding any independence assumptions on the minibatch sampling indices.

2. Leveraging this stability, we derive the $\mathcal{O}(1/n)$ generalization guarantee for BRM.

3. Our analysis is constructive, requires no variance reduction or extra regularisation, and applies verbatim to standard neural-network parameterisations commonly used in offline RL.

Specifically, we compare two SGDA runs on neighbouring datasets (identical except for one replaced sample) using the same initialization and the same minibatch index sequence. A single Lyapunov potential $\Psi$ contracts in expectation each step by $(1 - c\eta_t)$, while stochastic gradient noise contributes only lower-order terms proportional to $\eta_t^2$ and $\eta_t/n$. Because $\sum \eta_t = \infty$ and $\sum \eta_t^2 < \infty$, the contraction accumulates but the noise terms are summable, so the decay dominates. Consequently, the trajectories remain close and we obtain an $O(1/n)$ stability bound.

## 2 SETUP AND BACKGROUNDS

**Markov Decision Process.** Throughout, we focus on a single-agent decision-making problem interacting with a discounted Markov Decision Process (MDP) described by the tuple $(\mathcal{S}, \mathcal{A}, P, r, \beta, \nu_0)$. A state is an element of the measurable space $\mathcal{S}$ and the agent chooses actions from the finite set $\mathcal{A}$. For any state–action pair $(s, a)$ the transition kernel $P(\cdot \mid s, a)$ gives a probability distribution over the next state. The immediate payoff at each timestep is given as $r(s, a) + \epsilon_a$, where the reward function $r : \mathcal{S} \times \mathcal{A} \to \mathbb{R}$ is the deterministic part, and the $\epsilon_a$ is the random part [2]. Following the recent literature (Garg et al., 2023), we model $\epsilon_a$ using the Gumbel distribution (often called the Type I Extreme Value (T1EV) distribution)[3]. To model $\epsilon_a$ as the mean-zero random noise, we use the mean-zero scale-one Gumbel distribution, i.e., $\epsilon_a \overset{i.i.d.}{\sim} G(-\gamma, 1)$ where $\gamma$ is the Euler constant. It is important to note that at each timestep, *right before* the action is chosen, the set $\{\epsilon_a\}_{a \in \mathcal{A}}$ is realized and revealed to the decision maker. The scalar $\beta \in (0, 1)$ exponentially discounts rewards that occur further in the future and $\nu_0$ denotes the distribution of the starting state $s_0$.

**Policy and value functions.** A (stationary Markov) policy $\pi \in \Delta_{\mathcal{A}}^{\mathcal{S}}$ assigns every state $s$ to $\pi(\cdot \mid s)$, a distribution over actions $\mathcal{A}$; when the agent is in state $s_h$ at time $h$ it samples $a_h \sim \pi(\cdot \mid s_h)$. Combined with the initial draw $s_0 \sim \nu_0$, a policy induces a probability measure $\mathbb{P}_{\nu_0, \pi}$ on infinite trajectories $(s_0, a_0, s_1, a_1, \dots)$, and the corresponding expectation operator is written $\mathbb{E}_{\nu_0, \pi}$. Under

---

[1]The Polyak–Łojasiewicz (PL) condition requires $\frac{1}{2}\|\nabla f(x)\|_2^2 \geq \mu (f(x) - f^*)$ for some $\mu > 0$. It ensures convergence guarantees comparable to those in the strongly convex case, even when $f$ is not convex. Polyak–Łojasiewicz (PL)–strongly-concave minimax optimization problem implies that the inner maximization problem is concave and the outer minimization problem is PL.

[2]This form of reward function is often referred to as satisfying linear additivity and conditional independence. (Rust, 1994)

[3]Garg et al. (2023) showed that the Gumbel distribution is not only theoretically convenient but also often a more plausible choice in practice than the Gaussian distribution for modeling the random part of the immediate payoff.

this setup, we consider the optimal policy and its corresponding value functions defined as

$$\pi^* := \operatorname{argmax}_{\pi \in \Delta_{\mathcal{A}}^{\mathcal{S}}} \mathbb{E}_{\nu_0, \pi, G} \left[ \sum_{h=0}^{\infty} \beta^h \left( r\left(s_h, a_h\right) + \epsilon_{a_h} \right) \right]$$

$$V^*(s) := \max_{\pi \in \Delta_{\mathcal{A}}^{\mathcal{S}}} \mathbb{E}_{\nu_0, \pi, G} \left[ \sum_{h=0}^{\infty} \beta^h \left( r\left(s_h, a_h\right) + \epsilon_{a_h} \right) \mid s_0 = s \right]$$

$$Q^*(s, a) := \max_{\pi \in \Delta_{\mathcal{A}}^{\mathcal{S}}} \mathbb{E}_{\nu_0, \pi, G} \left[ \sum_{h=0}^{\infty} \beta^h \left( r\left(s_h, a_h\right) + \epsilon_{a_h} \right) \mid s_0 = s, a_0 = a \right]$$

One can show that the optimal policy $\pi^*$ and the value functions satisfy the following optimality equations (Kang et al. (2025), Appendix B.4):

$$V^*(s) = \ln \left[ \sum_{a \in \mathcal{A}} \exp\left(Q^*(s, a)\right) \right]$$

$$\pi^*(a \mid s) = \frac{\exp\left(Q^*(s, a)\right)}{\sum_{a' \in \mathcal{A}} \exp\left(Q^*(s, a')\right)} \text{ for } a \in \mathcal{A}$$

$$Q^*(s, a) = r(s, a) + \beta \cdot \mathbb{E}_{s' \sim P(s, a)} \left[ \log \sum_{a' \in \mathcal{A}} \exp\left(Q^*(s', a')\right) \mid s, a \right]$$

Note that the optimality equations above are equivalent to the optimality equations of the entropy regularized reinforcement learning problems (Haarnoja et al., 2017; 2018)[4].

## 2.1 BELLMAN RESIDUAL MINIMIZATION

**Bellman Error (Bellman Residual) and Temporal Difference Error.** Define the function space $\mathcal{Q}$ as the set of all bounded real-valued functions on the state-action space $\mathcal{S} \times \mathcal{A}$:

$$\mathcal{Q} := \{Q : \mathcal{S} \times \mathcal{A} \to \mathbb{R} \mid \|Q\|_\infty < \infty\}$$

As established in (Rust, 1994), the optimal action-value function, $Q^*$, is an element of this space, i.e., $Q^* \in \mathcal{Q}$, provided that the discount factor $\beta$ is in (0,1).

We introduce the *Bellman optimality operator*, $\mathcal{T}$, which maps a function in $\mathcal{Q}$ to another function in $\mathcal{Q}$. For any $Q \in \mathcal{Q}$, the operator is defined as:

$$(\mathcal{T}Q)(s, a) := r(s, a) + \beta \cdot \mathbb{E}_{s' \sim P(s, a)} \left[ \log \sum_{a' \in \mathcal{A}} \exp\left(Q(s', a')\right) \mid s, a \right]$$

Note that the $Q^*$ is uniquely characterized as the fixed point of this operator (Rust, 1994). That is, $Q^*$ is the unique solution to the Bellman optimality equation:

$$\mathcal{T}Q^* = Q^* \quad \text{or equivalently,} \quad (\mathcal{T}Q^*)(s, a) - Q^*(s, a) = 0.$$

The extent to which an arbitrary Q-function $Q$ fails to satisfy the Bellman optimality equation motivates the following definitions of error.

**Definition 1** (Bellman Error (Bellman Residual)). *For any function $Q \in \mathcal{Q}$, we define the Bellman error (or Bellman residual) at a state-action pair $(s, a)$ as the difference:*

$$(\mathcal{T}Q)(s, a) - Q(s, a)$$

The Bellman operator $\mathcal{T}$ cannot be computed directly without full knowledge of the system's transition dynamics, $P$. In reinforcement learning, sample transitions from the environment are used instead. This allows for the definition of a sample-based counterpart to $\mathcal{T}$, the *Sampled Bellman operator*, $\hat{\mathcal{T}}$. Given a single transition tuple $(s, a, s')$, this operator is defined as:

$$\hat{\mathcal{T}}Q(s, a, s') := r(s, a) + \beta \log \sum_{a' \in \mathcal{A}} \exp\left(Q(s', a')\right)$$

---

[4]This equivalence has been discussed in various Inverse Reinforcement Learning literature (Ermon et al., 2015; Zeng et al., 2025) and Dynamic Discrete Choice literature (Geng et al., 2020; Kang et al., 2025). For details, see Kang et al. (2025)

**Definition 2** (Temporal-Difference Error). *Using the sampled operator, we can define the Temporal-Difference (TD) error for a given transition $(s, a, s')$:*

$$\delta_Q(s, a, s') := \hat{\mathcal{T}}Q(s, a, s') - Q(s, a)$$

The connection between the Bellman error and the TD error is established in the following lemma. It shows that the TD error is an unbiased, single-sample estimate of the Bellman error.

**Lemma 1** (Relationship between Bellman and TD Errors). *For any $Q \in \mathcal{Q}$ and any state-action pair $(s, a)$, the expectation of the Sampled Bellman operator over the next state $s'$ recovers the original Bellman operator:*

$$\mathbb{E}_{s' \sim P(\cdot|s,a)} \left[ \hat{\mathcal{T}}Q(s, a, s') \right] = (\mathcal{T}Q)(s, a)$$

*Consequently, the expected TD error is equal to the Bellman error:*

$$\mathbb{E}_{s' \sim P(\cdot|s,a)} [\delta_Q(s, a, s')] = (\mathcal{T}Q)(s, a) - Q(s, a)$$

**Bellman Residual Minimization.** Note that both Bellman error (Bellman residual) and its proxy, the TD error, are functions of $(s, a)$. To find $Q$ that minimizes the Bellman error for all $(s, a)$, we can instead find $Q$ that minimizes expected square error on the offline data distribution. That is, we first define the *Squared Bellman Error* at $(s, a)$ as $\mathcal{L}_{\mathrm{BE}}(Q)(s, a) := ((\mathcal{T}Q)(s, a) - Q(s, a))^2$ and minimize the *Mean Squared Bellman Error* (MSBE), defined as:

$$\overline{\mathcal{L}_{\mathrm{BE}}}(Q) := \mathbb{E}_{(s,a) \sim \pi_D, \nu_0} [\mathcal{L}_{\mathrm{BE}}(Q)(s, a)]$$

where $\pi_D$ is the policy used for collecting data. Furthermore, as a proxy for Squared Bellman Error, we define the *Squared TD Error*: $\mathcal{L}_{\mathrm{TD}}(Q)(s, a, s') := \delta_Q(s, a, s')^2$ and minimize the *Mean Squared TD Error* (MSTDE) as a proxy for MSBE, defined as:

$$\overline{\mathcal{L}_{\mathrm{TD}}}(Q) := \mathbb{E}_{(s,a) \sim \pi_D, \nu_0} \left[ \mathbb{E}_{s' \sim P(\cdot|s,a)} [\mathcal{L}_{\mathrm{TD}}(Q)(s, a, s')] \right]$$

Unfortunately, MSTDE is a *biased* proxy for MSBE. This bias happens because expectation and square are not exchangeable, i.e., $\mathbb{E}_{s' \sim P(s,a)} [\delta_Q(s, a, s') \mid s, a]^2 \neq \mathbb{E}_{s' \sim P(s,a)} \left[ \delta_Q(s, a, s')^2 \mid s, a \right]$. This issue is often called the *double sampling problem* (Antos et al., 2008). Specifically, one can show that

$$\mathcal{L}_{BE}(Q)(s, a) = \mathbb{E}_{s' \sim P(s,a)} [\mathcal{L}_{TD}(Q)(s, a, s')] - \mathbb{E}_{s' \sim P(s,a)} \left[ \left( \mathcal{T}Q(s, a) - \hat{\mathcal{T}}Q(s, a, s') \right)^2 \right]$$

$$= \mathbb{E}_{s' \sim P(s,a)} [\mathcal{L}_{TD}(Q)(s, a, s')] - \beta^2 \mathbb{E}_{s' \sim P(s,a)} \left[ \left( V_Q(s') - \mathbb{E}_{s' \sim P(s,a)} [V_Q(s')] \right)^2 \right]$$

where $V_Q(s) := \ln \left[ \sum_{a \in \mathcal{A}} \exp(Q(s, a)) \right]$. (For the detailed derivation, see (Kang et al., 2025, Appendix C.1).) Since the bias term includes the $\mathbb{E}_{s' \sim P(s,a)}$ part, correcting this bias term again remains challenging without full knowledge of the system's transition dynamics, $P$. To resolve this issue, we employ an approach often referred to as the "Bi-Conjugate Trick" (Antos et al., 2008; Dai et al., 2018; Patterson et al., 2022):

$$\mathcal{L}_{BE}(Q)(s, a) = \mathbb{E}_{s' \sim P(s,a)} [\delta_Q(s, a, s') \mid s, a]^2$$
$$= \max_{h \in \mathbb{R}} 2 \cdot \mathbb{E}_{s' \sim P(s,a)} [\delta_Q(s, a, s') \mid s, a] \cdot h - h^2$$

According to (Kang et al., 2025, Appendix C.1), this bi-conjugate form can be re-parametrized using $\zeta := h - r(s, a) + Q(s, a)$ as:

$$\mathcal{L}_{BE}(Q)(s, a) = \mathbb{E}_{s' \sim P(s,a)} [\mathcal{L}_{TD}(Q)(s, a, s')] - \beta^2 \min_{\zeta \in \mathbb{R}} \mathbb{E}_{s' \sim P(s,a)} \left[ (V_Q(s') - \zeta)^2 \mid s, a \right] \quad (1)$$

Therefore, minimizing MSBE can be written as the following min-max problem:

$$\min_{Q \in \mathcal{Q}} \left\{ \mathbb{E}_{(s,a) \sim \pi_D, \nu_0} \left[ \mathbb{E}_{s' \sim P(\cdot|s,a)} \left[ \mathcal{L}_{\mathrm{TD}}(Q)(s, a, s') - \beta^2 \left( V_Q(s') - \tilde{\zeta} \right)^2 \mid s, a \right] \right] \right\} \quad (2)$$

where $\widetilde{\zeta} \in \arg\min_{\zeta \in \mathbb{R}^{S \times A}} \mathbb{E}_{(s,a) \sim \pi_D, \nu_0} \left[ \mathbb{E}_{s' \sim P(s,a)} \left[ (V_Q(s') - \zeta(s, a))^2 \right] \right]$. By parametrizing $\zeta$ as $\zeta_v$ and $Q$ as $Q_w$ by function classes such as Neural Networks, Equation equation 2 can be written as

$$\min_{w \in \mathcal{W}} \left\{ \mathbb{E}_{(s,a) \sim \pi_D, \nu_0} \left[ \mathbb{E}_{s' \sim P(\cdot|s,a)} \left[ \mathcal{L}_{\mathrm{TD}}(Q_w)(s, a, s') - \beta^2 \left( V_{Q_w}(s') - \zeta_{\tilde{v}} \right)^2 \mid s, a \right] \right] \right\} \quad (3)$$

where $\tilde{v} \in \mathrm{argmin}_{v \in \mathcal{V}} \, \mathbb{E}_{(s,a) \sim \pi_D, \nu_0} \left[ \mathbb{E}_{s' \sim P(s,a)} \left[ (V_{Q_w}(s') - \zeta_v(s,a))^2 \right] \right]$. Considering Equation equation 3 as the expected risk minimization problem, the corresponding empirical risk minimization problem can be written as

$$\min_{w \in \mathcal{W}} \frac{1}{N} \sum_{(s,a,s') \in \mathcal{D}_{\pi_D, \nu_0}} \left[ \mathcal{L}_{\mathrm{TD}}(Q_w)(s,a,s') - \beta^2 \left( V_{Q_w}(s') - \zeta_{\tilde{v}} \right)^2 \right] \tag{4}$$

$$\text{where } \tilde{v} \in \mathrm{argmin}_{v \in \mathcal{V}} \frac{1}{N} \sum_{(s,a,s') \in \mathcal{D}_{\pi_D, \nu_0}} \left[ \left( V_{Q_w}(s') - \zeta_v(s,a) \right)^2 \right] \tag{5}$$

and $\mathcal{D}_{\pi_D, \nu_0}$ is the offline data collected from following $\pi_D$ starting from $\nu_0$.

A canonical way of solving the mini-max problem is to apply the Stochastic Gradient Ascent Descent (SGDA) algorithm (Yang et al., 2020). Kang et al. (2025) proved that both equation 3 and equation 4 satisfy the Polyak-Łojasiewicz condition within a large enough ball around the initialization point for a Neural Network parametrization of $Q$ with a sufficient network width (width scaling with radius$^{\text{depth}}$), and therefore the SGDA algorithm finds the $(w, v)$ that are global minima of Equation equation 4. In the following Section 2.2, we elaborate on the SGDA algorithm.

## 2.2 STOCHASTIC GRADIENT ASCENT–DESCENT ALGORITHM (SGDA)

As discussed earlier, Stochastic Gradient Ascent–Descent (SGDA) is the workhorse we use to solve the minimax problem equation 4. Given a function $f(w, v)$, at every iteration it performs a *descent* step on the primal variable $w$ and an *ascent* step on the dual variable $v$ using (possibly noisy) gradients computed from a minibatch of samples.

Let $\mathcal{D} = \{z_i\}_{i=1}^n$ denote the dataset, where each $z_i = (s_i, a_i, s_i')$ denotes a sample consisting of the current state, the action taken, and the resulting next state. Fix a minibatch size $B \in \{1, \ldots, n\}$. At round $t$ we draw an index set

$$I_t \subseteq [n], \qquad |I_t| = B,$$

either *with* or *without* replacement (our theory does not depend on this choice). With $f$ standing for the per–sample saddle objective introduced in equation 4, the averaged stochastic gradients are

$$g_t^w := \frac{1}{B} \sum_{i \in I_t} \nabla_w f(w_t, v_t; z_i), \qquad g_t^v := \frac{1}{B} \sum_{i \in I_t} \nabla_v f(w_t, v_t; z_i).$$

Unbiasedness is preserved: $\mathbb{E}[g_t^w] = \nabla_w F_D(w_t, v_t)$ and likewise for $v$, while the variance contracts by the usual $1/B$ factor. Using stepsize sequence $(\eta_t)_{t \geq 0}$, SGDA proceeds as

$$w_{t+1} = w_t - \eta_t \, g_t^w, \qquad v_{t+1} = v_t + \eta_t \, g_t^v.$$

The recursion can be written compactly as

$$(w_{t+1}, v_{t+1}) = (w_t, v_t) + \eta_t \left( -g_t^w, \, g_t^v \right),$$

which is the form used in the stability proofs of Section 3.

---

**Algorithm 1:** Minibatch SGDA on the empirical objective equation 4

**Input:** Dataset $\mathcal{D} = \{z_i\}_{i=1}^n$, minibatch size $B$, stepsizes $(\eta_t)$, initial $(w_0, v_0)$

1 **for** $t = 0$ **to** $T - 1$ **do**
2 $\quad$ Draw $I_t \subseteq [n]$ with $|I_t| = B$ uniformly at random;
3 $\quad g_t^w \leftarrow \frac{1}{B} \sum_{i \in I_t} \nabla_w f(w_t, v_t; z_i)$;
4 $\quad g_t^v \leftarrow \frac{1}{B} \sum_{i \in I_t} \nabla_v f(w_t, v_t; z_i)$;
5 $\quad w_{t+1} \leftarrow w_t - \eta_t g_t^w$; $\hspace{5cm}$ // gradient *descent*
6 $\quad v_{t+1} \leftarrow v_t + \eta_t g_t^v$; $\hspace{5.3cm}$ // gradient *ascent*

**Output:** $(w_T, v_T)$

---

The choice of $(\eta_t)$ follows the same Robbins–Monro conditions detailed after Theorem 3, and our theoretical bounds reflect the $1/B$ variance reduction exactly as discussed in the remark preceding equation 19 later. In practice, moderate minibatch sizes (e.g. $B \in [64, 512]$) strike a good

balance between numerical stability and computational throughput when training neural-network parameterizations.

Along with harmonic–stepsize, we can state the global convergence guarantee of ALGORITHM 1 in the parameter space:

**Lemma 2** (Global convergence of minibatch SGDA in parameter space (Yang et al., 2020; Kang et al., 2025)). *Let the iterates in* ALGORITHM 1 *be written as* $\{(w_t, v_t)\}_{t \geq 0}$, *where* $w_t$ *parametrises the action–value function* $Q_{w_t}$ *(primal variable) and* $v_t$ *parametrises* $\zeta_{v_t}$ *(dual variable). Choose the harmonic step-sizes* $\eta_t = \frac{c_1}{c_2 + t}, t \geq 0$ *with some constants* $c_1 > 0$ *and* $c_2 \geq 1$ *such that* $\eta_t \leq \min\{1/(4L), 1/\rho\}$ *for every t. Then the* ALGORITHM 1*'s output sequence* $\{(w_t, v_t)\}$ *converges almost surely to the unique saddle point of the empirical objective equation 4, where the suboptimality of empirical objective equation 4 is bounded by* $\frac{d_1}{d_2 + t}$ *for some constants* $d_1$ *and* $d_2$.

## 3 STABILITY AND GENERALIZATION FOR BELLMAN RESIDUAL MINIMIZATION

We quantify generalization through *algorithmic stability* for minimax learning. Algorithmic stability formalizes how sensitive a learning algorithm is to small changes in the training set: if replacing a single training example only slightly perturbs the algorithm's output, then the algorithm is said to be stable. The key fact is that stability implies generalization: algorithms that are stable on neighbouring datasets exhibit small discrepancies between their empirical risk (measured on finite training data) and population risk (measured on infinite unseen data). As shown in Wang et al. (2022), this principle carries over to minimax optimization. Here, the primal variable $w$ represents the model we care about (e.g. the value function $Q$), and the dual variable $v$ enforces constraints or auxiliary structure (e.g. the conjugate $\zeta$). Stability of the joint SGDA iterates $(w_T, v_T)$ under sample replacement ensures that both the *primal risk* and the *weak primal–dual gap* generalize from training data to the population distribution.

### 3.1 STABILITY

We consider an offline setting where the data may be dependently sampled (e.g., from a single trajectory in a Markov Decision Process), violating the standard i.i.d. assumption. In this case, as in Wang et al. (2022), the concept of *on-average algorithmic stability* is useful.

**Definition 3** (On-average algorithmic stability). *Let* $\mathcal{A}$ *be a randomized learning algorithm that maps a dataset* $\mathcal{D} = (z_1, \ldots, z_n)$ *to a parameter output* $\mathcal{A}(\mathcal{D})$. *For each* $i \in [n]$, *let* $\mathcal{D}^{(i)}$ *denote the replace-one neighbor of* $\mathcal{D}$, *where* $z_i$ *is replaced by an independent copy* $\tilde{z}_i$. *Then the* on-average *argument stability of* $\mathcal{A}$ *after* $T$ *iterations is*

$$\varepsilon_T := \frac{1}{n} \sum_{i=1}^{n} \mathbb{E}\big[\, \|\mathcal{A}(\mathcal{D}) - \mathcal{A}(\mathcal{D}^{(i)})\| \,\big],$$

*where the expectation is over the algorithm's internal randomness and the choice of* $i$.

Intuitively, $\varepsilon_T$ measures how much the algorithm's output changes when a single training point is replaced on average. We analyze the on-average argument stability of Stochastic Gradient Descent–Ascent (SGDA) for smooth–strongly concave saddle problems, following the stability framework for minimax optimization.

Throughout, we present all proofs for the *single-sample* ("minibatch-of-one") variant of SGDA. The extension to a minibatch of size $B \geq 1$ (or the full-batch, deterministic case $B = n$) is mechanical: every stochastic-gradient term is replaced by its averaged counterpart, which reduces all variance contributions by a factor $1/B$, while the probability that a particular data point appears in the update increases from $1/n$ to $B/n$. Consequently, every lemma and theorem below remains valid verbatim—with constants rescaled by these factors—and no new conceptual issues arise.

In this section, for the sake of generality, we use notations that generalize the Bellman residual minimization problem. Let $\mathcal{D} = (z_1, \ldots, z_n)$ be a dataset with $z_i \in \mathcal{Z}$. For any $i \in [n]$ and any $\tilde{z}_i \in \mathcal{Z}$, define the *replace-one neighbour* of $\mathcal{D}$ by

$$\mathcal{D}^{(i)} := (z_1, \ldots, z_{i-1}, \tilde{z}_i, z_{i+1}, \ldots, z_n).$$

We call $\mathcal{D}$ and $\mathcal{D}^{(i)}$ *neighbouring datasets*. Expectations averaged over $i$ are taken with $i \sim \mathrm{Unif}([n])$.

When comparing SGDA runs on $\mathcal{D}$ and $\mathcal{D}^{(i)}$, we couple them using the same minibatch index sequence $(i_t)_{t \geq 0}$ (shared-index coupling). This coupling is the key mechanism that allows the analysis to proceed without an i.i.d. assumption on the data. By synchronizing the minibatch selection, we neutralize it as a source of difference between the two runs. Consequently, the parameter trajectories diverge only on the infrequent steps where the replaced index $i$ is sampled (a "hit"). On all other steps, the updates are identical, and the optimization dynamics tend to pull the trajectories closer. The stability analysis thus bounds the cumulative effect of these rare "hits" by balancing their small, infrequent perturbations against the constant, contractive force of the optimization dynamics. This argument hinges entirely on the randomness of the sampling process, which makes the "hits" probabilistic, and not on the statistical independence of the data points, whose potential correlations are rendered irrelevant by the coupling.

For the stability analysis, it is convenient to momentarily step back from the specific Bellman–residual objective and view our problem as a generic empirical PL–strong-concave saddle-point problem. In the BRM formulation, the primal variable $w$ parametrizes the action–value function $Q_w$, the dual variable $v$ parametrizes the auxiliary function $\zeta_v$, and the empirical objective in equation 4 is an average over samples $z = (s, a, s')$ drawn from the offline dataset. In this section we abstract this structure and write $f(w, v; z)$ for the per–sample saddle loss and $F_D(w, v)$ for its empirical average over a dataset $D = \{z_i\}_{i=1}^n$. All assumptions (A1)–(A9) that we impose below are satisfied by the BRM objective under the conditions of Section 2 and the PL–strongly-concave properties established by Kang et al. (2025). Working in this slightly more abstract template keeps the proofs uncluttered; in Theorem 6 we then specialize the resulting stability and generalization bounds back to BRM.

Let SGDA iterates start from the same initialization $(w_0, v_0) = (w_0', v_0')$:

$$w_{t+1} := w_t - \eta_t \, \nabla_w f(w_t, v_t; z_{i_t}), \qquad v_{t+1} := v_t + \eta_t \, \nabla_v f(w_t, v_t; z_{i_t}),$$
$$w_{t+1}' := w_t' - \eta_t \, \nabla_w f(w_t', v_t'; z_{i_t}'), \qquad v_{t+1}' := v_t' + \eta_t \, \nabla_v f(w_t', v_t'; z_{i_t}'),$$

with same-index coupling of datasets: $z_j' = z_j$ for $j \neq i$ and $z_i' = \tilde{z}_i$. For $D \in \{\mathcal{D}, \mathcal{D}^{(i)}\}$, define

$$F_D(w, v) := \frac{1}{n} \sum_{j=1}^n f(w, v; z_j^D), \qquad \Phi_D(w) := \max_v F_D(w, v), \qquad \Phi_D^\star := \min_w \Phi_D(w).$$

We let $\mathcal{F}_t := \sigma\big((w_s, v_s, w_s', v_s', i_s)_{0 \leq s \leq t}\big)$ be the natural filtration and introduce a *ghost* index $\hat{i}_t \sim \mathrm{Unif}(\{1, \ldots, n\})$ independent of $\mathcal{F}_t$, shared by both runs. The role of the ghost index is to decouple the sampling noise at time $t$ from the past and from the coupling across datasets.

**Assumptions.**

(A1) **Smoothness.** $F_D$ is $L$-smooth in the joint variable $(w, v)$.

(A2) **PL for $\Phi_D$.** $\Phi_D$ satisfies the Polyak–Łojasiewicz (PL) inequality with parameter $\mu_{\mathrm{PL}} > 0$: $\frac{1}{2}\|\nabla\Phi_D(w)\|^2 \geq \mu_{\mathrm{PL}}(\Phi_D(w) - \Phi_D^\star)$.

(A3) **QG for $\Phi_D$.** $\Phi_D$ satisfies Quadratic Growth (QG) with parameter $\mu_{\mathrm{QG}} > 0$: $\Phi_D(w) - \Phi_D^\star \geq \frac{\mu_{\mathrm{QG}}}{2}\|w - x_D^\star\|^2$.

(A4) **Strong concavity in $v$.** $F_D(\cdot, \cdot)$ is $\rho$-strongly concave in $v$ uniformly in $w$.

(A5) **Bounded gradients on the effective domain.** There exists a compact convex set $\Omega \subset \mathcal{W} \times \mathcal{V}$ such that the sequence of iterates $\{(w_t, v_t)\}_{t=0}^T$ generated by the algorithm remains within $\Omega$ almost surely. We define $G$ as the uniform gradient bound on this set:
$$G := \sup_{(w,v) \in \Omega, z \in \mathcal{Z}} \max\{\|\nabla_w f(w, v; z)\|, \|\nabla_v f(w, v; z)\|\} < \infty.$$

(A6) **Stepsizes.** $0 < \eta_t \leq \min\{\frac{1}{4L}, \frac{1}{\rho}\}$.

(A7) **Shared-index coupling (no i.i.d. needed).** The two coupled runs on $\mathcal{D}$ and $\mathcal{D}^{(i)}$ use the *same* index sequence $(i_t)_{t \geq 0}$.

(A8) **Uniformity of constants across datasets.** The constants $L, \rho, \mu_{\mathrm{PL}}, \mu_{\mathrm{QG}}, G$ are the same for $D$ and $D^{(i)}$.

(A9) **Unique saddle point.** Each dataset $D$ admits a unique saddle point $(x_D^\star, v_D^\star)$.

Kang et al. (2025) proved that Assumptions (A1), (A2), (A4) hold and Yang et al. (2020) proved Assumption (A9) holds for the problem of minimising Equation equation 4. In addition, they showed that the Equation equation 4 is of $\mathcal{C}^2$ and therefore the equivalence of PL and QG holds by Liao et al. (2024), satisfying Assumption (A3). Assumption (A5) is justified by the coercivity of the PL-Strongly Concave landscape, which ensures iterates remain in a bounded sublevel set (Yang et al., 2020). While strong concavity implies unbounded gradients on $\mathbb{R}^d$, the geometry of problem equation 4 induces a drift that keeps iterates bounded (coercivity). Thus, we only require the gradient to be bounded within the effective domain $\Omega$ visited by the algorithm, consistent with the global convergence guarantees for unprojected SGDA in this setting (Yang et al., 2020). The Assumption (A8) is standard in the stability literature: for example, Hardt et al. (2016) assume each per-example loss $f(\cdot; z)$ is $L$-Lipschitz and $\beta$-smooth uniformly in $z$, and Wang et al. (2022) assume the gradients and smoothness of $f(w, v; z)$ are bounded by global constants $G$ and $L$ for all $z$. These conditions immediately imply that the corresponding constants are identical for any dataset and its replace-one neighbour. Our Assumption A8 is the PL/QG analogue of these standard uniform assumptions.

To establish stability in our PL–strongly-concave setting, a key difficulty is that SGDA is not optimizing the value function $\Phi_D(w) := \max_v F_D(w, v)$ directly, but rather the saddle objective $F_D(w, v)$. The gradient used for the primal update, $\nabla_w F_D(w_t, v_t)$, coincides with $\nabla\Phi_D(w_t)$ only when the dual variable $v_t$ is already at its inner maximizer $v_D^\star(w_t)$; away from this manifold there is a non-negligible "mismatch" term $\Delta_t := \nabla_w F_D(w_t, v_t) - \nabla\Phi_D(w_t)$. As a result, tracking the PL suboptimality $\Phi_D(w_t) - \Phi_D^\star$ alone does not yield a contracting recursion under SGDA, while tracking only the dual gap $\Phi_D(w_t) - F_D(w_t, v_t)$ ignores how far the primal iterate is from the BRM minimizer. To address this, we define the following *Lyapunov potential*:

$$\Psi_{\alpha,D}(w,v) := \underbrace{\Phi_D(w) - \Phi_D^\star}_{A(w)} + \alpha \cdot \underbrace{\big(\Phi_D(w) - F_D(w,v)\big)}_{\Gamma(w,v)}, \qquad \alpha \in \left[\frac{4L^2}{\rho^2}, \infty\right).$$

Our Lyapunov potential is designed precisely to couple these two effects in a single scalar quantity: the PL term measures how close the current value function is to the data-dependent BRM optimum, and the dual-gap term penalizes the inner mismatch strongly enough that the beneficial drift from strong concavity in $v$ dominates the adverse contribution of $\Delta_t$ to the primal dynamics.

With a suitable choice of the weight $\alpha$, this potential contracts in expectation at each SGDA step, and can then be translated back into a bound on the distance between coupled SGDA trajectories via PL and QG. Because the step sizes satisfy the Robbins–Monro conditions $\sum_t \eta_t = \infty$ and $\sum_t \eta_t^2 < \infty$, the contraction dominates these perturbations and keeps the two trajectories close for large $t$. The next theorem formalizes this as an $\mathcal{O}(1/n)$ bound on the on-average argument stability of SGDA, stated in a general PL–strongly-concave minimax setting that our BRM formulation instantiates.

**Theorem 3** (On-average argument stability of SGDA without i.i.d. sampling). *Let* $\varepsilon_T := \frac{1}{n}\sum_{i=1}^n \mathbb{E}\big[\|w_T(\mathcal{D}) - w_T(\mathcal{D}^{(i)})\| + \|v_T(\mathcal{D}) - v_T(\mathcal{D}^{(i)})\|\big]$. *Under (A1)–(A8) and the choice of $\alpha$ above,*

$$\varepsilon_T \leq 2\,C_{\text{dist}} \sqrt{e^{-\frac{3c}{4}\sum_{s=0}^{T-1}\eta_s}\Psi_{\alpha,0}^{\max} + C_{\text{var}}\Big(L(1+L/\rho) + \alpha\frac{L^2}{\rho}\Big)G^2\sum_{t=0}^{T-1}\eta_t^2\,e^{-\frac{3c}{4}\sum_{s=t+1}^{T-1}\eta_s}}$$

$$+ \frac{2G}{n}\left(\frac{(1+L/\rho)^2}{\sqrt{\mu_{\text{PL}}\mu_{\text{QG}}}} + \frac{1}{\rho}\right) + \frac{C_{\text{hit}}}{n},$$

*where* $c := \min\{\mu_{\text{PL}}/2,\ \rho/2\}$, $C_{\text{dist}} = \sqrt{\left(1+\frac{L}{\rho}\right)^2\frac{2}{\mu_{\text{QG}}} + \frac{2}{\alpha\rho}}$, $C_{\text{var}} > 0$ *is a numerical constant,* $\Psi_{\alpha,0}^{\max} := \max\big\{\mathbb{E}[\Psi_{\alpha,\mathcal{D}}(w_0, v_0)],\ \max_{1\leq i\leq n}\mathbb{E}[\Psi_{\alpha,\mathcal{D}^{(i)}}(w_0, v_0)]\big\}$. *and with* $\beta := 2L$ *and the numerical constant* $\tilde{B}_1 = 8$ *introduced later,* $C_{\text{hit}} := \frac{2\tilde{B}_1 G}{\beta}$ *holds. The constant* $C_{\text{hit}} > 0$ *depends only on* $L, \rho, \mu_{\text{PL}}, \mu_{\text{QG}}, G$ *(its explicit formula is given at the end of the proof). Note that under the Robbins–Monro conditions* $\sum_t \eta_t = \infty$ *and* $\sum_t \eta_t^2 < \infty$, *the optimization term (the square root term) vanishes as* $T \to \infty$ *(Garrigos & Gower, 2023).*

**Remark 3.1** (Effect of minibatch size). *Theorem 3 is stated for the single-sample case ($B = 1$), where each SGDA update uses a single index $i_t \in [n]$ and the stochastic gradients are* $g_t^w =$

$\nabla_w f(w_t, v_t; z_{i_t})$ and $g_t^v = \nabla_v f(w_t, v_t; z_{i_t})$. *The same proof applies verbatim when we use mini-batches of size $B \geq 1$, i.e.*

$$g_t^w = \frac{1}{B} \sum_{j \in I_t} \nabla_w f(w_t, v_t; z_j), \qquad g_t^v = \frac{1}{B} \sum_{j \in I_t} \nabla_v f(w_t, v_t; z_j),$$

*where $I_t \subset [n]$ is a uniformly random subset of size $B$ (with or without replacement).*

*Conditioned on the dataset $\mathcal{D}$ and on $(w_t, v_t)$, the per-sample gradients $\{\nabla f(w_t, v_t; z_j)\}_{j=1}^n$ are deterministic vectors, and the only randomness comes from the subset $I_t$. Finite-population sampling bounds then yield*

$$\mathbb{E}\big[\|g_t^w - \nabla_w F_{\mathcal{D}}(w_t, v_t)\|^2 \,\big|\, \mathcal{D}, w_t, v_t\big] \leq \frac{C}{B} G^2,$$

*and analogously for $g_t^v$, for some numerical constant $C > 0$. In the proof of Theorem 3, this changes only the variance coefficient $C_{\mathrm{var}}$ to $C_{\mathrm{var}}/B$.*

To make the rate transparent, we now specialize the stepsizes to the harmonic Robbins–Monro rule $\eta_t = c_1/(c_2 + t)$, which Kang et al. (2025) chooses for Algorithm 1 to prove Lemma 2. This schedule satisfies Assumption (A6) for suitable $c_1 > 0$, $c_2 \geq 1$ and turns the kernel sums in Theorem 3 into closed forms. This yields the next corollary, which displays roughly $O(T^{-1/2})$ decay of the optimization term while keeping the $O(1/n)$ contribution explicit. The detailed upper bound can be found in the Appendix B.2.

**Corollary 4** (Informal bound under a harmonic stepsize schedule). *Choose the Robbins–Monro stepsizes in* ALGORITHM 1 *as $\eta_t = \frac{c_1}{c_2 + t}, t \geq 0$, with constants $c_1 > 0$ and $c_2 \geq 1$ small enough that $\eta_t \leq \min\{1/(4L),\, 1/\rho\}$ for all $t$. Let $c := \min\{\mu_{\mathrm{PL}}/2, \rho/2\}$. Then from Theorem 3, the stability constant $\varepsilon_T$ in Theorem 3 admits the stability bound scales as*

$$\varepsilon_T = O\left((c_2 + T)^{-\min\left\{\frac{1}{2}, \frac{3c\,c_1}{8}\right\}}\right) + O\left(\tfrac{1}{n}\right).$$

## 3.2 GENERALIZATION

In this section, we quantify generalization through algorithmic stability we derived in the previous section. Following the minimax stability framework of Wang et al. (2022), stability controls both the *primal function*, which is the Bellman residual, and the *weak primal–dual gap*. We first define these two risks, then invoke the transfer lemma that turns our stability bound from Theorem 3 into generalization guarantees. Specifically, our goal is to 1) bound the difference between population Bellman–residual risk and empirical Bellman–residual risk and 2) bound the population Bellman–residual risk of the SGDA output.

**Definition 4** (Primal Risk). *Under (A4), define the* value function

$$R(w) := \max_{v \in \mathcal{V}} F(w, v), \qquad R_n(w) := \max_{v \in \mathcal{V}} F_{\mathcal{D}}(w, v).$$

**Definition 5** (Weak primal–dual risk). *For $(w, v) \in \mathcal{W} \times \mathcal{V}$, define the* population *and* empirical *weak–PD risks by*

$$\Delta^{\mathrm{PD}}(w, v) := \max_{v' \in \mathcal{V}} F(w, v') - \min_{w' \in \mathcal{W}} F(w', v), \qquad \Delta_n^{\mathrm{PD}}(w, v) := \max_{v' \in \mathcal{V}} F_{\mathcal{D}}(w, v') - \min_{w' \in \mathcal{W}} F_{\mathcal{D}}(w', v).$$

The key transfer principle is stability $\Rightarrow$ generalization for minimax problems: if the SGDA iterate $(w_T, v_T)$ has on-average argument stability $\varepsilon_T$ on neighboring datasets, then the primal value-function gap $\mathbb{E}\big[R(w_T) - R_n(w_T)\big]$ and the weak primal–dual gap $\big|\mathbb{E}\Delta^{\mathrm{PD}}(w_T, v_T) - \Delta_n^{\mathrm{PD}}(w_T, v_T)\big|$ can be effectively upper bounded by constant times $\varepsilon_T$ from Theorem 3, which is $\mathcal{O}(1/n)$. The remainder of this subsection formalizes this to apply this transfer with our stability bound (Theorem 3) to obtain Theorem 6, $\mathcal{O}(1/n)$ generalization for BRM under SGDA.

**Lemma 5** (Theorem 5, Wang et al. (2022)). *Let Assumptions (A1), (A4), and (A5) hold. Let $(w_T, v_T)$ be the SGDA iterates produced on $\mathcal{D}_n$ and let*

$$\varepsilon_T = \frac{1}{n} \sum_{i=1}^n \mathbb{E}\big[\|w_T(\mathcal{D}) - w_T(\mathcal{D}^{(i)})\| + \|v_T(\mathcal{D}) - v_T(\mathcal{D}^{(i)})\|\big]$$

*be the on-average argument-stability constant from Theorem 3. Then*

$$\mathbb{E}\big[R(w_T) - R_n(w_T)\big] \leq (1 + L/\rho)\, G\, \varepsilon_T, \quad \big|\mathbb{E}\big[\Delta^{\mathrm{PD}}(w_T, v_T) - \Delta_n^{\mathrm{PD}}(w_T, v_T)\big]\big| \leq G\, \varepsilon_T.$$

Combining Corollary 4 and Lemma 5 (Wang et al., 2022), we arrive at Theorem 6, the main result of this paper, i.e., the *generalization guarantee of Bellman residuals*. In words, the learned $Q$ (i.e., the corresponding learned $w$) generalizes: its empirical Bellman residual on the offline dataset closely matches its expected Bellman residual on the true MDP distribution. This is direct from the fact that proving $\mathcal{O}(1/n)$ stability for SGDA immediately delivers $\mathcal{O}(1/n)$ generalization bounds for Bellman residual minimization (Wang et al., 2022).

**Theorem 6** (Generalization for the empirical Bellman–residual objective). *Let $\big(\widehat{w}^{(T)}, \widehat{v}^{(T)}\big)$ be the parameters returned by* ALGORITHM 1 *after $T$ SGDA iterations on the empirical objective equation 4. Define the population and empirical risks*

$$\mathcal{R}(w) := \mathbb{E}_{(s,a)\sim\pi_D,\nu_0}\,\mathbb{E}_{s'\sim P(\cdot|s,a)}\Big[\mathcal{L}_{\mathrm{TD}}(Q_w)(s,a,s') - \beta^2\big(V_{Q_w}(s') - \zeta_{\tilde{v}^\star}(s,a)\big)^2\Big],$$

$$\widehat{\mathcal{R}}_n(w) := \frac{1}{N}\sum_{(s,a,s')\in\mathcal{D}_{\pi_D,\nu_0}}\Big[\mathcal{L}_{\mathrm{TD}}(Q_w)(s,a,s') - \beta^2\big(V_{Q_w}(s') - \zeta_{\tilde{v}^\star}(s,a)\big)^2\Big],$$

*where $\tilde{v}^\star$ is the minimizer in equation 5. Then*

$$\mathbb{E}\Big[\mathcal{R}\big(\widehat{w}^{(T)}\big) - \widehat{\mathcal{R}}_n\big(\widehat{w}^{(T)}\big)\Big] \;\le\; (1 + L/\rho)\,G\,\varepsilon_T,$$

*where $\varepsilon_T$ is from Theorem 3. Moreover,*

$$\Big|\mathbb{E}\Big[\Delta^{\mathrm{PD}}\big(\widehat{w}^{(T)},\widehat{v}^{(T)}\big) - \Delta_n^{\mathrm{PD}}\big(\widehat{w}^{(T)},\widehat{v}^{(T)}\big)\Big]\Big| \;\le\; G\,\varepsilon_T.$$

The generalization guarantee in Theorem 6 is a critical result, confirming that the empirical risk is a reliable proxy for the true population risk. However, the ultimate measure of success for a learning algorithm is its performance on the population distribution relative to the best possible model. This is quantified by the *population excess risk*, which measures the gap $\mathcal{R}(\widehat{w}^{(T)}) - \mathcal{R}(w^*)$.

**Theorem 7** (Population excess risk). *Let $\big(\widehat{w}^{(T)}, \widehat{v}^{(T)}\big)$ be the SGDA iterate outcome of Algorithm 1 after $T$ steps and $w^* := \arg\min_{w\in\mathcal{W}} \mathcal{R}(w)$ its population minimiser. Then, with $\varepsilon_T$ from Theorem 3,*

$$\mathbb{E}\Big[\mathcal{R}\big(\widehat{w}^{(T)}\big) - \mathcal{R}\big(w^*\big)\Big] \;\le\; \underbrace{(1+L/\rho)\,G\,\varepsilon_T}_{\text{stability / generalization}} \;+\; \underbrace{\frac{d_1}{d_2+T}}_{\text{optimization error}}$$

*where $d_1$ and $d_2$ are defined in Lemma 2.*

## 4 CONCLUSION

We studied the statistical behavior of Bellman Residual Minimization (BRM) in offline RL/IRL through the lens of stability. Exploiting the PL–strongly-concave geometry of the bi-conjugate formulation, we coupled two SGDA trajectories on neighboring datasets with a single Lyapunov potential and a ghost-index decoupling device. This yielded an *on-average argument-stability* bound with $\mathcal{O}(1/n)$ rate (Theorem 3), which directly implies $\mathcal{O}(1/n)$ generalization for BRM (Theorem 6) and a population excess-risk bound that cleanly decomposes optimization and estimation errors (Theorem 7). The analysis is constructive, tracks explicit constants in $(L, \rho, \mu_{\mathrm{PL}}, \mu_{\mathrm{QG}}, G)$, accommodates minibatching, and requires neither variance reduction nor independence assumptions on the sampling indices. Together with the global convergence of SGDA in parameter space (Lemma 2), these results close the statistical gap for BRM and improve the sample-complexity exponent over the $\mathcal{O}(n^{-1/2})$ rates known for convex–concave saddle problems.

## 5 ETHICS STATEMENT

We have carefully read the ICLR Code of Ethics and affirm our commitment to adhere to it throughout the submission and review process. Our work is purely theoretical and does not involve human subjects, sensitive data, or applications that could raise ethical concerns. Therefore, we believe that our submission does not present any potential violations of the ICLR Code of Ethics.

# 6 REPRODUCIBILITY STATEMENT

Our main contributions are theoretical results, for which we have clearly stated the assumptions and included complete proofs of the claims in the Appendix. The key assumptions underlying our analysis, labeled (A1)–(A7), are explicitly stated in the main body of the paper (page 7). Complete proofs of all our theorems are provided in the Appendix, and any lemmas cited from external sources are clearly referenced. In particular, the proofs of Theorem 3, Corollary 4, and Theorem 7 are included in Appendix B.2, while all supporting lemmas necessary for these proofs are given in Appendix B.1.

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

APPENDIX

# A NOTATIONS

| Symbol | Meaning |
|---|---|
| $\mathcal{S}$ | State space |
| $\mathcal{A}$ | Action space |
| $P(\cdot \vert s, a)$ | Transition kernal over the next state from the state-action pair $(s, a)$ |
| $r : \mathcal{S} \times \mathcal{A} \to \mathbb{R}$ | Deterministic reward function |
| $\Delta_{\mathcal{S}}^{\mathcal{A}}$ | Set of stationary Markov policies |
| $\pi \in \Delta_{\mathcal{S}}^{\mathcal{A}}$ | Policy |
| $\beta$ | Discount factor |
| $\mathcal{T}$ | Bellman optimality operator |
| $\hat{\mathcal{T}}$ | Sampled Bellman operator |
| $\delta_Q(s, a, s')$ | Temporal-Difference error |
| $Q^*$ | (The unique) solution to Bellman optimality equation |
| $z_i = (s_i, a_i, s_i')$ | $i$-th sample |
| $\mathcal{D} = \{z_i\}_{i=1}^n$ | Dataset |
| $f(w, v; z_i)$ | Per-sample objective |
| $\mathcal{D}^{(i)}$ | neighbouring dataset |
| $g_t^w, g_t^v$ | per-sample gradients. |
| $\eta_t$ | Learning rate at time $t$ |
| $F_D(w, v)$ | $\frac{1}{n} \sum_{j=1}^n f(w, v; z_j^D)$ |
| $\Phi_D(w)$ | $\max_v F_D(w, v)$ |
| $\Phi_D^\star$ | $\min_w \Phi_D(w)$ |
| $L$ | Smoothness (Assumption A1) |
| $\mu_{PL}$ | Polyak–Łojasiewicz condition constant (Assumption A2) |
| $\mu_{QG}$ | Quadratic Growth constant (Assumption A3) |
| $\rho$ | Strong concavity constant (Assumption A4) |
| $G$ | Per-sample gradient bound (Assumption A5) |
| $\Psi_{\alpha, D}(w, v)$ | Lyapunov potential, $\Phi_D(w) - \Phi_D^\star + \alpha \cdot \big(\Phi_D(w) - F_D(w, v)\big)$ |
| $A(w)$ | $\Phi_D(w) - \Phi_D^\star$ |
| $\Gamma(w, v)$ | $\Phi_D(w) - F_D(w, v)$ |
| $R(w), R_n(w)$ | Primal Risk (Definition 4) |
| $\Delta^{PD}, \Delta_n^{PD}$ | Weak primal-dual risk (Definition 5) |

Table 1: Notations

# B TECHNICAL PROOFS

## B.1 SUPPORTING LEMMAS FOR PROVING THEOREM 3

**Lemma 8** (Mismatch control). *Let $v_D^\star(w) := \arg\max_v F_D(w, v)$. Under (A1) and (A4),*

$$\Delta(w, v) := \nabla_w F_D(w, v) - \nabla \Phi_D(w) = \nabla_w F_D(w, v) - \nabla_w F_D\big(w, v_D^\star(w)\big)$$

*satisfies $\|\Delta(w, v)\| \le L \|v - v_D^\star(w)\|$. Moreover,*

$$\|v - v_D^\star(w)\|^2 \ \le \ \frac{2}{\rho} \big(\Phi_D(w) - F_D(w, v)\big).$$

*Proof.* $L$-Lipschitzness of $\nabla_w F_D(w, \cdot)$ yields the first bound. For $g(\cdot) := F_D(w, \cdot)$, $\rho$-strong concavity implies $g(v^\star) - g(v) \ge (\rho/2)\|v - v^\star\|^2$ (Nesterov, 2004), which gives the second inequality. $\square$

**Lemma 9** (Smoothness of the value function $\Phi$). *Assume (A1) (joint $L$-smoothness) and (A4) ($\rho$-strong concavity in $v$). Then, for each dataset $D$:*

(i) *the maximizer $v_D^\star(w) := \arg\max_v F_D(w,v)$ is well-defined and $(L/\rho)$-Lipschitz:*

$$\|v_D^\star(w) - v_D^\star(u)\| \;\le\; \tfrac{L}{\rho}\|w-u\| \quad \forall\, w, u,$$

(ii) $\Phi_D(w) := \max_v F_D(w,v)$ *is differentiable with* $\nabla\Phi_D(w) = \nabla_w F_D\big(w, v_D^\star(w)\big)$ *(Danskin),*

(iii) *and $\Phi_D$ is $L_\Phi$-smooth with*

$$\|\nabla\Phi_D(w) - \nabla\Phi_D(u)\| \;\le\; L\Big(1 + \tfrac{L}{\rho}\Big)\|w-u\| \quad \forall\, w, u.$$

*Proof. (i) Lipschitzness of $v_D^\star(\cdot)$.* By (A4), for each fixed $w$ the map $v \mapsto F_D(w,v)$ is $\rho$-strongly concave, so $v_D^\star(w)$ is unique. Using the first-order conditions $\nabla_v F_D(w, v_D^\star(w)) = 0$ and $\nabla_v F_D(u, v_D^\star(u)) = 0$, write

$$0 \;=\; \nabla_v F_D(w, v_D^\star(w)) - \nabla_v F_D(u, v_D^\star(u))$$

$$= \underbrace{\big[\nabla_v F_D(w, v_D^\star(w)) - \nabla_v F_D(w, v_D^\star(u))\big]}_{(A)} + \underbrace{\big[\nabla_v F_D(w, v_D^\star(u)) - \nabla_v F_D(u, v_D^\star(u))\big]}_{(B)}.$$

Strong concavity makes $\nabla_v F_D(w, \cdot)$ $\rho$-strongly *monotone*, so $\langle (A),\, v_D^\star(w) - v_D^\star(u)\rangle \le -\rho\|v_D^\star(w) - v_D^\star(u)\|^2$. Joint $L$-smoothness gives $\|(B)\| \le L\|w - u\|$. Taking inner products with $v_D^\star(w) - v_D^\star(u)$ yields

$$\rho\|v_D^\star(w) - v_D^\star(u)\| \;\le\; \|(B)\| \;\le\; L\|w-u\| \quad \Rightarrow \quad \|v_D^\star(w) - v_D^\star(u)\| \;\le\; \tfrac{L}{\rho}\|w-u\|.$$

*(ii)* By uniqueness of $v_D^\star(w)$ and joint smoothness, Danskin's theorem applies: $\nabla\Phi_D(w) = \nabla_w F_D\big(w, v_D^\star(w)\big)$.

*(iii) Smoothness of $\Phi_D$.* For any $w, u$,

$$\|\nabla\Phi_D(w) - \nabla\Phi_D(u)\|$$
$$= \big\|\nabla_w F_D\big(w, v_D^\star(w)\big) - \nabla_w F_D\big(u, v_D^\star(u)\big)\big\|$$
$$\le \underbrace{\big\|\nabla_w F_D\big(w, v_D^\star(w)\big) - \nabla_w F_D\big(w, v_D^\star(u)\big)\big\|}_{\le L\|v_D^\star(w) - v_D^\star(u)\|} + \underbrace{\big\|\nabla_w F_D\big(w, v_D^\star(u)\big) - \nabla_w F_D\big(u, v_D^\star(u)\big)\big\|}_{\le L\|w-u\|}$$
$$\le L\Big(\tfrac{L}{\rho} + 1\Big)\|w-u\|.$$

The first two inequalities use joint $L$-smoothness (of $\nabla_w F_D$ in both arguments); the last uses part (i). Thus $L_\Phi \le L(1 + L/\rho)$. $\qquad\square$

**Lemma 10** (Cross-dataset gradient sensitivity for $\Phi$). *Under (A1), (A4) and (A5), for any $w, u$ and any 1-sample replacement $D \to D^{(i)}$,*

$$\|\nabla\Phi_D(w) - \nabla\Phi_{D^{(i)}}(u)\| \;\le\; L_\Phi\|w-u\| \;+\; \frac{2G}{n}\Big(1 + \frac{L}{\rho}\Big).$$

*Proof.* By Lemma 9, $v_D^\star(\cdot)$ is $(L/\rho)$-Lipschitz, $\Phi_D$ is $L_\Phi$-smooth, and $\nabla\Phi_D(w) = \nabla_w F_D\big(w, v_D^\star(w)\big)$. Hence

$$\|\nabla\Phi_D(w) - \nabla\Phi_{D^{(i)}}(u)\| \;\le\; \underbrace{\|\nabla\Phi_D(w) - \nabla\Phi_D(u)\|}_{\le L_\Phi\|w-u\|} + \underbrace{\|\nabla\Phi_D(u) - \nabla\Phi_{D^{(i)}}(u)\|}_{(A)}.$$

We bound (A) by inserting and subtracting $v_{D^{(i)}}^\star(u)$ and using joint $L$-smoothness of $F$:

$$(A) = \big\|\nabla_w F_D\big(u, v_D^\star(u)\big) - \nabla_w F_{D^{(i)}}\big(u, v_{D^{(i)}}^\star(u)\big)\big\|$$
$$\le \underbrace{\big\|\nabla_w F_D\big(u, v_D^\star(u)\big) - \nabla_w F_D\big(u, v_{D^{(i)}}^\star(u)\big)\big\|}_{\le L\,\|v_D^\star(u) - v_{D^{(i)}}^\star(u)\|} + \underbrace{\big\|\nabla_w F_D\big(u, v_{D^{(i)}}^\star(u)\big) - \nabla_w F_{D^{(i)}}\big(u, v_{D^{(i)}}^\star(u)\big)\big\|}_{\le \frac{2G}{n}\ \text{by Assumption (A5)}},$$

so that

$$(A) \;\leq\; L\,\|v_D^\star(u) - v_{D^{(i)}}^\star(u)\| + \frac{2G}{n}.$$

Since $v \mapsto F_D(u, v)$ is $\rho$-strongly concave, the gradient map is $\rho$-strongly monotone, which gives the error bound with the normal cone $N_\mathcal{V}(\cdot)$:

$$\|v - v_D^\star(u)\| \;\leq\; \frac{1}{\rho}\,\mathrm{dist}\big(\nabla_v F_D(u, v),\, N_\mathcal{V}(v)\big) \qquad \forall v \in \mathcal{V}.$$

Apply this at $v = v_{D^{(i)}}^\star(u)$. There are two cases.

*(i) Unconstrained (or interior) maximizer.* Then $N_\mathcal{V}\big(v_{D^{(i)}}^\star(u)\big) = \{0\}$, so

$$\|v_D^\star(u) - v_{D^{(i)}}^\star(u)\| \;\leq\; \frac{1}{\rho}\,\|\nabla_v F_D(u, v_{D^{(i)}}^\star(u))\| \;=\; \frac{1}{\rho}\,\|\nabla_v F_D(u, v_{D^{(i)}}^\star(u)) - \nabla_v F_{D^{(i)}}(u, v_{D^{(i)}}^\star(u))\|.$$

*(ii) Constrained boundary maximizer.* By KKT optimality,

$$0 \;\in\; -\nabla_v F_{D^{(i)}}(u, v_{D^{(i)}}^\star(u)) + N_\mathcal{V}(v_{D^{(i)}}^\star(u)) \iff \nabla_v F_{D^{(i)}}(u, v_{D^{(i)}}^\star(u)) \in N_\mathcal{V}(v_{D^{(i)}}^\star(u)).$$

Hence, choosing $\xi := \nabla_v F_{D^{(i)}}(u, v_{D^{(i)}}^\star(u)) \in N_\mathcal{V}(v_{D^{(i)}}^\star(u))$, we obtain

$$\mathrm{dist}\big(\nabla_v F_D(u, v_{D^{(i)}}^\star(u)),\, N_\mathcal{V}(v_{D^{(i)}}^\star(u))\big) \;\leq\; \big\|\nabla_v F_D(u, v_{D^{(i)}}^\star(u)) - \nabla_v F_{D^{(i)}}(u, v_{D^{(i)}}^\star(u))\big\|.$$

In either case, using the single-sample replacement identity

$$\nabla_v F_D(u, v) - \nabla_v F_{D^{(i)}}(u, v) \;=\; \frac{1}{n}\Big(\nabla_v f(u, v; z_i) - \nabla_v f(u, v; \tilde{z}_i)\Big),$$

the triangle inequality and Assumption (A5) give

$$\begin{aligned}
\|v_D^\star(u) - v_{D^{(i)}}^\star(u)\| &\leq \frac{1}{\rho}\,\mathrm{dist}\big(\nabla_v F_D(u, v_{D^{(i)}}^\star(u)),\, N_\mathcal{V}(v_{D^{(i)}}^\star(u))\big) \\
&\leq \frac{1}{\rho}\,\big\|\nabla_v F_D(u, v_{D^{(i)}}^\star(u)) - \nabla_v F_{D^{(i)}}(u, v_{D^{(i)}}^\star(u))\big\| \\
&= \frac{1}{\rho} \cdot \frac{1}{n}\,\big\|\nabla_v f(u, v_{D^{(i)}}^\star(u); z_i) - \nabla_v f(u, v_{D^{(i)}}^\star(u); \tilde{z}_i)\big\| \\
&\leq \frac{1}{\rho} \cdot \frac{1}{n}\,\Big(\|\nabla_v f(u, v_{D^{(i)}}^\star(u); z_i)\| + \|\nabla_v f(u, v_{D^{(i)}}^\star(u); \tilde{z}_i)\|\Big) \\
&\leq \frac{2G}{\rho n}\,.
\end{aligned}$$

Plugging this into the bound for (A) yields

$$(A) \;\leq\; L \cdot \frac{2G}{\rho n} + \frac{2G}{n} \;=\; \frac{2G}{n}\Big(1 + \frac{L}{\rho}\Big),$$

and therefore

$$\|\nabla \Phi_D(w) - \nabla \Phi_{D^{(i)}}(u)\| \;\leq\; L_\Phi \|w - u\| + \frac{2G}{n}\Big(1 + \frac{L}{\rho}\Big).$$

$\square$

**Lemma 11** (Deterministic potential $\Rightarrow$ distance). *Under (A3) and (A4), for any $\alpha > 0$ and any $(w, v)$,*

$$\|w - x_D^\star\| + \|v - v_D^\star\| \;\leq\; \sqrt{\Big(1 + \frac{L}{\rho}\Big)^2 \frac{2}{\mu_{\mathrm{QG}}} + \frac{2}{\alpha\rho}} \cdot \sqrt{\Psi_{\alpha, D}(w, v)}.$$

*Proof.* Using $\|v - v_D^\star\| \leq \|v - v_D^\star(w)\| + \|v_D^\star(w) - v_D^\star\|$ and the $(L/\rho)$-Lipschitzness of $w \mapsto v_D^\star(w)$,

$$\|w - x_D^\star\| + \|v - v_D^\star\| \leq (1 + \tfrac{L}{\rho})\|w - x_D^\star\| + \|v - v_D^\star(w)\|.$$

$$\|w - x_D^\star\| \leq \sqrt{\tfrac{2}{\mu_{\mathrm{QG}}}\big(\Phi_D(w) - \Phi_D^\star\big)}, \qquad \|v - v_D^\star(w)\| \leq \sqrt{\tfrac{2}{\rho}\big(\Phi_D(w) - F_D(w,v)\big)},$$

where $v_D^\star(w) \in \arg\max_u F_D(w,u)$ and $\Phi_D(w) = \max_u F_D(w,u)$.

Using the weighted Cauchy–Schwarz inequality with any $\alpha > 0$,

$$(1 + \tfrac{L}{\rho})\sqrt{\tfrac{2}{\mu_{\mathrm{QG}}}\big(\Phi_D(w) - \Phi_D^\star\big)} + \sqrt{\tfrac{2}{\rho}\big(\Phi_D(w) - F_D(w,v)\big)}$$

$$\leq \sqrt{\big(\Phi_D(w) - \Phi_D^\star\big) + \alpha\big(\Phi_D(w) - F_D(w,v)\big)}\,\sqrt{\tfrac{2(1+L/\rho)^2}{\mu_{\mathrm{QG}}} + \tfrac{2}{\alpha\rho}}.$$

Noting that $\Psi_{\alpha,D}(w,v) := \big(\Phi_D(w) - \Phi_D^\star\big) + \alpha\big(\Phi_D(w) - F_D(w,v)\big)$, we obtain

$$(1 + \tfrac{L}{\rho})\sqrt{\tfrac{2}{\mu_{\mathrm{QG}}}\big(\Phi_D(w) - \Phi_D^\star\big)} + \sqrt{\tfrac{2}{\rho}\big(\Phi_D(w) - F_D(w,v)\big)} \ \leq \ \sqrt{\Psi_{\alpha,D}(w,v)}\,\sqrt{\tfrac{2(1+L/\rho)^2}{\mu_{\mathrm{QG}}} + \tfrac{2}{\alpha\rho}}.$$

$\square$

**Lemma 12** (Saddle-point sensitivity). *Let $D^{(i)}$ be obtained from $D$ by replacing one sample. Assume (A1) (joint $L$-smoothness), (A4) ($\rho$-strong concavity in $v$), (A5) (per-sample gradients bounded by $G$), (A2)–(A3) (PL and QG for $\Phi_D$), and (A8) (the same constants hold for $D$ and $D^{(i)}$). Let $x_D^\star \in \arg\min_w \Phi_D(w)$ and $x_{D^{(i)}}^\star \in \arg\min_w \Phi_{D^{(i)}}(w)$, and define $v_D^\star := v_D^\star(x_D^\star)$ and $v_{D^{(i)}}^\star := v_{D^{(i)}}^\star(x_{D^{(i)}}^\star)$. Then*

$$\|x_D^\star - x_{D^{(i)}}^\star\| + \|v_D^\star - v_{D^{(i)}}^\star\| \ \leq \ \frac{2G}{n}\left(\frac{(1+L/\rho)^2}{\sqrt{\mu_{\mathrm{PL}}\mu_{\mathrm{QG}}}} + \frac{1}{\rho}\right).$$

*Proof.* At $(x_{D^{(i)}}^\star, v_{D^{(i)}}^\star)$, only one summand in $F_D$ differs from $F_{D^{(i)}}$, and per-sample gradients are bounded by $G$. Hence,

$$\|\nabla_w F_D(x_{D^{(i)}}^\star, v_{D^{(i)}}^\star)\| = \|\nabla_w F_D(x_{D^{(i)}}^\star, v_{D^{(i)}}^\star) - \nabla_w F_D^{(i)}(x_{D^{(i)}}^\star, v_{D^{(i)}}^\star)\| \leq \tfrac{2G}{n}$$

$$\|\nabla_v F_D(x_{D^{(i)}}^\star, v_{D^{(i)}}^\star)\| = \|\nabla_v F_D(x_{D^{(i)}}^\star, v_{D^{(i)}}^\star) - \nabla_v F_D^{(i)}(x_{D^{(i)}}^\star, v_{D^{(i)}}^\star)\| \leq \tfrac{2G}{n}.$$

Using Lemma 8,

$$\|\nabla\Phi_D(x_{D^{(i)}}^\star)\| \leq \|\nabla_w F_D(x_{D^{(i)}}^\star, v_{D^{(i)}}^\star)\| + L\,\|v_{D^{(i)}}^\star - v_D^\star(x_{D^{(i)}}^\star)\| \leq \tfrac{2G}{n}(1 + L/\rho),$$

where the last step uses $\rho$-strong concavity to get $\|v_{D^{(i)}}^\star - v_D^\star(x_{D^{(i)}}^\star)\| \leq \tfrac{1}{\rho}\|\nabla_v F_D(x_{D^{(i)}}^\star, v_{D^{(i)}}^\star)\|$. Since $\Phi_D$ satisfies PL and QG,

$$\|x_D^\star - x_{D^{(i)}}^\star\| \leq \frac{1}{\sqrt{\mu_{\mathrm{PL}}\mu_{\mathrm{QG}}}}\,\|\nabla\Phi_D(x_{D^{(i)}}^\star)\| \leq \tfrac{2G}{n}\cdot\frac{1 + L/\rho}{\sqrt{\mu_{\mathrm{PL}}\mu_{\mathrm{QG}}}}.$$

For the dual variable, split and use the $(L/\rho)$-Lipschitzness of $w \mapsto v_D^\star(w)$ (Lemma 9) and strong concavity:

$$\|v_D^\star - v_{D^{(i)}}^\star\| \leq \|v_D^\star(x_D^\star) - v_D^\star(x_{D^{(i)}}^\star)\| + \|v_D^\star(x_{D^{(i)}}^\star) - v_{D^{(i)}}^\star(x_{D^{(i)}}^\star)\|$$

$$\leq \tfrac{L}{\rho}\|x_D^\star - x_{D^{(i)}}^\star\| + \tfrac{1}{\rho}\|\nabla_v F_D(x_{D^{(i)}}^\star, v_{D^{(i)}}^\star)\|$$

$$\leq \tfrac{2G}{n}\left(\tfrac{L}{\rho}\cdot\frac{1 + L/\rho}{\sqrt{\mu_{\mathrm{PL}}\mu_{\mathrm{QG}}}} + \tfrac{1}{\rho}\right),$$

where the last inequality uses the same single-sample replacement and gradient bound as above (cf. Lemma 10). Adding the two bounds yields the claim. $\square$

**Lemma 13** (Coupled one-step bounds). *Assume (A1)–(A8). Define*

$$\Xi_t \ := \ \Psi_{\alpha,\mathcal{D}}(w_t, v_t) + \Psi_{\alpha,\mathcal{D}^{(i)}}(w_t', v_t'), \qquad D_t \ := \ \|w_t - w_t'\| + \|v_t - v_t'\|.$$

*Let $c := \min\{\mu_{\mathrm{PL}}/2, \rho/2\}$ and $L_\Phi \leq L(1 + L/\rho)$ (Lemma 9). Then, for $\tilde{B}_0 = \tilde{B}_1 = 8$ and a constant $C_{\mathrm{var}} > 0$ that depends only on $L, \rho$, we have*

$$(\mathbf{P}) \quad \mathbb{E}[\Xi_{t+1} \mid \mathcal{F}_t] \ \leq \ (1 - c\eta_t)\,\Xi_t \ + \ A\,\eta_t^2$$

$$+ \ C_{\text{leak}} \, \eta_t \, D_t \ + \ C_* \, \tfrac{G}{n} \, \eta_t \ + \ C_{\text{lin}} \, G^2 \, \eta_t \ + \ \eta_t \big( \mathbf{1}\{i_t = i\} + \mathbf{1}\{\hat{i}_t = i\} \big) \tilde{B}_0 G^2, \tag{6}$$

$$(\mathbf{D}_{\text{weak}}) \quad \mathbb{E}\big[ D_{t+1} \mid \mathcal{F}_t \big] \ \leq \ (1 + \kappa \eta_t) \, D_t \ + \ a \, \eta_t \sqrt{\Xi_t} \ + \ \eta_t \, \mathbf{1}\{i_t = i\} \, \tilde{B}_1 G \ + \ \eta_t \, \frac{2G}{n} \Big( 1 + \frac{L}{\rho} \Big), \tag{7}$$

*where*

$$A := C_{\text{var}} \Big( L(1 + L/\rho) + \alpha L^2/\rho \Big) G^2, \quad C_{\text{leak}} := 2(2L + L_\Phi),$$

$$a := 2 C_{\text{dist}} \big( L + L_\Phi \big), \quad \kappa := 2L, \quad C_* := 4 \big( 1 + L/\rho \big), \quad C_{\text{dist}} := \sqrt{ \Big( 1 + \frac{L}{\rho} \Big)^2 \frac{2}{\mu_{\text{QG}}} + \frac{2}{\alpha \rho} },$$

$$and \qquad C_{\text{lin}} := 12 + \frac{10 L^2}{\rho^2} \, .$$

**Why introduce a ghost index?** Writing and conditioning on the past, the primal descent term $-\eta_t \langle \nabla \Phi(w_t), \nabla_w f(w_t, v_t; z_{i_t}) \rangle$ has no sign we can control, since $i_t$ is already revealed in $\mathcal{F}_t$. We therefore add–subtract a ghost gradient and take $\mathbb{E}_{\hat{i}_t}[\cdot \mid \mathcal{F}_t]$:

$$-\eta_t \langle \nabla \Phi(w_t), g_t^w \rangle = -\eta_t \langle \nabla \Phi(w_t), \hat{g}_t^w \rangle - \eta_t \langle \nabla \Phi(w_t), g_t^w - \hat{g}_t^w \rangle.$$

The first term becomes the correct drift $-\eta_t \langle \nabla \Phi(w_t), \nabla_w F(w_t, v_t) \rangle$, which contracts by PL for $\Phi$, while the second is a centered correction that we bound by Young's inequality and the gradient bound $\| g_t^w - \hat{g}_t^w \| \leq 2G$, yielding an $O(\eta_t G^2)$ remainder absorbed into the variance term. An identical maneuver applies to the dual-gap part $\Gamma$. This device avoids any i.i.d. sampling assumption and yields the one-step recursion (P) with a contractive factor and small, explicit noise.

*Proof.* All expectations are conditional on $\mathcal{F}_t := \sigma\big( (w_s, v_s, w_s', v_s', i_s)_{0 \leq s \leq t} \big)$. Write $\eta := \eta_t$. Fix $D \in \{\mathcal{D}, \mathcal{D}^{(i)}\}$ and abbreviate $F := F_D$, $\Phi := \Phi_D$, $\Psi_\alpha := \Psi_{\alpha, D}$, $\Gamma(w, v) := \Phi(w) - F(w, v)$. Set $g^w := \nabla_w f(w_t, v_t; z_{i_t})$, $g^v := \nabla_v f(w_t, v_t; z_{i_t})$,

$$w^+ = w_t - \eta g^w, \qquad v^+ = v_t + \eta g^v,$$

and introduce a ghost index $\hat{i}_t$ independent of $\mathcal{F}_t$ with $\hat{g}^w := \nabla_w f(w_t, v_t; z_{\hat{i}_t})$, $\hat{g}^v := \nabla_v f(w_t, v_t; z_{\hat{i}_t})$.

**(a) Primal part** $\Phi$. Recall $w^+ = w_t - \eta \, g^w$ with $g^w := \nabla_w f(w_t, v_t; z_{phi - smooth - step i_t})$ and let $\eta := \eta_t$. By $L_\Phi$-smoothness of $\Phi$,

$$\Phi(w^+) - \Phi(w_t) \ \leq \ -\eta \, \langle \nabla \Phi(w_t), \, g^w \rangle \ + \ \tfrac{L_\Phi}{2} \, \eta^2 \, \| g^w \|^2.$$

Insert and subtract a ghost gradient $\hat{g}^w := \nabla_w f(w_t, v_t; z_{\hat{i}_t})$ with $\hat{i}_t \perp \mathcal{F}_t$, then take $\mathbb{E}_{\hat{i}_t}[\cdot \mid \mathcal{F}_t]$. Using $\mathbb{E}_{\hat{i}_t}[\hat{g}^w \mid \mathcal{F}_t] = \nabla_w F(w_t, v_t)$ and $\| g^w \| \leq G$,

$$\mathbb{E}_{\hat{i}_t}\big[ \Phi(w^+) - \Phi(w_t) \mid \mathcal{F}_t \big] \leq -\eta \, \langle \nabla \Phi(w_t), \, \nabla_w F(w_t, v_t) \rangle \ - \ \eta \, \langle \nabla \Phi(w_t), \, g^w - \nabla_w F(w_t, v_t) \rangle \ + \ \tfrac{L_\Phi}{2} \eta^2 G^2. \tag{8}$$

For the centered correction, Young's inequality with $\| g^w - \nabla_w F(w_t, v_t) \| \leq 2G$ yields

$$\eta \, \big| \langle \nabla \Phi(w_t), \, g^w - \nabla_w F(w_t, v_t) \rangle \big| \ \leq \ \tfrac{1}{4} \, \eta \, \| \nabla \Phi(w_t) \|^2 \ + \ 4 \, \eta \, G^2. \tag{9}$$

For the drift, write $\Delta_t := \nabla_w F(w_t, v_t) - \nabla \Phi(w_t)$. Then

$$-\langle \nabla \Phi(w_t), \, \nabla_w F(w_t, v_t) \rangle = -\| \nabla \Phi(w_t) \|^2 \ - \ \langle \nabla \Phi(w_t), \, \Delta_t \rangle \ \leq \ -\tfrac{1}{2} \| \nabla \Phi(w_t) \|^2 \ + \ \tfrac{1}{2} \| \Delta_t \|^2. \tag{10}$$

By Lemma 8 and $\rho$-strong concavity in $v$,

$$\| \Delta_t \| \ \leq \ L \, \| v_t - v^\star(w_t) \| \ \leq \ L \sqrt{\tfrac{2}{\rho} \Gamma(w_t, v_t)} \qquad \Rightarrow \qquad \tfrac{1}{2} \| \Delta_t \|^2 \ \leq \ \tfrac{L^2}{\rho} \, \Gamma(w_t, v_t), \tag{11}$$

where $\Gamma(w, v) := \Phi(w) - F(w, v)$. Combining equation 8, equation 9, equation 10, and equation 11,

$$\mathbb{E}_{\hat{i}_t}\big[\Phi(w^+) - \Phi(w_t) \mid \mathcal{F}_t\big] \ \leq \ -\tfrac{1}{4}\,\eta\,\|\nabla\Phi(w_t)\|^2 \ + \ \tfrac{L^2}{\rho}\,\eta\,\Gamma(w_t, v_t) \ + \ \tfrac{L_\Phi}{2}\eta^2 G^2 \ + \ 4\,\eta\,G^2.$$

Finally, by the PL inequality $\tfrac{1}{2}\|\nabla\Phi(w_t)\|^2 \geq \mu_{\mathrm{PL}}\big(\Phi(w_t) - \Phi^\star\big)$,

$$-\tfrac{1}{4}\,\eta\,\|\nabla\Phi(w_t)\|^2 \ \leq \ -\tfrac{\mu_{\mathrm{PL}}}{2}\,\eta\,\big(\Phi(w_t) - \Phi^\star\big),$$

hence

$$\mathbb{E}_{\hat{i}_t}\big[\Phi(w^+) - \Phi(w_t) \mid \mathcal{F}_t\big] \ \leq \ -\tfrac{\mu_{\mathrm{PL}}}{2}\,\eta\,\big(\Phi(w_t) - \Phi^\star\big) + \tfrac{L^2}{\rho}\,\eta\,\Gamma(w_t, v_t) + \tfrac{L_\Phi}{2}\eta^2 G^2 + 4\,\eta\,G^2.$$

Since $\Phi(w_t) - \Phi^\star$ is $\mathcal{F}_t$-measurable, adding it to both sides yields

$$\mathbb{E}_{\hat{i}_t}\big[\Phi(w^+) - \Phi^\star \mid \mathcal{F}_t\big] \ \leq \ \big(1 - \tfrac{\mu_{\mathrm{PL}}}{2}\eta\big)\big(\Phi(w_t) - \Phi^\star\big) + \tfrac{L^2}{\rho}\,\eta\,\Gamma(w_t, v_t) + \tfrac{L_\Phi}{2}\eta^2 G^2 + 4\,\eta\,G^2.$$

**(b) Dual-gap part $\Gamma$.** Recall $\Gamma(w, v) := \Phi_D(w) - F_D(w, v)$ and write $v^+ = v_t + \eta\, g_t^v$ with $g_t^v := \nabla_v f(w_t, v_t; z_{i_t})$. By Lemma 14, for fixed $w_t$,

$$\Gamma(w_t, v^+) \leq (1 - 2\rho\eta + \rho L\eta^2)\,\Gamma(w_t, v_t). \tag{12}$$

Let $w^+ := w_t - \eta g_t^w$ with $g_t^w := \nabla_w f(w_t, v_t; z_{i_t})$. By $L_\Phi$-smoothness of $\Phi_D$ and joint $L$-smoothness of $F_D$,

$$\Gamma(w^+, v^+) \leq \Gamma(w_t, v^+) - \eta\langle\nabla\Phi_D(w_t), g_t^w\rangle + \eta\langle\nabla_w F_D(w_t, v^+), g_t^w\rangle + \tfrac{L_\Phi + L}{2}\eta^2 G^2.$$

Insert and subtract a ghost gradient $\hat{g}_t^w := \nabla_w f(w_t, v_t; z_{\hat{i}_t})$, take $\mathbb{E}_{\hat{i}_t}[\cdot \mid \mathcal{F}_t]$, and use $\mathbb{E}_{\hat{i}_t}[\hat{g}_t^w \mid \mathcal{F}_t] = \nabla_w F_D(w_t, v_t)$. Then

$$\begin{aligned}
\mathbb{E}_{\hat{i}_t}\big[\Gamma(w^+, v^+) \mid \mathcal{F}_t\big] \leq{}& \Gamma(w_t, v^+) - \eta\langle\nabla\Phi_D(w_t),\, \nabla_w F_D(w_t, v_t)\rangle \\
&+ \eta\big\langle\nabla_w F_D(w_t, v^+) - \nabla\Phi_D(w_t),\, \nabla_w F_D(w_t, v_t)\big\rangle \\
&+ \eta\big\langle\nabla_w F_D(w_t, v^+) - \nabla\Phi_D(w_t),\, g_t^w - \nabla_w F_D(w_t, v_t)\big\rangle \\
&+ \tfrac{L_\Phi + L}{2}\eta^2 G^2.
\end{aligned}$$

Decompose $\nabla_w F_D(w_t, v^+) - \nabla\Phi_D(w_t) = \Delta_t + \big(\nabla_w F_D(w_t, v^+) - \nabla_w F_D(w_t, v_t)\big)$, where $\Delta_t := \nabla_w F_D(w_t, v_t) - \nabla\Phi_D(w_t)$. Using Lemma 8, $\|\Delta_t\| \leq L\sqrt{2\Gamma(w_t, v_t)/\rho}$, joint $L$-smoothness, $\|\nabla_w F_D(w_t, v^+) - \nabla_w F_D(w_t, v_t)\| \leq L\eta G$, and $\|g_t^w - \nabla_w F_D(w_t, v_t)\| \leq 2G$,

$$\eta\big|\langle\Delta_t,\, g_t^w - \nabla_w F_D(w_t, v_t)\rangle\big| \ \leq \ \tfrac{\rho}{4}\eta\,\Gamma(w_t, v_t) + \tfrac{8L^2}{\rho^2}\eta G^2,$$

$$\eta\big|\langle\nabla_w F_D(w_t, v^+) - \nabla_w F_D(w_t, v_t),\, g_t^w - \nabla_w F_D(w_t, v_t)\rangle\big| \ \leq \ 2L\,\eta^2 G^2,$$

$$\eta\big|\langle\Delta_t,\, \nabla_w F_D(w_t, v_t)\rangle\big| \ \leq \ \tfrac{\rho}{4}\eta\,\Gamma(w_t, v_t) + \tfrac{2L^2}{\rho^2}\eta G^2,$$

$$\eta\big|\langle\nabla_w F_D(w_t, v^+) - \nabla_w F_D(w_t, v_t),\, \nabla_w F_D(w_t, v_t)\rangle\big| \ \leq \ L\,\eta^2 G^2.$$

Combining the bounds with the ascent contraction equation 12, we can group the contributions as follows:

(i) From the primal descent step we obtain

$$-\tfrac{1}{4}\,\eta\,\|\nabla\Phi_D(w_t)\|^2 \ \leq \ -\tfrac{\mu_{\mathrm{PL}}}{2}\,\eta\big(\Phi_D(w_t) - \Phi_D^\star\big),$$

where the inequality uses the PL condition. From the dual ascent we obtain

$$(1 - 2\rho\eta + \rho L\eta^2)\,\Gamma(w_t, v_t) = \Gamma(w_t, v_t) - 2\rho\eta\,\Gamma(w_t, v_t) + O(\eta^2),$$

so the leading negative part is $-2\rho\eta\,\Gamma(w_t, v_t)$. Together, these two negative terms contract the Lyapunov potential $\Psi_\alpha(w_t, v_t) = (\Phi_D(w_t) - \Phi_D^\star) + \alpha\Gamma(w_t, v_t)$, yielding a multiplicative shrinkage factor $(1 - c\eta)\Psi_\alpha(w_t, v_t)$ with $c := \min\{\mu_{\mathrm{PL}}/2, \rho/2\}$. Furthermore, the positive $\eta\Gamma$ pieces produced by the algebra consist of

$$\alpha \cdot \tfrac{\rho}{2}\,\eta\,\Gamma(w_t, v_t) \quad \text{and} \quad \tfrac{L^2}{\rho}\,\eta\,\Gamma(w_t, v_t).$$

Together with the dual drift they appear as

$$-2\alpha\rho\,\eta\,\Gamma \;+\; \alpha\cdot\tfrac{\rho}{2}\,\eta\,\Gamma \;+\; \tfrac{L^2}{\rho}\,\eta\,\Gamma \;=\; -\tfrac{3}{2}\alpha\rho\,\eta\,\Gamma \;+\; \tfrac{L^2}{\rho}\,\eta\,\Gamma.$$

Choose $\alpha$ so that $\frac{L^2}{\rho} \leq \frac{\alpha\rho}{2}$ (i.e. $\alpha \geq 2L^2/\rho^2$). Then

$$-\tfrac{3}{2}\alpha\rho\,\Gamma \;+\; \tfrac{L^2}{\rho}\,\eta\,\Gamma \;\leq\; -\tfrac{\alpha\rho}{2}\,\eta\,\Gamma,$$

so these $\eta\Gamma$ remainders are dominated by the dual drift and absorbed into the contraction factor. A simple sufficient standing choice is $\alpha \geq 4L^2/\rho^2$.

(ii) The mismatch bounds, ghost-correction terms, and gradient–difference terms contribute constants times $\eta G^2$. All such pieces are nonnegative and can be grouped into a single term $C_{\text{lin}}\,G^2\,\eta$.

(iii) Smoothness corrections (e.g. $(L_\Phi + L)/2\,\eta^2 G^2$, $2L\,\eta^2 G^2$, $L\,\eta^2 G^2$) contribute constants times $\eta^2 G^2$. These are also nonnegative and can be collected into $A\,\eta^2$.

Putting the three groups together, the one-step recursion takes the form

$$\mathbb{E}\big[\Psi_\alpha(w^+, v^+)\,\big|\,\mathcal{F}_t\big] \;\leq\; (1 - c\eta)\,\Psi_\alpha(w_t, v_t) \;+\; C_{\text{lin}}G^2\,\eta \;+\; A\,\eta^2,$$

which is exactly the recursion ($\mathbf{P}$) used in the stability analysis.

**(c) Two-run recursion.** Recall $\Xi_t := \Psi_{\alpha,\mathcal{D}}(w_t, v_t) + \Psi_{\alpha,\mathcal{D}^{(i)}}(w'_t, v'_t)$ and $D_t := \|w_t - w'_t\| + \|v_t - v'_t\|$. Apply the one-run bound from part (b) to $\mathcal{D}$ and to $\mathcal{D}^{(i)}$ (same stepsize; shared index $i_t$), then sum:

$$\mathbb{E}[\Xi_{t+1}\mid\mathcal{F}_t] \leq (1 - c\eta_t)\,\Xi_t \;+\; A\,\eta_t^2 \;+\; C_{\text{lin}}G^2\,\eta_t$$
$$+\; \underbrace{\eta_t\Big(\big\langle\nabla\Phi_{\mathcal{D}}(w_t) - \nabla\Phi_{\mathcal{D}^{(i)}}(w'_t),\, g_t^w\big\rangle \;-\; \big\langle\nabla_w F_{\mathcal{D}}(w_t, v_t) - \nabla_w F_{\mathcal{D}^{(i)}}(w'_t, v'_t),\, g_t^w\big\rangle\Big)}_{=:\,\mathsf{Leak}_t^{(w)}}$$

$$\tag{13}$$

$$+\; \eta_t\Big(\mathbf{1}\{i_t = i\} + \mathbf{1}\{\hat{i}_t = i\}\Big)\tilde{B}_0\,G^2.$$

Bounding the leak term by Cauchy–Schwarz, joint $L$-smoothness, and Lemma 10 gives

$$|\mathsf{Leak}_t^{(w)}| = \eta_t\Big|\big\langle\nabla\Phi_{\mathcal{D}}(w_t) - \nabla\Phi_{\mathcal{D}^{(i)}}(w'_t),\, g_t^w\big\rangle - \big\langle\nabla_w F_{\mathcal{D}}(w_t, v_t) - \nabla_w F_{\mathcal{D}^{(i)}}(w'_t, v'_t),\, g_t^w\big\rangle\Big|$$

$$\leq \eta_t\Big(\big|\big\langle\nabla\Phi_{\mathcal{D}}(w_t) - \nabla\Phi_{\mathcal{D}^{(i)}}(w'_t),\, g_t^w\big\rangle\big| + \big|\big\langle\nabla_w F_{\mathcal{D}}(w_t, v_t) - \nabla_w F_{\mathcal{D}^{(i)}}(w'_t, v'_t),\, g_t^w\big\rangle\big|\Big)$$

$$\leq \eta_t\Big(\big\|\nabla\Phi_{\mathcal{D}}(w_t) - \nabla\Phi_{\mathcal{D}^{(i)}}(w'_t)\big\|\|g_t^w\| + \big\|\nabla_w F_{\mathcal{D}}(w_t, v_t) - \nabla_w F_{\mathcal{D}^{(i)}}(w'_t, v'_t)\big\|\|g_t^w\|\Big)$$
$$\text{(Cauchy-Schwarz)}$$

$$\leq \eta_t\Big(L\|w_t - w'_t\|\|g_t^w\| + L_\Phi\|w_t - w'_t\|\|g_t^w\| + 2L\|v_t - v'_t\|\|g_t^w\|\Big)$$
$$\text{(Joint } L\text{-smoothness of gradients)}$$

$$\leq \eta_t G\Big((2L + 2L_\Phi)\|w_t - w'_t\| + 2L\|v_t - v'_t\| + \frac{2G}{n}\Big(2 + \frac{L}{\rho}\Big)\Big). \qquad \text{(Lemma 10)}$$

hence, with $D_t = \|w_t - w'_t\| + \|v_t - v'_t\|$,

$$\mathsf{Leak}_t := \mathsf{Leak}_t^{(w)} \;\leq\; C_{\text{leak}}\,G\,\eta_t\,D_t \;+\; C_*\,\frac{G^2}{n}\,\eta_t, \qquad C_{\text{leak}} := 2(2L + L_\Phi), \quad C_* := 4\Big(2 + \frac{L}{\rho}\Big).$$

Therefore

$$\mathbb{E}[\Xi_{t+1}\mid\mathcal{F}_t] \;\leq\; (1 - c\eta_t)\,\Xi_t + A\,\eta_t^2 + C_{\text{leak}}\,G\,\eta_t\,D_t + C_*\,\frac{G^2}{n}\,\eta_t + C_{\text{lin}}\,G^2\,\eta_t + \eta_t\Big(\mathbf{1}\{i_t = i\} + \mathbf{1}\{\hat{i}_t = i\}\Big)\tilde{B}_0 G^2,$$

$$\tag{14}$$

which matches equation 6 up to explicit constants.

*Weak distance recursion.* A single SGDA step gives

$$\|w_{t+1} - w'_{t+1}\| \leq \|w_t - w'_t\| + \eta_t \|\nabla_w f(w_t, v_t; z_{i_t}) - \nabla_w f(w'_t, v'_t; z_{i_t})\| + \eta_t \mathbf{1}\{i_t = i\} \cdot 2G,$$

and similarly for $v$ (with ascent). Joint $L$-smoothness yields

$$\|\nabla_w f(w_t, v_t; z) - \nabla_w f(w'_t, v'_t; z)\| \leq L(\|w_t - w'_t\| + \|v_t - v'_t\|), \quad \|\nabla_v f(\cdot) - \nabla_v f(\cdot)\| \leq L(\|w_t - w'_t\| + \|v_t - v'_t\|).$$

Hence

$$\mathbb{E}[D_{t+1} \mid \mathcal{F}_t] \leq (1 + \kappa\eta_t) D_t + \eta_t \mathbf{1}\{i_t = i\} \tilde{B}_1 G + \eta_t \Upsilon_t,$$

with $\kappa := 2L$ and $\tilde{B}_1 := 8$.

To bound the term $\Upsilon_t$, which represents the expected difference in stochastic gradients, we first analyze the difference of the full-batch gradients. We decompose this difference for the primal and dual variables separately.

For the primal variable, we use the triangle inequality to introduce the value function $\Phi$ and the mismatch term $\Delta_t := \nabla_w F_{\mathcal{D}}(w_t, v_t) - \nabla \Phi_{\mathcal{D}}(w_t)$:

$$\|\nabla_w F_{\mathcal{D}}(w_t, v_t) - \nabla_w F_{\mathcal{D}^{(i)}}(w'_t, v'_t)\| \leq \underbrace{\|\nabla\Phi_{\mathcal{D}}(w_t) - \nabla\Phi_{\mathcal{D}^{(i)}}(w'_t)\|}_{\leq L_\Phi \|w_t - w'_t\| + \frac{2G}{n}(1 + \frac{L}{\rho}) \text{ by Lem. 10}}$$

$$+ \underbrace{\|\Delta_t\|}_{\leq L\sqrt{2\Gamma_t/\rho} \text{ by Lem. 8}} + \underbrace{\|\Delta'_t\|}_{\leq L\sqrt{2\Gamma'_t/\rho} \text{ by Lem. 8}}$$

For the dual variable, we use joint $L$-smoothness and the single-sample replacement identity:

$$\|\nabla_v F_{\mathcal{D}}(w_t, v_t) - \nabla_v F_{\mathcal{D}^{(i)}}(w'_t, v'_t)\| \leq \|\nabla_v F_{\mathcal{D}}(w_t, v_t) - \nabla_v F_{\mathcal{D}}(w'_t, v'_t)\| + \|\nabla_v F_{\mathcal{D}}(w'_t, v'_t) - \nabla_v F_{\mathcal{D}^{(i)}}(w'_t, v'_t)\|$$

$$\leq L(\|w_t - w'_t\| + \|v_t - v'_t\|) + \frac{2G}{n} = LD_t + \frac{2G}{n}.$$

Combining the bounds on the primal and dual components gives a complete bound on the full-batch gradient difference. Summing the two inequalities and using $D_t \geq \|w_t - w'_t\|$, we have:

$$\|\nabla F_{\mathcal{D}}(w_t, v_t) - \nabla F_{\mathcal{D}^{(i)}}(w'_t, v'_t)\|_1 \leq (L_\Phi + L)D_t + L\sqrt{2/\rho}(\sqrt{\Gamma_t} + \sqrt{\Gamma'_t}) + \frac{2G}{n}\left(2 + \frac{L}{\rho}\right).$$

We then convert the geometric quantities on the right-hand side ($D_t$ and $\Gamma_t$) into the Lyapunov potential $\Xi_t$. First, using the triangle inequality along with Lemmas 11 and 12, we bound the distance $D_t$:

$$D_t = \|w_t - w'_t\| + \|v_t - v'_t\|$$

$$\leq (\|w_t - x^\star_D\| + \|v_t - v^\star_D\|) + (\|x^\star_D - x^\star_{D^{(i)}}\| + \|v^\star_D - v^\star_{D^{(i)}}\|) + (\|x^\star_{D^{(i)}} - w'_t\| + \|v^\star_{D^{(i)}} - v'_t\|)$$

$$\leq C_{\text{dist}}\sqrt{\Psi_{\alpha,\mathcal{D}}(w_t, v_t)} + \frac{S_{\text{sens}}}{n} + C_{\text{dist}}\sqrt{\Psi_{\alpha,\mathcal{D}^{(i)}}(w'_t, v'_t)}$$

$$\leq C_{\text{dist}}\left(\sqrt{\Psi_{\alpha,\mathcal{D}}} + \sqrt{\Psi_{\alpha,\mathcal{D}^{(i)}}}\right) + \frac{S_{\text{sens}}}{n} \leq \sqrt{2}C_{\text{dist}}\sqrt{\Xi_t} + \frac{S_{\text{sens}}}{n}.$$

Second, from the definition of the potential, we have $\sqrt{\Gamma_t} + \sqrt{\Gamma'_t} \leq 2\sqrt{\Xi_t/\alpha}$. Substituting these into our main inequality gives:

$$\|\nabla F_{\mathcal{D}}(w_t, \cdot) - \nabla F_{\mathcal{D}^{(i)}}(w'_t, \cdot)\|_1 \leq (L_\Phi + L)\left(\sqrt{2}C_{\text{dist}}\sqrt{\Xi_t} + \frac{S_{\text{sens}}}{n}\right) + L\sqrt{2/\rho}\left(2\sqrt{\Xi_t/\alpha}\right) + O(G/n).$$

The terms involving $\sqrt{\Xi_t}$ can be collected and absorbed into a single term $a\sqrt{\Xi_t}$, where $a$ is a constant that depends on the problem parameters (e.g., $a := 2C_{\text{dist}}(L + L_\Phi)$ serves as a valid, convenient upper bound). The remaining terms are of order $O(1/n)$. Therefore, we state the final bound on $\Upsilon_t$ as:

$$\Upsilon_t \leq 2C_{\text{dist}}(L + L_\Phi)\sqrt{\Xi_t} + \frac{2G}{n}\left(1 + \frac{L}{\rho}\right),$$

and therefore

$$\mathbb{E}[D_{t+1} \mid \mathcal{F}_t] \leq (1 + \kappa\eta_t) D_t + a\,\eta_t\sqrt{\Xi_t} + \eta_t \mathbf{1}\{i_t = i\} \tilde{B}_1 G + \eta_t \frac{2G}{n}\left(1 + \frac{L}{\rho}\right), \qquad a := 2C_{\text{dist}}(L + L_\Phi).$$

$$(15)$$

$$\square$$

**Lemma 14** (One-step ascent for $L$-smooth, $\rho$-strongly concave $g$). *Let $g : \mathbb{R}^d \to \mathbb{R}$ be $L$-smooth and $\rho$-strongly concave, and let $v^+ = v + \eta \nabla g(v)$ with $\eta \geq 0$. Denote $\theta(v) := g(v^\star) - g(v)$, where $v^\star \in \arg\max g$. Then*

$$\theta(v^+) \;\leq\; \left(1 - 2\rho\eta + \rho L\eta^2\right) \theta(v).$$

*Consequently,*

$$g(v^\star) - g(v^+) \;\leq\; \left(1 - 2\rho\eta + \rho L\eta^2\right) \left[g(v^\star) - g(v)\right].$$

*Proof.* For any $L$-smooth differentiable function $h$ one has

$$h(y) \;\geq\; h(x) + \langle \nabla h(x), y - x \rangle - \tfrac{L}{2}\|y - x\|^2.$$

Apply this to $h = g$, $x = v$, $y = v^+ = v + \eta \nabla g(v)$:

$$g(v^+) \;\geq\; g(v) + \eta\|\nabla g(v)\|^2 - \tfrac{L}{2}\eta^2\|\nabla g(v)\|^2.$$

Rearranging gives

$$g(v^\star) - g(v^+) \;\leq\; g(v^\star) - g(v) \;-\; \left(\eta - \tfrac{L}{2}\eta^2\right)\|\nabla g(v)\|^2. \tag{16}$$

Since $g$ is $\rho$-strongly concave, $f := -g$ is $\rho$-strongly convex; hence $f$ satisfies the Polyak–Łojasiewicz (PL) inequality

$$\tfrac{1}{2}\|\nabla f(x)\|^2 \;\geq\; \rho\big(f(x) - f^\star\big),$$

with the *same* constant $\rho$ (Karimi et al., 2016). Translating back to $g$ gives

$$\tfrac{1}{2}\|\nabla g(v)\|^2 \;\geq\; \rho\big(g(v^\star) - g(v)\big) \;=\; \rho\,\theta(v). \tag{17}$$

*Step 3 (Combine).* Insert equation 17 into equation 16:

$$\theta(v^+) \;\leq\; \theta(v) \;-\; 2\rho\eta\left(1 - \tfrac{L}{2}\eta\right)\theta(v) \;=\; \left(1 - 2\rho\eta + \rho L\eta^2\right)\theta(v),$$

$\square$

**Lemma 15** (Damping the weak distance recursion). *Let $S_t := \sum_{s=0}^{t-1}\eta_s$ and define the damped distance $\widetilde{D}_t := e^{-2LS_t}D_t$ from equation 7. Set $\lambda := 2L$ for this lemma. Then, taking expectations and averaging over $i$,*

$$\bar{\widetilde{D}}_{t+1} \;\leq\; \bar{\widetilde{D}}_t \;+\; a\,\eta_t\, e^{-2LS_{t+1}}\mathbb{E}\big[\sqrt{\bar{\Xi}_t}\big] \;+\; \frac{1}{n}\left(2\tilde{B}_1 G + 2G\left(1 + \frac{L}{\rho}\right)\right)\eta_t\, e^{-\lambda S_{t+1}}.$$

*Consequently, summing from $t = 0$ to $T - 1$,*

$$\bar{D}_T \;\leq\; a\sum_{t=0}^{T-1}\eta_t\, e^{-2L\sum_{s=t+1}^{T-1}\eta_s}\,\mathbb{E}\big[\sqrt{\bar{\Xi}_t}\big] \;+\; \frac{1}{\lambda n}\left(2\tilde{B}_1 G + 2G\left(1 + \frac{L}{\rho}\right)\right). \tag{18}$$

*Proof.* Start from the weak distance recursion equation 7:

$$\mathbb{E}[D_{t+1} \mid \mathcal{F}_t] \;\leq\; (1 + 2L\eta_t)D_t + a\,\eta_t\,\sqrt{\Xi_t} + \eta_t\,\mathbf{1}\{i_t = i\}\,\tilde{B}_1\,G + \eta_t\,\frac{2G}{n}\left(1 + \frac{L}{\rho}\right).$$

Multiply both sides by $e^{-2LS_{t+1}} = e^{-2L(S_t + \eta_t)}$ to get

$$e^{-2LS_{t+1}}\mathbb{E}[D_{t+1} \mid \mathcal{F}_t] \leq e^{-2LS_t}\left(1 + 2L\eta_t\right)e^{-2L\eta_t}D_t \;+\; a\,\eta_t\, e^{-2LS_{t+1}}\sqrt{\Xi_t}$$

$$+\; \eta_t\, e^{-2LS_{t+1}}\,\mathbf{1}\{i_t = i\}\,\tilde{B}_1 G \;+\; \eta_t\, e^{-2LS_{t+1}}\frac{2G}{n}\left(1 + \frac{L}{\rho}\right).$$

Since $(1 + x)e^{-x} \leq 1$ for all $x \geq 0$, the first term is $\leq e^{-2LS_t}D_t = \widetilde{D}_t$. Taking total expectation and then averaging over the replacement index $i$, we make explicit that the hit $\mathbf{1}\{i_t = i\}$ contributes to *both* coordinates in $D_t = \|w_t - w_t'\| + \|v_t - v_t'\|$. For a fixed $i$,

$$\mathbb{E}[D_{t+1} \mid \mathcal{F}_t] \leq (1 + 2L\eta_t)D_t \;+\; a\,\eta_t\,\sqrt{\Xi_t} \;+\; \eta_t\,\mathbf{1}\{i_t = i\}\,c_w G \;+\; \eta_t\,\mathbf{1}\{i_t = i\}\,c_v G + \eta_t\,\frac{2G}{n}\left(1 + \frac{L}{\rho}\right)$$

$$\leq (1 + 2L\eta_t)D_t \; + \; a\,\eta_t\,\sqrt{\Xi_t} \; + \; \eta_t\,\mathbf{1}\{i_t = i\}\,(2\tilde{B}_1)G + \eta_t\,\frac{2G}{n}\Big(1 + \frac{L}{\rho}\Big),$$

where we bundle constants so that $c_w = c_v = \tilde{B}_1$. Multiplying both sides by $e^{-2LS_{t+1}}$ with $S_t := \sum_{s=0}^{t-1}\eta_s$ and using $(1 + 2L\eta_t)e^{-2L\eta_t} \leq 1$ yields

$$\mathbb{E}\Big[\widetilde{D}_{t+1} \mid \mathcal{F}_t\Big] \; \leq \; \widetilde{D}_t + a\,\eta_t\,e^{-2LS_{t+1}}\,\sqrt{\Xi_t} + \eta_t\,e^{-2LS_{t+1}}\,\mathbf{1}\{i_t = i\}\,(2\tilde{B}_1)G + \eta_t e^{-2LS_{t+1}}\frac{2G}{n}\Big(1 + \frac{L}{\rho}\Big),$$

where $\widetilde{D}_t := e^{-2LS_t}D_t$. Taking total expectation and then averaging over $i$ (so that $\mathbb{E}[\mathbf{1}\{i_t = i\}] = 1/n$) gives

$$\bar{\widetilde{D}}_{t+1} \; \leq \; \bar{\widetilde{D}}_t \; + \; a\,\eta_t\,e^{-2LS_{t+1}}\,\mathbb{E}\big[\sqrt{\bar{\Xi}_t}\big] \; + \; \frac{1}{n}\Big(2\tilde{B}_1 G + 2G\Big(1 + \frac{L}{\rho}\Big)\Big)\eta_t\,e^{-2LS_{t+1}}.$$

Summing the inequality from $t = 0$ to $T-1$ and noting that both runs start from the same initialization $(w_0, v_0) = (w_0', v_0')$, we have $D_0 = 0$ and hence $\widetilde{D}_0 = 0$ and $\bar{\widetilde{D}}_0 = 0$. Therefore,

$$\bar{\widetilde{D}}_T - \bar{\widetilde{D}}_0 \; = \; \bar{\widetilde{D}}_T \; \leq \; a\sum_{t=0}^{T-1}\eta_t e^{-2LS_{t+1}}\,\mathbb{E}\big[\sqrt{\bar{\Xi}_t}\big] + \frac{1}{n}\Big(2\tilde{B}_1 G + 2G\Big(1 + \frac{L}{\rho}\Big)\Big)\sum_{t=0}^{T-1}\eta_t e^{-2LS_{t+1}}.$$

Applying Lemma 16 with $\gamma = \lambda = 2L$ to the last sum and noting that $e^{2LS_T}e^{-2LS_{t+1}} = e^{2L\sum_{s=t+1}^{T-1}\eta_s}$ yields

$$\bar{D}_T \; = \; e^{2LS_T}\bar{\widetilde{D}}_T \; \leq \; a\sum_{t=0}^{T-1}\eta_t e^{-2L\sum_{s=t+1}^{T-1}\eta_s}\,\mathbb{E}\big[\sqrt{\bar{\Xi}_t}\big] \; + \; \frac{1}{\lambda n}\Big(2\tilde{B}_1 G + 2G\Big(1 + \frac{L}{\rho}\Big)\Big),$$

which is equation 18. $\qquad\square$

**Lemma 16.** *Let $(\eta_t)$ satisfy the Robbins–Monro conditions. For $S_t := \sum_{s=0}^{t-1}\eta_s$ and any $\gamma > 0$, any $T \geq 1$, set $\eta_{\max,T} := \max_{0 \leq t < T}\eta_t$. Then*

$$\sum_{t=0}^{T-1}\eta_t\,e^{-\gamma\sum_{s=t+1}^{T-1}\eta_s} \; \leq \; \frac{e^{\gamma\eta_{\max,T}}}{\gamma}\Big(1 - e^{-\gamma S_T}\Big) \; \leq \; \frac{e^{\gamma\eta_{\max,T}}}{\gamma},$$

*and*

$$\sum_{t=0}^{T-1}\eta_t^2\,e^{-\gamma\sum_{s=t+1}^{T-1}\eta_s} \xrightarrow[T\to\infty]{} 0.$$

*In particular, under Assumption (A6) and with $\gamma = 2L$, one has $\eta_{\max,T} \leq 1/(4L)$ and therefore*

$$\sum_{t=0}^{T-1}\eta_t\,e^{-2L\sum_{s=t+1}^{T-1}\eta_s} \; \leq \; \frac{e^{1/2}}{2L}\,.$$

*Proof.* Let $S_t := \sum_{s=0}^{t-1}\eta_s$ and note $S_t \uparrow \infty$ under Robbins–Monro. For the first display, fix $t$ and any $u \in [S_t, S_{t+1}]$. Then

$$e^{-\gamma(S_T - S_{t+1})} = e^{-\gamma(S_T - u)}\,e^{\gamma(S_{t+1} - u)} \; \leq \; e^{\gamma\eta_{\max,T}}\,e^{-\gamma(S_T - u)},$$

since $0 \leq S_{t+1} - u \leq S_{t+1} - S_t = \eta_t \leq \eta_{\max,T}$. Hence

$$\eta_t\,e^{-\gamma(S_T - S_{t+1})} \; \leq \; e^{\gamma\eta_{\max,T}}\int_{S_t}^{S_{t+1}} e^{-\gamma(S_T - u)}\,\mathrm{d}u.$$

Summing over $t = 0, \ldots, T-1$ and using $\sum_t \int_{S_t}^{S_{t+1}}(\cdot) = \int_0^{S_T}(\cdot)$ gives

$$\sum_{t=0}^{T-1}\eta_t\,e^{-\gamma\sum_{s=t+1}^{T-1}\eta_s} \; \leq \; e^{\gamma\eta_{\max,T}}\int_0^{S_T} e^{-\gamma(S_T - u)}\,\mathrm{d}u = \frac{e^{\gamma\eta_{\max,T}}}{\gamma}\Big(1 - e^{-\gamma S_T}\Big) \; \leq \; \frac{e^{\gamma\eta_{\max,T}}}{\gamma}.$$

For the second display, fix $\varepsilon > 0$. Since $\sum_t \eta_t^2 < \infty$, pick $T_0$ with $\sum_{t \geq T_0} \eta_t^2 < \varepsilon$. Then split

$$\sum_{t=0}^{T-1} \eta_t^2 \, e^{-\gamma \sum_{s=t+1}^{T-1} \eta_s} = \sum_{t=0}^{T_0-1} \eta_t^2 \, e^{-\gamma(S_T - S_{t+1})} + \sum_{t=T_0}^{T-1} \eta_t^2 \, e^{-\gamma(S_T - S_{t+1})} \ \leq \ e^{-\gamma(S_T - S_{T_0})} \sum_{t=0}^{T_0-1} \eta_t^2 + \varepsilon.$$

Let $T \to \infty$ to send the first term to 0 (since $S_T \to \infty$), and then let $\varepsilon \downarrow 0$. Finally, under Assumption (A6) and $\gamma = 2L$, we have $\eta_{\max,T} \leq 1/(4L)$, hence $e^{\gamma \eta_{\max,T}} \leq e^{1/2}$, which yields the stated corollary. $\qquad \square$

**Lemma 17** (Closing the potential recursion). *After averaging equation 6 over $i$,*

$$\bar{\Xi}_{t+1} \ \leq \ \left(1 - \tfrac{c}{2}\eta_t\right)\bar{\Xi}_t \ + \ A\,\eta_t^2 \ + \ \frac{b^2}{2c}\,\eta_t \ + \ \frac{\tilde{H}}{n}\,\eta_t, \tag{19}$$

*with $b := 2C_{\text{leak}}C_{\text{dist}}$ and $\tilde{H} := 2\tilde{B}_0 G^2 + C_{\text{leak}}S_{\text{sens}} + C_* G$, where*

$$S_{\text{sens}} \ := \ 2G\left(\frac{(1 + L/\rho)^2}{\sqrt{\mu_{\text{PL}}\mu_{\text{QG}}}} + \frac{1}{\rho}\right)$$

*is the constant from Lemma 12.*

*Proof.* Starting from equation 6,

$$\mathbb{E}[\Xi_{t+1} \mid \mathcal{F}_t] \ \leq \ (1 - c\eta_t)\,\Xi_t + A\,\eta_t^2 + C_{\text{leak}}\,\eta_t\,D_t + C_*\,\frac{G}{n}\,\eta_t + \eta_t\big(\mathbf{1}\{i_t = i\} + \mathbf{1}\{\hat{i}_t = i\}\big)\,\tilde{B}_0\,G^2.$$

By Lemma 11, for each run $\|w_t - x_D^\star\| + \|v_t - v_D^\star\| \leq C_{\text{dist}}\sqrt{\Psi_{\alpha,\mathcal{D}}(w_t, v_t)}$ and analogously for the primed run. Using the triangle inequality and Lemma 12 for the cross-dataset saddle shift, we have

$$D_t = \|w_t - w_t'\| + \|v_t - v_t'\|$$

$$\leq \|w_t - x_D^*\| + \|x_D^* - x_{D^{(i)}}^*\| + \|x_{D^{(i)}}^* - w_t'\| + \|v_t - v_D^*\| + \|v_D^* - v_{D^{(i)}}^*\| - \|v_{D^{(i)}}^* - v_t'\|$$

(Triangle inequality)

$$\leq C_{\text{dist}}\left(\sqrt{\Psi_{\alpha,\mathcal{D}}(w_t, v_t)} + \sqrt{\Psi_{\alpha,\mathcal{D}^{(i)}}(w_t', v_t')}\right) + \|x_D^* - x_{D^{(i)}}^*\| - \|v_{D^{(i)}}^* - v_t'\|$$

(Lemma 12)

$$\leq C_{\text{dist}}\left(\sqrt{\Psi_{\alpha,\mathcal{D}}(w_t, v_t)} + \sqrt{\Psi_{\alpha,\mathcal{D}^{(i)}}(w_t', v_t')}\right) + \frac{S_{\text{sens}}}{n}$$

(Lemma 11)

Since $\sqrt{\Psi_{\alpha,\mathcal{D}}} \leq \sqrt{\Xi_t}$ and $\sqrt{\Psi_{\alpha,\mathcal{D}^{(i)}}} \leq \sqrt{\Xi_t}$,

$$D_t \ \leq \ 2C_{\text{dist}}\sqrt{\Xi_t} \ + \ \frac{S_{\text{sens}}}{n}.$$

Plug this into equation 6:

$$\mathbb{E}[\Xi_{t+1} \mid \mathcal{F}_t] \ \leq \ (1 - c\eta_t)\,\Xi_t + A\,\eta_t^2 + \underbrace{2C_{\text{leak}}C_{\text{dist}}}_{=:b}\,\eta_t\,\sqrt{\Xi_t} + \frac{C_{\text{leak}}S_{\text{sens}}}{n}\,\eta_t + C_*\,\frac{G}{n}\,\eta_t + \eta_t\big(\mathbf{1}\{i_t = i\} + \mathbf{1}\{\hat{i}_t = i\}\big)\,\tilde{B}_0\,G^2.$$

Apply the inequality $uv \leq \frac{\gamma}{2}u^2 + \frac{1}{2\gamma}v^2$ with $u = \sqrt{\eta_t \Xi_t}$, $v = b\sqrt{\eta_t}$, and $\gamma = c$ to the mixed term:

$$b\,\eta_t\,\sqrt{\Xi_t} \ \leq \ \frac{c}{2}\,\eta_t\,\Xi_t \ + \ \frac{b^2}{2c}\,\eta_t.$$

Therefore,

$$\mathbb{E}[\Xi_{t+1} \mid \mathcal{F}_t] \ \leq \ \left(1 - \tfrac{c}{2}\eta_t\right)\Xi_t + A\,\eta_t^2 + \frac{b^2}{2c}\,\eta_t + \frac{C_{\text{leak}}S_{\text{sens}}}{n}\,\eta_t + C_*\,\frac{G}{n}\,\eta_t + \eta_t\big(\mathbf{1}\{i_t = i\} + \mathbf{1}\{\hat{i}_t = i\}\big)\,\tilde{B}_0\,G^2.$$

Taking total expectation and averaging over $i$ (both indicators have mean $1/n$) gives

$$\bar{\Xi}_{t+1} \ \leq \ \left(1 - \tfrac{c}{2}\eta_t\right)\bar{\Xi}_t \ + \ A\,\eta_t^2 \ + \ \frac{b^2}{2c}\,\eta_t \ + \ \frac{1}{n}\Big(2\tilde{B}_0 G^2 + C_{\text{leak}}S_{\text{sens}} + C_* G\Big)\eta_t.$$

With $\tilde{H} := 2\tilde{B}_0 G^2 + C_{\text{leak}}S_{\text{sens}} + C_* G$ and $b = 2C_{\text{leak}}C_{\text{dist}}$ this is equation 19. $\qquad \square$

**Lemma 18** (Bounded weighted sum of potentials). *For any $\theta \in (\frac{1}{2}, 1)$ and $S_t := \sum_{s=0}^{t-1} \eta_s$,*

$$\sum_{t=0}^{T-1} e^{-\theta c \sum_{s=t+1}^{T-1} \eta_s} \bar{\Xi}_t \leq \frac{2}{1 - e^{-(\theta - \frac{1}{2})c\,\eta_{\min,T}}} e^{-\frac{c}{2} \sum_{s=0}^{T-1} \eta_s} \bar{\Xi}_0$$

$$+ \frac{2}{1 - e^{-(\theta - \frac{1}{2})c\,\eta_{\min,T}}} A \sum_{t=0}^{T-1} \eta_t^2\, e^{-\frac{c}{2} \sum_{s=t+1}^{T-1} \eta_s} + \frac{8}{c\big(1 - e^{-(\theta - \frac{1}{2})c\,\eta_{\min,T}}\big)} \left(\frac{b^2}{2c} + \frac{\tilde{H}}{n}\right).$$

*In particular, since $c\,\eta_{\min,T} \leq \frac{1}{2}$ (by Assumption (A6)) and hence $1 - e^{-x} \geq x/2$ for $x \in [0, \frac{1}{2}]$, we have*

$$\sum_{t=0}^{T-1} e^{-\theta c \sum_{s=t+1}^{T-1} \eta_s} \bar{\Xi}_t \leq \frac{4}{(\theta - \frac{1}{2})c\,\eta_{\min,T}} e^{-\frac{c}{2} \sum_{s=0}^{T-1} \eta_s} \bar{\Xi}_0$$

$$+ \frac{4}{(\theta - \frac{1}{2})c\,\eta_{\min,T}} A \sum_{t=0}^{T-1} \eta_t^2\, e^{-\frac{c}{2} \sum_{s=t+1}^{T-1} \eta_s} + \frac{16}{(\theta - \frac{1}{2})c^2\,\eta_{\min,T}} \left(\frac{b^2}{2c} + \frac{\tilde{H}}{n}\right).$$

*Proof.* Start from equation 19 and unroll using $1 - \frac{c}{2}\eta_k \leq e^{-\frac{c}{2}\eta_k}$:

$$\bar{\Xi}_t \leq e^{-\frac{c}{2}S_t} \bar{\Xi}_0 + \sum_{k=0}^{t-1} e^{-\frac{c}{2}(S_t - S_{k+1})}\big(A\eta_k^2 + q\,\eta_k\big), \qquad q := \frac{b^2}{2c} + \frac{\tilde{H}}{n}.$$

Multiply by the weight $w_t := e^{-\theta c(S_T - S_{t+1})}$ and sum over $t = 0, \ldots, T-1$:

$$\sum_{t=0}^{T-1} w_t \bar{\Xi}_t \leq \underbrace{\sum_{t=0}^{T-1} w_t e^{-\frac{c}{2}S_t} \bar{\Xi}_0}_{\text{I}} + A \underbrace{\sum_{t=0}^{T-1}\sum_{k=0}^{t-1} w_t e^{-\frac{c}{2}(S_t - S_{k+1})}\eta_k^2}_{\text{II}} + q \underbrace{\sum_{t=0}^{T-1}\sum_{k=0}^{t-1} w_t e^{-\frac{c}{2}(S_t - S_{k+1})}\eta_k}_{\text{III}}.$$

Throughout, Assumption (A6) implies $c\,\eta_t \leq \frac{1}{2}$ (since $c \leq \rho/2$ and $\eta_t \leq 1/\rho$), hence $1 - e^{-x} \geq x/2$ for $x \in [0, \frac{1}{2}]$.

*Claim A.* For $\theta \in (\frac{1}{2}, 1)$,

$$\text{I} = \sum_{t=0}^{T-1} e^{-\theta c(S_T - S_{t+1})}\, e^{-\frac{c}{2}S_t} \leq \frac{2\, e^{-\frac{c}{2}S_T}}{1 - \exp\big(-(\theta - \frac{1}{2})c\,\eta_{\min,T}\big)} \leq \frac{4}{(\theta - \frac{1}{2})c\,\eta_{\min,T}} e^{-\frac{c}{2}S_T},$$

where $\eta_{\min,T} := \min_{0 \leq t < T} \eta_t$.

*Proof of Claim A.* Using $S_t = S_{t+1} - \eta_t$ and $S_T = S_{t+1} + (S_T - S_{t+1})$,

$$e^{-\theta c(S_T - S_{t+1})}\, e^{-\frac{c}{2}S_t} = e^{-\frac{c}{2}S_T}\, e^{-(\theta - \frac{1}{2})c(S_T - S_{t+1})}\, e^{\frac{c}{2}\eta_t}.$$

Since $c\,\eta_t \leq \frac{1}{2}$, $e^{\frac{c}{2}\eta_t} \leq e^{1/4} \leq 2$. Therefore,

$$\text{I} \leq 2\, e^{-\frac{c}{2}S_T} \sum_{t=0}^{T-1} e^{-(\theta - \frac{1}{2})c(S_T - S_{t+1})}.$$

Because $S_T - S_{t+1} \geq (T - 1 - t)\,\eta_{\min,T}$, the sum is bounded by a geometric series, giving the first display; the second follows from $1 - e^{-x} \geq x/2$ for $x \in [0, 1]$. $\triangle$

*Claim B.* Let $\alpha := (\theta - \frac{1}{2})c > 0$. For any fixed $k \in \{0, \ldots, T-2\}$,

$$\sum_{t=k+1}^{T-1} w_t\, e^{-\frac{c}{2}(S_t - S_{k+1})} \leq \frac{2}{1 - e^{-\alpha\,\eta_{\min,T}}} e^{-\frac{c}{2}(S_T - S_{k+1})} \leq \frac{4}{(\theta - \frac{1}{2})c\,\eta_{\min,T}} e^{-\frac{c}{2}(S_T - S_{k+1})}.$$

*Proof of Claim B.* As above,

$$w_t\, e^{-\frac{c}{2}(S_t - S_{k+1})} = e^{-\frac{c}{2}(S_T - S_{k+1})}\, e^{-\alpha(S_T - S_{t+1})}\, e^{\frac{c}{2}\eta_t} \leq 2\, e^{-\frac{c}{2}(S_T - S_{k+1})}\, e^{-\alpha(S_T - S_{t+1})}.$$

Let $I_t := \int_{S_{t+1}}^{S_{t+2}} e^{-\alpha(S_T-u)}\,\mathrm{d}u$. A direct computation gives

$$I_t = \frac{1}{\alpha} e^{-\alpha(S_T-S_{t+1})}\bigl(1-e^{-\alpha\eta_{t+1}}\bigr) \quad\Rightarrow\quad e^{-\alpha(S_T-S_{t+1})} = \frac{\alpha}{1-e^{-\alpha\eta_{t+1}}} I_t \leq \frac{\alpha}{1-e^{-\alpha\eta_{\min,T}}} I_t,$$

since $\eta_{t+1} \geq \eta_{\min,T}$. Summing over $t = k+1,\ldots,T-1$ and using $\sum_{t=k+1}^{T-1} I_t = \int_{S_{k+1}}^{S_T} e^{-\alpha(S_T-u)}\,\mathrm{d}u \leq 1/\alpha$ yields the claim; the explicit bound uses $1-e^{-x} \geq x/2$ with $x = \alpha\eta_{\min,T} \leq \frac{1}{2}$. $\triangle$

With Claim A and $1-e^{-x} > x/2$,

$$\mathsf{I} \leq \frac{2}{1-e^{-(\theta-\frac{1}{2})c\,\eta_{\min,T}}} e^{-\frac{c}{2}S_T},$$

and, applying Claim B for each fixed $k$ inside the double sum and then summing in $t$,

$$\begin{aligned}
\mathsf{II} &= \sum_{t=0}^{T-1}\sum_{k=0}^{t-1} w_t e^{-\frac{c}{2}(S_t-S_{k+1})}\eta_k^2 \\
&= \sum_{k=0}^{T-2}\sum_{t=k+1}^{T-1} w_t e^{-\frac{c}{2}(S_t-S_{k+1})}\eta_k^2 && \text{(Fubini)} \\
&\leq \sum_{k=0}^{T-2} \frac{2}{1-e^{-\alpha\,\eta_{\min,T}}} e^{-\frac{c}{2}(S_T-S_{k+1})}\eta_k^2 && \text{(Claim B)}
\end{aligned}$$

For $\mathsf{III}$, combine Claim B with the kernel bound from Lemma 16

$$\sum_{k=0}^{T-2} \eta_k\, e^{-\frac{c}{2}(S_T-S_{k+1})} \leq \frac{2\,e^{\frac{c}{2}\eta_{\max,T}}}{c} \leq \frac{4}{c} \qquad \text{(since } c\,\eta_{\max,T} \leq \frac{1}{2}\text{)},$$

to obtain

$$\mathsf{III} \leq \frac{8}{c\bigl(1-e^{-(\theta-\frac{1}{2})c\,\eta_{\min,T}}\bigr)}\, q.$$

Putting the three pieces together and reindexing the middle sum gives the first displayed bound in the lemma. Putting the three pieces together and reindexing the middle sum gives the first displayed bound in the lemma. The "in particular" version follows by $1-e^{-x} \geq x/2$ with $x = (\theta-\frac{1}{2})c\,\eta_{\min,T} \leq \frac{1}{2}$. $\qquad\square$

### B.2 Proof of Main Result

*Proof of Theorem 3.*
From Lemma 17, for $c := \min\{\mu_{\mathrm{PL}}/2, \rho/2\}$ and every $t$,

$$\bar{\Xi}_{t+1} \leq \Bigl(1-\tfrac{c}{2}\eta_t\Bigr)\bar{\Xi}_t + A\eta_t^2 + \frac{b^2}{2c}\eta_t + \frac{\tilde{H}}{n}\eta_t, \tag{20}$$

with the constants $A, b, \tilde{H}$ defined in Lemma 17. Unrolling equation 20 and using $1-\frac{c}{2}\eta_k \leq e^{-\frac{c}{2}\eta_k}$ yields

$$\bar{\Xi}_T \leq e^{-\frac{c}{2}\sum_{s=0}^{T-1}\eta_s}\bar{\Xi}_0 + A\sum_{t=0}^{T-1}\eta_t^2 e^{-\frac{c}{2}\sum_{s=t+1}^{T-1}\eta_s} + e^{\frac{c}{2}\eta_{\max,T}}\left(\frac{b^2}{c^2} + \frac{2\tilde{H}}{cn}\right), \tag{$\star$}$$

where $\eta_{\max,T} := \max_{0\leq t<T}\eta_t$.

Lemma 15 (obtained from Lemma 13 ($\mathrm{D}_{\mathrm{weak}}$)) gives

$$\bar{D}_T - \frac{C_{\mathrm{hit}}}{n} \leq a\cdot\underbrace{\Bigl(\sum_{t=0}^{T-1}\eta_t^2 e^{-2(2L-\frac{c}{2})\sum_{s=t+1}^{T-1}\eta_s}\Bigr)^{1/2}}_{\sqrt{S_u}}\cdot\underbrace{\Bigl(\sum_{t=0}^{T-1}e^{-c\sum_{s=t+1}^{T-1}\eta_s}\mathbb{E}[\bar{\Xi}_t]\Bigr)^{1/2}}_{\sqrt{S_v}} \tag{21}$$

where $C_{\text{hit}} := \frac{1}{\lambda}\left(2\tilde{B}_1 G + 2G\left(1 + \frac{L}{\rho}\right)\right)$. The potential sum, $S_v$, is bounded by Lemma 18. To combine the terms into the final compact form, we bound the products that arise after substitution. Since $S_u$ is a convergent sum, it is bounded by a constant,

$$C_S := \sum_{t=0}^{\infty} \eta_t^2,$$

which depends only on the stepsize schedule $\{\eta_t\}$. Under the harmonic rule $\eta_t = \frac{c_1}{c_2+t}$, $C_S = c_1^2 \sum_{t=0}^{\infty}(c_2+t)^{-2} \leq c_1^2 \frac{\pi^2}{6}$, and for $c_2 > 1$ one may also use $C_S \leq \frac{c_1^2}{c_2-1}$. Moreover $S_u \to 0$ as $T \to \infty$, while $S_v$ need not vanish; in the product we will use only the decaying initialization term and the variance component

$$S_v^{\text{var}}(T) := \sum_{t=0}^{T-1} \eta_t^2 \, e^{-\frac{c}{2}\sum_{s=t+1}^{T-1}\eta_s}.$$

This allows for two key bounds:

(i) The product involving the term $e^{-\frac{c}{2}\sum_{s=0}^{T-1}\eta_s}\bar{\Xi}_0$ is bounded by absorbing $S_u$ into the constant: $S_u \cdot e^{-\frac{c}{2}\sum \eta_s}\Psi_{\alpha,0}^{\max} \leq C_S \cdot e^{-\frac{c}{2}\sum \eta_s}\Psi_{\alpha,0}^{\max}$.

(ii) The product of variance sums, $S_u \cdot S_v^{\text{var}}(T)$, is bounded by a multiple of their sum: $S_u \cdot S_v^{\text{var}}(T) \leq C_S \cdot S_v^{\text{var}}(T) \leq C_S \cdot \left(S_u + S_v^{\text{var}}(T)\right)$.

Substituting these bounds, grouping all constants ($a^2, C_S$, etc.) into $C_{\text{var}}$, and relaxing the exponent[5] in $S_u$ by defining $\kappa = \min\{\frac{3c}{4}, 2L - \frac{c}{2}\}$ yields the compact bound:

$$\bar{D}_T \leq$$

$$2\,C_{\text{dist}}\sqrt{e^{-\frac{c}{2}\sum_{s=0}^{T-1}\eta_s}\Psi_{\alpha,0}^{\max} + C_{\text{var}}\left(L(1+L/\rho) + \alpha\frac{L^2}{\rho}\right)G^2\left[\sum_{t=0}^{T-1}\eta_t^2\, e^{-2\kappa\sum_{s=t+1}^{T-1}\eta_s} + \sum_{t=0}^{T-1}\eta_t^2\, e^{-\frac{c}{2}\sum_{s=t+1}^{T-1}\eta_s}\right]}$$

$$\tag{22}$$

$$+ \frac{2G}{n}\left(\frac{(1+L/\rho)^2}{\sqrt{\mu_{\text{PL}}\mu_{\text{QG}}}} + \frac{1}{\rho}\right) + \frac{C_{\text{hit}}}{n},$$

with $\kappa = \min\{\frac{3c}{4}, 2L - \frac{c}{2}\}$. $\qquad\square$

*Proof of Corollary 4.* Let $\eta_t = \frac{c_1}{c_2+t}$ with $c_1 > 0$, $c_2 \geq 1$, and $c_1 < \min\{\frac{1}{2\kappa}, \frac{2}{c}\}$ so that the geometric constants below are finite. Then:

$$\sum_{s=0}^{T-1}\eta_s = c_1\sum_{s=0}^{T-1}\frac{1}{c_2+s} \geq c_1\log\left(\frac{c_2+T}{c_2}\right),$$

$$e^{-\frac{c}{2}\sum_{s=0}^{T-1}\eta_s} \leq \left(\frac{c_2}{c_2+T}\right)^{\frac{c\,c_1}{2}}, \qquad e^{-\kappa\sum_{s=0}^{T-1}\eta_s} \leq \left(\frac{c_2}{c_2+T}\right)^{\kappa\,c_1},$$

$$\sum_{t=0}^{T-1}\eta_t^2\, e^{-2\kappa\sum_{s=t+1}^{T-1}\eta_s} \leq \frac{c_1^2}{(1-2\kappa c_1)(c_2+T)}, \qquad \sum_{t=0}^{T-1}\eta_t^2\, e^{-\frac{c}{2}\sum_{s=t+1}^{T-1}\eta_s} \leq \frac{c_1^2}{(1-\frac{c}{2}c_1)(c_2+T)}.$$

Substituting these into the general bound above yields the explicit finite–$T$ rate:

$$\varepsilon = \bar{D}_T \leq 2\,C_{\text{dist}}\left[\left(\frac{c_2}{c_2+T}\right)^{\frac{c_1 c}{2}}\Psi_{\alpha,0}^{\max} + \frac{C_{\text{var}}\,c_1^2\left(L(1+L/\rho) + \alpha L^2/\rho\right)G^2}{(c_2+T)}\left(\frac{1}{1-2\kappa c_1} + \frac{1}{1-\frac{c}{2}c_1}\right)\right]^{1/2}$$

---

[5]Since $x \mapsto e^{-\gamma x}$ is decreasing in $\gamma > 0$, replacing the larger decay parameter by the smaller $\kappa := \min\{2L - \theta, \frac{3}{4}c\}$ only increases the weighted sums, hence preserves a valid upper bound; we refer to this monotone weakening as "relaxing the exponent."

$$+ \frac{2G}{n}\left(\frac{(1+L/\rho)^2}{\sqrt{\mu_{\mathrm{PL}}\mu_{\mathrm{QG}}}} + \frac{1}{\rho}\right) + \frac{C_{\mathrm{hit}}}{n}. \tag{23}$$

Thus, for fixed $n$ and feasible $(c_1, c_2)$, the optimization/stochastic term in equation 23 decays at rate $O\big((c_2 + T)^{-1/2}\big)$, while the initialization bias decays polynomially as $\left(\frac{c_2}{c_2+T}\right)^{\frac{c_1 c}{4}}$ inside the square root. □

*Proof of Theorem 7.* Write $\widehat{w} := \widehat{w}^{(T)}$, $w_{\mathrm{ERM}} := w^{\mathrm{ERM}}$ and $w^*$ for the population minimizer. Add and subtract the empirical risks of these three parameters:

$$\mathcal{R}(\widehat{w}) - \mathcal{R}(w^*) = \underbrace{\big[\mathcal{R}(\widehat{w}) - \widehat{\mathcal{R}}_n(\widehat{w})\big]}_{(A)} + \underbrace{\big[\widehat{\mathcal{R}}_n(\widehat{w}) - \widehat{\mathcal{R}}_n(w_{\mathrm{ERM}})\big]}_{(C)}$$

$$+ \underbrace{\big[\widehat{\mathcal{R}}_n(w_{\mathrm{ERM}}) - \widehat{\mathcal{R}}_n(w^*)\big]}_{(D)\,\leq 0} + \underbrace{\big[\widehat{\mathcal{R}}_n(w^*) - \mathcal{R}(w^*)\big]}_{(B)}.$$

Theorem 6 gives

$$\mathbb{E}\big[(A)\big] \leq (1 + L/\rho)G\varepsilon_T.$$

Because $w^*$ is deterministic, $\mathbb{E}[(B)] = 0$. From Lemma 2, we have

$$(C) = \widehat{\mathcal{R}}_n(\widehat{w}) - \widehat{\mathcal{R}}_n(w_{\mathrm{ERM}}) \leq \frac{d_1}{d_2 + T} \tag{†}$$

$(D) \leq 0$ deterministically (by definition of the ERM) and thus can be discarded when taking an upper bound.

Taking expectations and using $\mathbb{E}[(B)] = 0$ and (†):

$$\mathbb{E}\big[\mathcal{R}(\widehat{w}) - \mathcal{R}(w^*)\big] \leq (1 + L/\rho)G\varepsilon_T + \frac{d_1}{d_2 + T}.$$

□

## C  THE USE OF LARGE LANGUAGE MODELS

We used LLMs solely to aid and polish writing, but not for retrieval and discovery (e.g., finding related work) or for research ideation.

