# STABILITY AND GENERALIZATION FOR BELLMAN RESIDUALS

## ABSTRACT

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

_{\boldsymbol{\theta}_2})(s, a, s') - \beta^2 \left( V_{Q_{\boldsymbol{\theta}_2}}(s') - \zeta_{\widetilde{\boldsymbol{\theta}_1}} \right)^2 \mid s, a \right] \right] \right\} \quad (3)$$

where $\widetilde{\boldsymbol{\theta}}_1 \in \operatorname{argmin}_{\boldsymbol{\theta}_1 \in \boldsymbol{\Theta}} \mathbb{E}_{(s,a)\sim\pi_D,\nu_0} \left[ \mathbb{E}_{s'\sim P(s,a)} \left[ \left( V_{Q_{\boldsymbol{\theta}_2}}(s') - \zeta_{\boldsymbol{\theta}_1}(s,a) \right)^2 \right] \right]$. Considering Equation equation 3 as the expected risk minimization problem, the corresponding empirical risk minimization problem can be written as

$$\min_{\boldsymbol{\theta}_2 \in \boldsymbol{\Theta}} \frac{1}{N} \sum_{(s,a,s')\in\mathcal{D}_{\pi_D,\nu_0}} \left[ \mathcal{L}_{\mathrm{TD}}(Q_{\boldsymbol{\theta}_2})(s,a,s') - \beta^2 \left( V_{Q_{\boldsymbol{\theta}_2}}(s') - \zeta_{\widetilde{\boldsymbol{\theta}}_1} \right)^2 \right] \tag{4}$$

$$\text{where } \widetilde{\boldsymbol{\theta}}_1 \in \operatorname{argmin}_{\boldsymbol{\theta}_1 \in \boldsymbol{\Theta}} \frac{1}{N} \sum_{(s,a,s')\in\mathcal{D}_{\pi_D,\nu_0}} \left[ \left( V_{Q_{\boldsymbol{\theta}_2}}(s') - \zeta_{\boldsymbol{\theta}_1}(s,a) \right)^2 \right] \tag{5}$$

and $\mathcal{D}_{\pi_D,\nu_0}$ is the offline data collected from following $\pi_D$ starting from $\nu_0$.

A canonical way of solving the mini-max problem is to apply the Stochastic Gradient Ascent Descent (SGDA) algorithm (Yang et al., 2020). Kang et al. (2025) proved that both Equation equation 3 and equation 4 satisfy the Polyak-Łojasiewicz condition for the parametrization with popular function classes such as Neural Network, and therefore the SGDA algorithm finds the $(\boldsymbol{\theta}_1, \boldsymbol{\theta}_2)$ that are global minima of Equation equation 4. In the following Section 2.2, we elaborate on the SGDA algorithm.

## 2.2 STOCHASTIC GRADIENT ASCENT–DESCENT ALGORITHM (SGDA)

As discussed earlier, Stochastic Gradient Ascent–Descent (SGDA) is the workhorse we use to solve the minimax problem equation 4. Given a function $f(w, v)$, at every iteration it performs a *descent* step on the primal variable $w$ (or $\theta_2$ in equation 3) and an *ascent* step on the dual variable $v$ (or $\theta_1$ in equation 3) using (possibly noisy) gradients computed from a minibatch of samples.

Let $\mathcal{D} = \{z_i\}_{i=1}^n$ denote the dataset, where each $z_i = (s_i, a_i, s_i')$ denotes a sample consisting of the current state, the action taken, and the resulting next state. Fix a minibatch size $B \in \{1, \ldots, n\}$. At round $t$ we draw an index set

$$I_t \subseteq [n], \qquad |I_t| = B,$$

either *with* or *without* replacement (our theory does not depend on this choice). With $f$ standing for the per–sample saddle objective introduced in equation 4, the averaged stochastic gradients are

$$g_t^w := \frac{1}{B} \sum_{i\in I_t} \nabla_w f\left(w_t, v_t; z_i\right), \qquad g_t^v := \frac{1}{B} \sum_{i\in I_t} \nabla_v f\left(w_t, v_t; z_i\right).$$

Unbiasedness is preserved: $\mathbb{E}[g_t^w] = \nabla_w F_D(w_t, v_t)$ and likewise for $v$, while the variance contracts by the usual $1/B$ factor. Using stepsize sequence $(\eta_t)_{t\geq 0}$, SGDA proceeds as

$$w_{t+1} = w_t - \eta_t\, g_t^w, \qquad v_{t+1} = v_t + \eta_t\, g_t^v.$$

The recursion can be written compactly as

$$(w_{t+1}, v_{t+1}) = (w_t, v_t) + \eta_t\left(-g_t^w,\, g_t^v\right),$$

which is the form used in the stability proofs of Section 3.

---

**Algorithm 1:** Minibatch SGDA on the empirical objective equation 4

**Input:** Dataset $\mathcal{D} = \{z_i\}_{i=1}^n$, minibatch size $B$, stepsizes $(\eta_t)$, initial $(w_0, v_0)$

1 **for** $t = 0$ **to** $T - 1$ **do**
2     Draw $I_t \subseteq [n]$ with $|I_t| = B$ uniformly at random;
3     $g_t^w \leftarrow \frac{1}{B} \sum_{i\in I_t} \nabla_w f(w_t, v_t; z_i)$;
4     $g_t^v \leftarrow \frac{1}{B} \sum_{i\in I_t} \nabla_v f(w_t, v_t; z_i)$;
5     $w_{t+1} \leftarrow w_t - \eta_t g_t^w$ ;             // gradient *descent*
6     $v_{t+1} \leftarrow v_t + \eta_t g_t^v$ ;             // gradient *ascent*

**Output:** $(w_T, v_T)$

---

The choice of $(\eta_t)$ follows the same Robbins–Monro conditions detailed after Theorem 3, and our theoretical bounds reflect the $1/B$ variance reduction exactly as discussed in the remark preceding equation 19 later. In practice, moderate minibatch sizes (e.g. $B \in [64, 512]$) strike a good

balance between numerical stability and computational throughput when training neural-network parameterizations.

Along with harmonic–stepsize, we can state the global convergence guarantee of ALGORITHM 1 in the parameter space:

**Lemma 2** (Global convergence of minibatch SGDA in parameter space (Yang et al., 2020; Kang et al., 2025)). *Let the iterates in* ALGORITHM 1 *be written as* $\{(\boldsymbol{\theta}_{2,t}, \boldsymbol{\theta}_{1,t})\}_{t \geq 0}$, *where* $\boldsymbol{\theta}_{2,t}$ *parametrises the action–value function* $Q_{\boldsymbol{\theta}_{2,t}}$ *(primal variable) and* $\boldsymbol{\theta}_{1,t}$ *parametrises* $\bar{\zeta}_{\boldsymbol{\theta}_{1,

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

}[R(w_T) - R_n(w_T)]$ and the weak primal–dual gap $|\mathbb{E}[\Delta^{\text{PD}}(w_T, v_T) - \Delta_n^{\text{PD}}(w_T, v_T)]|$ can be effectively upper bounded by constant times $\varepsilon_T$ from Theorem 3, which is $\mathcal{O}(1/n)$. The remainder of this subsection formalizes this to apply this transfer with our stability bound (Theorem 3) to obtain Theorem 6, $\mathcal{O}(1/n)$ generalization for BRM under SGDA.

**Lemma 5** (Theorem 5, Wang et al. (2022)). *Let Assumptions (A1), (A4), and (A5) hold. Let* $(w_T, v_T)$ *be the SGDA iterates produced on* $\mathcal{D}_n$ *and let*

$$\varepsilon_T \;=\; \frac{1}{n} \sum_{i=1}^{n} \mathbb{E}\big[ \|w_T(\mathcal{D}) - w_T(\mathcal{D}^{(i)})\| + \|v_T(\mathcal{D}) - v_T(\mathcal{D}^{(i)})\| \big]$$

*be the on-average argument-stability constant from Theorem 3. Then*

$$\mathbb{E}\big[R(w_T) - R_n(w_T)\big] \;\le\; (1 + L/\rho)\, G\, \varepsilon_T, \quad \big|\mathbb{E}\big[\Delta^{\mathrm{PD}}(w_T, v_T) \,-\, \Delta_n^{\mathrm{PD}}(w_T, v_T)\big]\big| \;\le\; G\,\varepsilon_T.$$

Combining Corollary 4 and Lemma 5 (Wang et al., 2022), we arrive at Theorem 6, the main result of this paper, i.e., the *generalization guarantee of Bellman residuals*. In words, the learned $Q$ (i.e., the corresponding learned $\boldsymbol{\theta}_2$) generalizes: its empirical Bellman residual on the offline dataset closely matches its expected Bellman residual on the true MDP distribution. This is direct from the fact that proving $\mathcal{O}(1/n)$ stability for SGDA immediately delivers $\mathcal{O}(1/n)$ generalization bounds for Bellman residual minimization (Wang et al., 2022).

**Theorem 6** (Generalization for the empirical Bellman–residual objective). *Let $\big(\widehat{\boldsymbol{\theta}}_1^{(T)}, \widehat{\boldsymbol{\theta}}_2^{(T)}\big)$ be the parameters returned by* ALGORITHM 1 *after $T$ SGDA iterations on the empirical objective equation 4. Define the population and empirical risks*

$$\mathcal{R}(\boldsymbol{\theta}_2) := \mathbb{E}_{(s,a)\sim\pi_D,\nu_0}\, \mathbb{E}_{s'\sim P(\cdot|s,a)}\Big[\mathcal{L}_{\mathrm{TD}}\big(Q_{\boldsymbol{\theta}_2}\big)(s,a,s') - \beta^2\big(V_{Q_{\boldsymbol{\theta}_2}}(s') - \zeta_{\widetilde{\boldsymbol{\theta}}_1^\star}(s,a)\big)^2\Big],$$

$$\widehat{\mathcal{R}}_n(\boldsymbol{\theta}_2) := \frac{1}{N} \sum_{(s,a,s')\in\mathcal{D}_{\pi_D,\nu_0}} \Big[\mathcal{L}_{\mathrm{TD}}\big(Q_{\boldsymbol{\theta}_2}\big)(s,a,s') - \beta^2\big(V_{Q_{\boldsymbol{\theta}_2}}(s') - \zeta_{\widetilde{\boldsymbol{\theta}}_1^\star}(s,a)\big)^2\Big],$$

*where $\widetilde{\boldsymbol{\theta}}_1^\star$ is the minimizer in equation 5. Then*

$$\mathbb{E}\Big[\mathcal{R}\big(\widehat{\boldsymbol{\theta}}_2^{(T)}\big) - \widehat{\mathcal{R}}_n\big(\widehat{\boldsymbol{\theta}}_2^{(T)}\big)\Big] \;\le\; (1 + L/\rho)\, G\, \varepsilon_T,$$

*where $\varepsilon_T$ is from Theorem 3. Moreover,*

$$\Big|\mathbb{E}\Big[\Delta^{\mathrm{PD}}\big(\widehat{\boldsymbol{\theta}}_2^{(T)}, \widehat{\boldsymbol{\theta}}_1^{(T)}\big) - \Delta_n^{\mathrm{PD}}\big(\widehat{\boldsymbol{\theta}}_2^{(T)}, \widehat{\boldsymbol{\theta}}_1^{(T)}\big)\Big]\Big| \;\le\; G\,\varepsilon_T.$$

The generalization guarantee in Theorem 6 is a critical result, confirming that the empirical risk is a reliable proxy for the true population risk. However, the ultimate measure of success for a learning algorithm is its performance on the population distribution relative to the best possible model. This is quantified by the *population excess risk*, which measures the gap $\mathcal{R}(\widehat{\boldsymbol{\theta}}_2^{(T)}) - \mathcal{R}(\boldsymbol{\theta}_2^*)$.

**Theorem 7** (Population excess risk). *Let $\big(\widehat{\boldsymbol{\theta}}_2^{(T)}, \widehat{\boldsymbol{\theta}}_1^{(

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

*Proof of Theorem 7.* Write $\widehat{\theta} := \widehat{\boldsymbol{\theta}}_2^{(T)}$, $\theta_{ERM} := \boldsymbol{\theta}_2^{\mathrm{ERM}}$ and $\theta^* := \boldsymbol{\theta}_2^*$. Add and subtract the empirical risks of these three parameters:

$$\mathcal{R}(\widehat{\theta}) - \mathcal{R}(\theta^*) = \underbrace{\left[\mathcal{R}(\widehat{\theta}) - \widehat{\mathcal{R}}_n(\widehat{\theta})\right]}_{(A)} + \underbrace{\left[\widehat{\mathcal{R}}_n(\widehat{\theta}) - \widehat{\mathcal{R}}_n(\theta_{ERM})\right]}_{(C)}$$
$$+ \underbrace{\left[\widehat{\mathcal{R}}_n(\theta_{ERM}) - \widehat{\mathcal{R}}_n(\theta^*)\right]}_{(D)\ \leq 0} + \underbrace{\left[\widehat{\mathcal{R}}_n(\theta^*) - \mathcal{R}(\theta^*)\right]}_{(B)}.$$

Theorem 6 gives

$$\mathbb{E}\big[(A)\big] \ \leq \ (1 + L/\rho)G\varepsilon_T.$$

Because $\theta^*$ is deterministic, $\mathbb{E}[(B)] = 0$. From Lemma 2, we have

$$(C) \ = \ \widehat{\mathcal{R}}_n(\widehat{\theta}) - \widehat{\mathcal{R}}_n(\theta_{ERM}) \ \leq \ \frac{d_1}{d_2 + T} \tag{†}$$

$(D) \leq 0$ deterministically (by definition of the ERM) and thus can be discarded when taking an upper bound.

Taking expectations and using $\mathbb{E}[(B)] = 0$ and (†):

$$\mathbb{E}\big[\mathcal{R}(\widehat{\theta}) - \mathcal{R}(\theta^*)\big] \ \leq \ (1 + L/\rho)G\varepsilon_T \ + \ \frac{d_1}{d_2 + T}.$$

$\qquad\square$

## C  THE USE OF LARGE LANGUAGE MODELS

We used LLMs solely to aid and polish writing, but not for retrieval and discovery (e.g., finding related work) or for research ideation.