# OpenReview forum: "Stability and Generalization for Bellman Residuals"
_ICLR.cc/2026/Conference — Submitted to ICLR 2026_

### Official Review · Reviewer_kFyc · 2025-11-01

**Soundness:** 3
**Presentation:** 3
**Contribution:** 3
**Rating:** 6
**Confidence:** 4

**Summary:**

This paper analyzes the statistical behavior of Bellman Residual Minimization (BRM) for offline RL/IRL. Building on the recent optimization view that the bi-conjugate BRM objective induces a PL–strongly-concave minimax structure, the authors couple two SGDA runs on neighboring datasets via (i) a single Lyapunov potential that mixes primal suboptimality and primal–dual mismatch, and (ii) a “ghost-index” device to decouple sampling noise. They prove on-average argument stability of SGDA with an O(1/n) rate (under Robbins–Monro stepsizes), and transfer this to O(1/n) generalization and an excess-risk bound that cleanly decomposes optimization and estimation errors. The setup, assumptions (A1–A9), and the transfer to weak PD-gap follow the minimax stability framework.

**Strengths:**

Originality
- Closes a real theoretical gap: Prior minimax stability analyses (e.g., Wang–Lei–Ying–Zhou, NeurIPS 2022) deliver O(n^{-1/2}) rates under convex–concave assumptions. This paper’s O(1/n) stability and generalization results for SGDA in a PL–strongly-concave regime appear novel.
- Combines multiple theoretical tools—bi-conjugate BRM formulation, PL geometry, a Lyapunov potential, and ghost-index coupling—into a coherent analysis without variance reduction or independence assumptions.
- The unification of optimization and generalization analysis through a single Lyapunov potential is an elegant methodological contribution.

Quality
- The proofs are internally consistent and technically sound under the stated assumptions (A1–A9). The Lyapunov-based stability recursion is clearly constructed and all major theorems are proven in full.
- The paper avoids dependence on variance-reduction or mixing assumptions, deriving O(1/n) bounds via standard SGDA under Robbins–Monro step sizes.
- The key limitations lie in the strong assumptions—bounded per-sample gradients, uniform constants across neighboring datasets, and uniqueness of the saddle—that may not strictly hold for deep neural networks.

Clarity
- The exposition is clear, particularly in articulating the problem gap (“optimization picture is clear; statistical picture remains open”).
- The algorithmic setup, potential function, and contraction argument are well explained with intuitive justification for summability of noise terms.
- Proof dependencies and structure are explicitly cross-referenced in the reproducibility statement, ensuring transparency.

Significance
- The results provide the first O(1/n) generalization bound for Bellman Residual Minimization in offline reinforcement learning, doubling the exponent achieved in prior convex–concave analyses.
- The theoretical framework may generalize to other PL-minimax problems beyond BRM, influencing theoretical and algorithmic directions in RL and IRL.
- While the assumptions restrict direct practical application, the analysis sets a higher theoretical standard for understanding statistical generalization in nonconvex–concave RL objectives.

**Weaknesses:**

1) Assumptions feel strong and under-motivated for neural BRM
Issue: The analysis depends on assumptions such as bounded per-sample gradients, uniform constants across neighboring datasets, and uniqueness of the saddle. These are not linked to concrete architectural or data-level conditions.
Actionable Fixes:
- Provide sufficient conditions (e.g., Lipschitz activations, spectral normalization, weight decay) ensuring these assumptions hold.
- Add perturbation lemmas for small constant drift across neighboring datasets.
- Explain how regularization ensures uniqueness of the saddle.

2) Positioning vs. existing stability literature could be sharper
Issue: The claimed novelty (O(1/n) vs O(1/√n)) relative to convex–concave minimax works (e.g., Wang et al., NeurIPS 2022) lacks a clear side-by-side comparison.
Actionable Fixes:
- Include a comparison table contrasting assumptions, settings, and rates.
- Explicitly highlight which steps rely on PL–strong concavity and would fail otherwise.

3) Minibatch dependence not clearly quantified
Issue: Theorems mention minibatch adaptation “verbatim” without giving explicit batch-size-dependent constants.
Actionable Fixes:
- Add a corollary deriving ε_T(B) with explicit 1/B scaling and its impact on generalization and excess-risk bounds.
- Provide practical guidance on choosing batch size B.

4) Lack of empirical sanity checks
Issue: The paper claims parametric O(1/n) scaling but shows no supporting experiment.
Actionable Fixes:
- Include a toy experiment using linear BRM satisfying all assumptions to empirically verify slope ≈ –1 in log–log plots.
- Compare against convex–concave baselines to show contrast.

5) Clarity gaps in bi-conjugate BRM formulation
Issue: The connection from the bi-conjugate Bellman residual to the minimax form is hard to follow for non-experts.
Actionable Fixes:
- Add a concise boxed derivation linking the BRM objective to the dual variable.
- Include a diagram illustrating shared-index coupling and “hit” events.

6) Excess-risk decomposition underemphasized
Issue: The clean decomposition between stability and optimization error appears late and without clear interpretation.
Actionable Fixes:
- Promote the decomposition as a boxed equation in the main text.
- Explain how tuning T and η_t balances the two error terms.

7) Limited discussion beyond entropy-regularized BRM
Issue: It is unclear whether the results extend to non-entropy (hard-max) BRM formulations.
Actionable Fixes:
- Add remarks outlining when PL–strongly-concave structure persists under different smoothings (e.g., Moreau envelopes).

8) Ambiguity in “one pass over n samples” phrasing
Issue: The notion of “one pass” may be misread without clarifying total gradient calls or sampling scheme.
Actionable Fixes:
- Specify whether T ≈ n steps correspond to one epoch and whether sampling is with or without replacement.

Overall, the paper would improve by making its assumptions verifiable in practice, providing explicit batch-size scaling, and including minimal empirical verification. These additions would make the theory more credible, checkable, and actionable for the ICLR audience.

**Questions:**

1. On Assumptions and Applicability
- Could you provide explicit sufficient conditions on the neural-network architecture or data distribution that ensure assumptions (A5) and (A8) hold? For example, do ReLU or tanh activations satisfy the Lipschitz and gradient-boundedness assumptions under spectral normalization or weight clipping?
- The analysis assumes a unique saddle point, yet neural networks are often overparameterized. Is uniqueness strictly necessary, or could the analysis extend to a set of equivalent saddles?

2. On Novelty and Positioning
- The claimed improvement from O(n^{-1/2}) to O(1/n) hinges on the PL–strongly-concave structure. Could you explicitly summarize which elements of your proof break down in purely convex–concave settings?
- To what extent could your Lyapunov and ghost-index coupling analysis extend to other PL-minimax settings (e.g., actor–critic or distributional RL formulations)?

3. On Practical Interpretability
- You mention that the minibatch setting follows “verbatim” with rescaled constants. Could you please provide the explicit scaling law of ε_T(B) in terms of B and n?
- When stating that you achieve the O(1/n) rate “after one pass over n samples,” do you mean T ≈ n SGDA steps, one epoch with sampling with or without replacement?

4. On Theoretical Sharpness
- Your current bounds are in expectation. Do you think similar rates could hold with high probability using martingale inequalities (e.g., Azuma or Freedman)? If so, how would the constants or rates degrade?
- Could you comment on how sensitive your results are to the condition numbers L/μ_PL and L/ρ?

5. On Empirical Verification
- Would you be open to adding a toy experiment (e.g., linear-quadratic BRM under the assumptions you make) to confirm the slope of the generalization error versus sample size?
- Even a small-scale plot could visually substantiate the theoretical rate and convince a broader ICLR audience.

6. On Extensions and Generality
- Your analysis focuses on the softmax (entropy-regularized) case. Could you clarify whether the PL–strongly-concave geometry and stability proof extend to hard-max or Moreau-smooth Bellman operators?
- Would your argument still hold under Markovian dependence rather than i.i.d. samples? If not directly, what modifications would be necessary to handle the mixing-time dependence?

7. On Presentation and Readability
- Could you include a short boxed derivation showing how the Bellman residual minimization problem transforms into the minimax form involving the dual variable?
- The final decomposition separating optimization and generalization errors is one of your most interpretable results. Consider moving it earlier into the main body with a brief intuitive discussion.

8. On Possible Future Directions
- How do you envision extending your analysis to policy-based or actor–critic settings, where the loss is not strictly bi-convex/bi-concave?

---

> ### Author Response · Authors · 2025-11-26
> **We thank the reviewer kFyc for helping us improve the paper's representation.**
>
> We thank the reviewer kFyc for helping us improve the paper's representation.
>
> ### **Answer to comments in "Weaknesses"**
>
> **[Weakness 1.] **
> > Assumptions feel strong and under-motivated for neural BRM Issue: The analysis depends on assumptions such as bounded per-sample gradients, uniform constants across neighboring datasets, and uniqueness of the saddle. These are not linked to concrete architectural or data-level conditions.
>
> **[Author response to Weakness 1.]**
>
> **1. On assumption of bounded per-sample gradients:**
> Thank you for giving us the chance to improve the paper. Fortunately, we found that we can avoid this assumption, as we only use a weaker one in the proof procedure. We appreciate the opportunity to correct this.
>
> We have updated the new draft, substituting the Assumption A5 to "effective domain" bound. That is, the assumed gradient's boundedness is only "along the optimization trajectory". Specifically, we say:
> * **(A5 [updated]) Bounded gradients on the effective domain.**
> There exists a compact convex set $\Omega \subset \mathcal{W} \times \mathcal{V}$ such that the sequence of iterates $\\{(w\_t, v\_t)\\}\_{t=0}^{T}$ generated by the algorithm remains within $\Omega$ almost surely.  We define $G$ as the uniform gradient bound on this set:
> \\[
> G := \\sup_{(w,v) \\in \\Omega, z \\in \\mathcal{Z}} \\max \\left\\{ \\|\\nabla_w f(w,v; z)\\|, \\|\\nabla_v f(w,v; z)\\| \\right\\} < \\infty.
> \\]
>
> And then we add these phrases right after:
>
> *Assumption A5 is justified by the coercivity of the PL-Strongly Concave landscape, which ensures iterates remain in a bounded sublevel set [(Yang et al., 2020)](https://arxiv.org/pdf/2002.09621) While strong concavity implies unbounded gradients on $\\mathbb{R}^d$, the geometry of problem equation 4 induces a drift that keeps iterates bounded (coercivity). Thus, we only require the gradient to be bounded within the effective domain $\\Omega$ visited by the algorithm, consistent with the global convergence guarantees for unprojected SGDA in this setting [(Yang et al., 2020)](https://arxiv.org/pdf/2002.09621).}*
> .
>
> **2. On assumption of uniform constants across neighboring datasets:**
> Thank you for pointing this out. We indeed lacked a paragraph on this, so we added the following in the new draft:
> Assumption A8 is standard in the stability literature: for example, Hardt et al. (2016) assume each per-example loss $f(\cdot ; z)$ is $L$-Lipschitz and $\beta$-smooth uniformly in $z$, and Wang et al. (2022) assume the gradients and smoothness of $f(w, v ; z)$ are bounded by global constants $G$ and $L$ for all $z$. These conditions immediately imply that the corresponding constants are identical for any dataset and its replace-one neighbour. Our Assumption A8 is the PL/QG analogue of these standard uniform assumptions.
>
> **3. On assumption of uniqueness of saddle:**
> Again, thank you for pointing this out. In the new draft, we emphasized that Assumption A9, the unique saddle point assumption, was shown to hold in Yang et al. (2020), which we cite in the paper.
>
> **[Weakness 2. ]**
> > Positioning vs. existing stability literature could be sharper Issue: The claimed novelty (O(1/n) vs O(1/√n)) relative to convex–concave minimax works (e.g., Wang et al., NeurIPS 2022) lacks a clear side-by-side comparison.
>
> **[Author response to Weakness 2.]**
> Thank you for pointing this out. In the new draft, we added the following table in the Appendix:
> | Work | Geometry | Setting | Statistical rate (in $n$) | Optimization rate (in $T$) | Independence assumptions |
> |------|----------|---------|----------------------------|-----------------------------|---------------------------|
> | This paper | PL–strongly concave | Offline BRM (Markov data) | $\\mathcal{O}\\!\\left(\\frac{1}{n}\\right)$ | $\\mathcal{O}\\!\\left((c\_2+T)^{-\\min\\{\\tfrac12,\\,\\tfrac{3c c\_1}{8}\\}}\\right)$ | No independence on minibatch indices |
> | Wang et al.\ 2022 | Convex–concave | Generic minimax (Markov data) | $\\mathcal{O}\\!\\left(\\frac{1}{\\sqrt{n}}\\right)$ | $\\mathcal{O}\\!\\left(\\frac{1}{\\sqrt{T}}\\right)$ | Similar shared-index coupling |

---

> ### Author Response · Authors · 2025-11-26
> **(Continued) Answer to comments in "Weaknesses" - 2**
>
> **[Weakness 3. ]**
> > Minibatch dependence not clearly quantified. Issue: Theorems mention minibatch adaptation "verbatim" without giving explicit batch-size-dependent constants.
>
> **[Author response to Weakness 3.]**
> Thank you for pointing this out. In the new draft, right after Theorem 3, we added the following remark clarifying that part:
>
> **Remark.** Theorem 3 stated for the single-sample case ($B=1$),
> where each SGDA update uses a single index $i\_t\\in[n]$ and the stochastic gradients are
> $g\_t^w=\\nabla\_w f(w\_t,v\_t;z\_{i\_t})$ and $g\_t^v=\\nabla\_v f(w\_t,v\_t;z\_{i\_t})$.
> The same proof applies verbatim when we use minibatches of size $B\\ge 1$, i.e.
> \\[
> g\_t^w \;=\; \\frac{1}{B}\\sum\_{j\\in I\_t} \\nabla\_w f(w\_t,v\_t;z\_j),
> \\qquad
> g\_t^v \;=\; \\frac{1}{B}\\sum\_{j\\in I\_t} \\nabla\_v f(w\_t,v\_t;z\_j),
> \\]
> where $I\_t\\subset[n]$ is a uniformly random subset of size $B$ (with or without replacement).
>
> Conditioned on the dataset $\\mathcal D$ and on $(w\_t,v\_t)$, the per-sample gradients
> $\\{\\nabla f(w\_t,v\_t;z\_j)\\}\_{j=1}^n$ are deterministic vectors, and the only randomness comes
> from the subset $I\_t$. Finite-population sampling bounds then yield
> \\[
> \\mathbb{E}\\bigl[\\|g\_t^w-\\nabla\_w F\_{\\mathcal D}(w\_t,v\_t)\\|^2 \\,\\big|\\,\\mathcal D,w\_t,v\_t\\bigr]
> \\;\\le\\; \\frac{C}{B}\\,G^2,
> \\]
> and analogously for $g\_t^v$, for some numerical constant $C>0$.
> In the proof of Theorem 3, this changes only the
> variance coefficient $C\_{\\mathrm{var}}$ to $C\_{\\mathrm{var}}/B$.
>
>
>
> **[Weakness 4. ]**
> > Lack of empirical sanity checks Issue: The paper claims parametric O(1/n) scaling but shows no supporting experiment. Actionable Fixes:
>
> **[Author response to Weakness 4.]**
> Thank you for pointing this out. We hope the reviewer will generously understand that it is not a usual practice for theoretical papers discussing stability bounds and corresponding generalization guarantees to include experiments. Please refer to:
> [Wang et al. 2022](https://proceedings.neurips.cc/paper_files/paper/2022/file/f61538f83b0f19f9306d9d801c15f41c-Supplemental-Conference.pdf),
> [Bousquet and Elisseeff 2002](https://www.jmlr.org/papers/volume2/bousquet02a/bousquet02a.pdf),
> [Charles and Papailiopoulos 2018](https://proceedings.mlr.press/v80/charles18a/charles18a.pdf)
> [Feldman Vondrak](https://proceedings.neurips.cc/paper/2018/file/05a624166c8eb8273b8464e8d9cb5bd9-Paper.pdf)

---

> ### Author Response · Authors · 2025-11-26
> **(Continued) Answer to comments in "Weaknesses" - 3**
>
> **[Weakness 6. ]**
> > Excess-risk decomposition underemphasized Issue: The clean decomposition between stability and optimization error appears late and without clear interpretation.
>
> **[Author response to Weakness 6.]**
> Thank you for pointing this out. We modified the representation of the Theorem 7 as:
>
> Let $\\bigl(\\widehat{w}^{(T)},\\widehat{v}^{(T)}\\bigr)$
> be the SGDA iterate outcome of Algorithm 1 after $T$ steps and
> $w^{\\ast}:=\\arg\\min\_{w\\in\\mathcal{W}}\\mathcal R(w)$
> its population minimiser.
> Then, with $\\varepsilon\_T$ from
> Theorem 3,
> \\[
> \\mathbb{E}\\Bigl[
>       \\mathcal R\\!\\bigl(\\widehat{w}^{(T)}\\bigr)
>       -\\mathcal R\\!\\bigl(w^{\\ast}\\bigr)
> \\Bigr]
> \\;\\le\\;\\underbrace{(1+L/\\rho)\\,G\\,\\varepsilon\_T}\_{\\text{stability / generalization}}
> \\;+\\;
> \\underbrace{\\frac{d\_{1}}{d\_{2}+T}}\_{\\text{optimization error}}
> \\]
> where $d\_1$ and $d\_2$ are defined in Lemma 2.
>
> **[Weakness 7. ]**
> > Limited discussion beyond entropy-regularized BRM Issue: It is unclear whether the results extend to non-entropy (hard-max) BRM formulations.
>
> **[Author response to Weakness 7.]**
> Thank you for pointing this out. PL-strongly-concavity, the key structure of BRM we utilize, relies crucially on the log-sum-exp structure induced by the Gumbel noise model now typically used in RL literature. For the hard-max Bellman operator, the minimax objective becomes non-smooth, and we cannot, at present, prove an analogous PL-strongly-concave structure. Extending our analysis directly to the hard max is not the scope of our paper.
>
> **[Weakness 8. ]**
> > Ambiguity in “one pass over n samples” phrasing Issue: The notion of “one pass” may be misread without clarifying total gradient calls or sampling scheme.
>
> **[Author response to Weakness 8.]**
> Thank you for pointing this out. As this is unnecessary complication, we removed the sentence from our paper.

---

> ### Author Response · Authors · 2025-11-26
> **Answer to comments in "Questions"**
>
> ### Answers to comments in "Questions"
>
> **[Question 1.]**
> > Could you provide explicit sufficient conditions on (A5) and (A8) hold?
>
> **[Author response to Question 1.]**
>
> Please refer to our question in [Author response to Weakness 1.].
>
> **[Question 2.]**
> > Could you provide explicit sufficient conditions on (A5) and (A8) hold?
> The claimed improvement from O(n^{-1/2}) to O(1/n) hinges on the PL–strongly-concave structure. Could you explicitly summarize which elements of your proof break down in purely convex–concave settings?
>
> **[Author response to Question 2.]**
>
> Please refer to [Author response to Weakness 2.].
>
> **[Question 3.]**
> > To what extent could your Lyapunov and ghost-index coupling analysis extend to other PL-minimax settings (e.g., actor–critic or distributional RL formulations)? How do you envision extending your analysis to policy-based or actor–critic settings, where the loss is not strictly bi-convex/bi-concave?
>
> **[Author response to Question 3.]**
>
> We don't have a definite answer to this question as this is out of our paper's scope, but it is a great extension for the future research. Thank you for asking this question.
>
> **[Question 4.]**
> > You mention that the minibatch setting follows “verbatim” with rescaled constants. Could you please provide the explicit scaling law of ε_T(B) in terms of B and n?
> When stating that you achieve the O(1/n) rate “after one pass over n samples,” do you mean T ≈ n SGDA steps, one epoch with sampling with or without replacement?
>
> **[Author response to Question 4.]**
> Please refer to [Author response to Weakness 3.]
>
>
> **[Question 5.]**
> > Your current bounds are in expectation. Do you think similar rates could hold with high probability using martingale inequalities (e.g., Azuma or Freedman)? If so, how would the constants or rates degrade?
> Could you comment on how sensitive your results are to the condition numbers L/μ_PL and L/ρ?
>
> **[Author response to Question 5.]**
>
> We don't have a definite answer to this question as this is out of our paper's scope, but it is a great extension for the future research. Thank you for asking this question.
>
>
>
> **[Question 6.]**
> > Would you be open to adding a toy experiment (e.g., linear-quadratic BRM under the assumptions you make) to confirm the slope of the generalization error versus sample size?
> Even a small-scale plot could visually substantiate the theoretical rate and convince a broader ICLR audience.
>
> **[Author response to Question 6.]**
>
> Please refer to our answer in [Author response to Weakness 4.]
>
>
> **[Question 6.]**
> > Your analysis focuses on the softmax (entropy-regularized) case. Could you clarify whether the PL–strongly-concave geometry and stability proof extend to hard-max or Moreau-smooth Bellman operators?
>
> **[Author response to Question 6.]**
>
> We don't have a definite answer to this question as this is out of our paper's scope, but it is a great extension for the future research. Thank you for asking this question.
>
> **[Question 7.]**
> > Would your argument still hold under Markovian dependence rather than i.i.d. samples? If not directly, what modifications would be necessary to handle the mixing-time dependence?
>
> **[Author response to Question 7.]**
> Thank you for this question. This paper's motivation is to address Markovian dependence rather than i.i.d. samples. For i.i.d. sample cases, please refer to [Charles and Papailiopoulos 2018.](https://proceedings.mlr.press/v80/charles18a/charles18a.pdf)

---

> ### Comment · Reviewer_kFyc · 2025-11-28
>
> The rebuttal makes several useful clarifications: it refines assumption A5 to effective-domain gradient boundedness, relates A8 to standard Lipschitz/smoothness conditions in the stability literature, clarifies the role of a unique saddle, spells out the minibatch scaling, and highlights the excess-risk decomposition more clearly. I also appreciate the comparison table against Wang et al. (2022) and the explicit statement that the analysis is designed for Markov (rather than i.i.d.) data.
>
> The following concerns from my review were addressed in the rebuttal and revised draft:
>
> Revision of Assumption A5 to effective-domain gradient boundedness
> Clarification of Assumption A8 (uniform constants across neighboring datasets)
> Clarification regarding uniqueness of the saddle (Assumption A9)
> Sharper positioning versus convex–concave minimax literature (comparison table added)
> Explicit minibatch scaling and variance reduction factor (Weakness 3)
> Removal of ambiguous “one pass over n samples” phrasing
> Clearer presentation of the excess-risk decomposition
> Clarification that the analysis is motivated by Markovian (not i.i.d.) data (Question 7)
>
> Partially addressed or unresolved points:
>
> 1. Empirical sanity checks (Weakness 4, Question 6 – toy experiment).
>
> My suggestion was for a very small-scale experiment (e.g., linear-quadratic BRM under your assumptions) to empirically confirm the O(1/n) slope and make the work more accessible to the ICLR audience. The rebuttal argues that it is not usual practice in stability papers to include experiments and refers to prior theoretical work. While I understand this perspective, the main claimed advance here is an improved rate in a concrete RL setting, and a simple synthetic plot would substantially strengthen the paper. This remains, in my view, an unrealized opportunity.
>
> 2. Extensions to other PL–minimax and actor–critic settings (Question 3).
>
> You state that this is outside the scope of the current paper and a direction for future work. That is reasonable, but it means the potential generality of the Lyapunov + ghost-index coupling framework remains speculative.
>
> 3. High-probability bounds and condition-number sensitivity (Question 5).
>
> Again, you defer this to future work. This is fine, but readers should understand that the current results are purely in expectation and that high-probability analogues are nontrivial.
>
> 4. Hard-max / other smoothings (Weakness 7, Question 6 – hard-max / Moreau).
>
> The rebuttal clarifies that PL–strong concavity relies on the log-sum-exp/Gumbel structure and does not directly extend to the hard-max Bellman operator. This is a good clarification, though it also emphasizes that the results are specific to entropy-regularized BRM. A brief remark in the main text about what exactly fails for hard-max would be very helpful for readers.
>
> 5. Practical verifiability of assumptions.
>
> Even after the improvements, there is still relatively little guidance on concrete architectural or regularization choices that would ensure the assumptions for neural BRM in practice (e.g., specific norms or spectral controls). This is not fatal for a theory paper but limits direct applicability.
>
> Overall assessment and score
>
> Several assumptions are better justified, the connection to prior minimax stability work is clearer, minibatch scaling is explicit, and the excess-risk decomposition is more visible. The core theoretical contribution—a clean O(1/n) stability/generalization rate for BRM in a PL–strongly-concave minimax setting—remains intact.
>
> At the same time, some of the more practice-facing aspects (minimal empirical confirmation, concrete sufficient conditions, and discussion of extensions) remain either deferred or only qualitatively addressed.
>
> Balancing these considerations, I maintain my original overall assessment and keep my score at 6 (marginally above the acceptance threshold). But I don't mind if this paper is rejected based on other reviewers' opinion.

---

### Official Review · Reviewer_uiVV · 2025-11-02

**Soundness:** 3
**Presentation:** 3
**Contribution:** 3
**Rating:** 6
**Confidence:** 2

**Summary:**

This paper studies Bellman Residual Minimization (BRM) for offline RL. Using a bi-conjugate reformulation, minimizing MSBE is turned into a Polyak--Łojasiewicz (PL)–strongly-concave minimax problem that can be solved by SGDA, thereby avoiding the double sampling problem. The analysis couples two SGDA runs on neighboring datasets and proves on-average algorithmic stability with an $O(1/n)$ rate, without requiring variance reduction or independence assumptions. By stability-to-generalization transfer, the work bounds (i) the gap between population and empirical Bellman-residual risks and (ii) the population Bellman-residual risk of the SGDA output.

**Strengths:**

- Without requiring independence assumptions on the sample indices nor variance reduction, the paper establishes an $O(1/n)$ on-average stability and, via stability-to-generalization transfer, an $O(1/n)$ generalization bound for BRM, doubling the exponent from $1/2$ to $1$ over prior work.

- The population excess risk is cleanly decomposed into an optimization term that decays with training and a sample-size–dominated statistical term, naturally aligning with standard minibatch SGDA.

- All assumptions are stated explicitly and clearly, making the analysis easy to follow.

**Weaknesses:**

- It would be helpful to add illustrative examples and comparisons to aid understanding (see Q 1 and 2).


- Sections~2 and 3 include substantial repetition of well-known material, and the exposition feels overly long. For example, the standard SGDA routine could be moved to the appendix for brevity.

**Questions:**

- How strong is Assumption A8? Do the constants remain unchanged under a single-sample replacement in general setting, and could the authors provide a concrete example illustrating when A8 holds or fails?

- In Corollary~4, could you quantify the iteration threshold $T^\star$ at which the optimization term is below the statistical term formally? Additionally, for the small-$T$, could you provide a comparison with prior methods?

- Would it be possible to use one of $(w,v)$ or $(\theta_1,\theta_2)$ to unify the notation since these seem to denote the same primal/dual variables?

---

> ### Author Response · Authors · 2025-11-26
> **We thank the reviewer uiVV for helping us improve the paper's representation.**
>
> We thank the reviewer uiVV for helping us improve the paper's representation.
>
> ### **Answer to comments in "Weaknesses":**
>
> **Weakness 1.**
> > It would be helpful to add illustrative examples and comparisons to aid understanding (see Q 1 and 2).
>
> **Author response to Weakness 1:**
> Thank you for pointing this out. We will do our best to address this in our answers to Q1 and Q2 below.
>
>
> **Weakness 2.**
> > Sections~2 and 3 include substantial repetition of well-known material, and the exposition feels overly long. For example, the standard SGDA routine could be moved to the appendix for brevity.
>
> **Author response to Weakness 2:**
> We agree that we included substantial repetition of materials, considering potential variance in reviewer backgrounds. Given the chance to write the camera-ready version, we consider moving those parts to the Appendix.
>
> ### **Answer to comments in "Questions":**
>
> **Question 1.**
> > How strong is Assumption A8? Do the constants remain unchanged under a single-sample replacement in a general setting, and could the authors provide a concrete example illustrating when A8 holds or fails?
>
> **Author response to Question 1:**
>
> Thank you for giving us the chance to improve the draft. The previous draft indeed lacked discussion of this, and we appreciate you pointing it out. In the new draft, we included the following missing paragraph:
>
> *Assumption A8 is standard in the stability literature: for example, Hardt et al. (2016) assume each per-example loss $f(\cdot ; z)$ is $L$-Lipschitz and $\beta$-smooth uniformly in $z$, and Wang et al. (2022) assume the gradients and smoothness of $f(w, v ; z)$ are bounded by global constants $G$ and $L$ for all $z$. These conditions immediately imply that the corresponding constants are identical for any dataset and its replace-one neighbour. Our Assumption A8 is the PL/QG analogue of these standard uniform assumptions.*
>
> **Question 2.**
> > In Corollary $4$, could you quantify the iteration threshold $T^{\\star}$ at which the optimization term is below the statistical term formally? Additionally, for the small-$T$, could you provide a comparison with prior methods?
>
> **Author response to Question 2:**
> Thank you for pointing this out. In the new draft, we added the following clarification in the Appendix:
>
> | Work | Geometry | Setting | Statistical rate (in $n$) | Optimization rate (in $T$) | Independence assumptions |
> |------|----------|---------|----------------------------|-----------------------------|---------------------------|
> | This paper | PL–strongly concave | Offline BRM (Markov data) | $\\mathcal{O}\\!\\left(\\frac{1}{n}\\right)$ | $\\mathcal{O}\\!\\left((c\_2+T)^{-\\min\\{\\tfrac12,\\,\\tfrac{3c c\_1}{8}\\}}\\right)$ | No independence on minibatch indices |
> | Wang et al.\ 2022 | Convex–concave | Generic minimax (Markov data) | $\\mathcal{O}\\!\\left(\\frac{1}{\\sqrt{n}}\\right)$ | $\\mathcal{O}\\!\\left(\\frac{1}{\\sqrt{T}}\\right)$ | Similar shared-index coupling |
>
> **Question 3.**
> > Would it be possible to use one of $(w, v)$ or $\left(\theta_1, \theta_2\right)$ to unify the notation since these seem to denote the same primal/dual variables?
>
> **Author response to Question 3:**
> Thank you for your response. We agree wholeheartedly that unification significantly improves readability, and we replaced all $\left(\theta_1, \theta_2\right)$ with $(w, v)$.
>
> Again, we enormously appreciate the efforts you put into reviewing this paper. We were able to enjoy the representation of the paper significantly thanks to your comments.

---

### Official Review · Reviewer_yTNX · 2025-11-02

**Soundness:** 3
**Presentation:** 2
**Contribution:** 2
**Rating:** 2
**Confidence:** 3

**Summary:**

The paper analyzes the excess risk bound for offline reinforcement learning in view of Bellman residual minimization.

**Strengths:**

The problem of analyzing the excess risk bound for Bellman residual minimization does seem open so far.

**Weaknesses:**

- The comparison to existing works in approximate dynamic programming methods e.g. projected Bellman equation-based approaches seems inadequate. Is Bellman residual minimization the only way to accommodate the difficulty of enforcing Bellman consistency? What are the other existing risk bounds when incorporating function approximations and how do these results compare?
- The techniques used seem to be standard, e.g. PL for analyzing SGDA etc. It seems unclear from the manuscript what are the technical challenges and the techniques developed in this paper that are independent of the developments from combining Kang et al. 2025 and Wang et al 2022. What is the motivation when defining the Lyapunov potential? Some discussions around lines 369-375 when introducing this object would greatly help the reader.
- The presentation of Theorem 6 and in general Section 3 can be improved. As far as I understand, this paper is considering the specific problem of learning the (action)-value function, and thus introducing 9 assumptions for a general function F and auxiliary results about general risks introduces additional notation while not clear to what extent they are helpful in elucidating the final result (Theorem 6). I would think a clearer explanation why value functions and lyapunov potential satisfy the assumptions needed to establish Theorem 6 and intuition of the result would be more helpful than the results about general F along with 9 additional assumptions (that will automatically be satisfied).


Minor points:
- There are superfluous "equation" when referring to equations throughout the paper, e.g., Equation 4, etc. Please remove those.
- Line 80: "Throughout, focus on single-agent decision making problem interacting with a discounted Markov Decision Process (MDP) described by the tuple ( S, A, P, r, β , ν 0)" is lacking a subject.
- Bellman consistency in line 38 comes out directly without motivation or explanation. Why do we want consistency and what does it mean? In the last sentence you said "satisfies the Bellman optimality equations even though no new state–action pairs can be queried." but Bellman consistency means fixed point of Bellman equations, which is not shown here.

**Questions:**

See previous section

---

> ### Author Response · Authors · 2025-11-25
> **We thank the reviewer yTNX for helping us improve the paper's representation.**
>
> We thank the reviewer yTNX for helping us improve the paper's representation. We do not want to confuse readers about the literature and contributions, and your comments on those points are extremely valuable to us in improving the value of this paper.
>
> ### **Answer to comments in "Weaknesses"**
> **Comment 1.**
> > "The comparison to existing works in approximate dynamic programming methods, e.g., projected Bellman equation-based approaches, seems inadequate. Is Bellman residual minimization the only way to accommodate the difficulty of enforcing Bellman consistency? What are the other existing risk bounds when incorporating function approximations and how do these results compare?."
>
> **Author response to Comment 1:**
> You are absolutely right that discussing methods such as the projected mean-squared projected Bellman error (MSPBE) minimization methods will be valuable to readers. We will add [Patterson et al. 2024](https://arxiv.org/abs/2104.13844) as the reference.
> 1. **The difference between mean-squared projected Bellman error (MSPBE) minimization and BRM.** In contrast to BRM, which directly minimizes the mean-squared Bellman residual, MSPBE-based methods minimize the projected Bellman error, i.e., the squared norm of the Bellman residual after projection onto the approximation space. Thus, MSPBE is a surrogate for the true MSBE that is exact only at the level of the projected Bellman equation, not the original Bellman operator. To see this, define
> - \\( V\_\\theta : \\mathcal S \\to \\mathbb R \\) be a value function in a function class \\( \\mathcal F = \\{ V\_\\theta \\} \\),
> - \\( T \\) be the Bellman operator for a fixed policy,
> - \\( D \\) be a reference distribution over states,
> - \\( \\lVert \\cdot \\rVert\_D \\) be the \\(L^2(D)\\) norm.
>
> The **Bellman residual** at \\(V\_\\theta\\) is
> \\[
> e\_\\theta := T V\_\\theta - V\_\\theta.
> \\]
>
> The **Mean Squared Bellman Error (MSBE)** is
> \\[
> \\mathrm{MSBE}(\\theta)
> := \\lVert e\_\\theta \\rVert\_D^2
> = \\lVert T V\_\\theta - V\_\\theta \\rVert\_D^2.
> \\]
>
> Now let \\(\\Pi\\) be the projection onto the function class \\(\\mathcal F\\) (e.g., onto the span of features in the linear case), with respect to \\(\\lVert \\cdot \\rVert\_D\\). The **Mean Squared Projected Bellman Error (MSPBE)** is defined as
> \\[
> \\mathrm{MSPBE}(\\theta)
> := \\lVert \\Pi e\_\\theta \\rVert\_D^2
> = \\lVert \\Pi (T V\_\\theta - V\_\\theta) \\rVert\_D^2
> = \\lVert \\Pi T V\_\\theta - V\_\\theta \\rVert\_D^2.
> \\]
>
> Since \\(\\Pi\\) is an orthogonal projection (in the linear case), we have the Pythagorean decomposition
> \\[
> e\_\\theta
> = \\Pi e\_\\theta + (I - \\Pi)e\_\\theta,
> \\]
> with $\Pi e_\theta$ in the function class and $(I-\Pi) e_\theta$ orthogonal to it.
>
> Therefore, we have $\\left\\|e\_\\theta\\right\\|\_D^2
> =\\left\\|\\Pi e\_\\theta\\right\\|\_D^2
> +\\left\\|(I-\\Pi) e\_\\theta\\right\\|\_D^2
> =\\operatorname{MSPBE}(\\theta)
> +\\underbrace{\\left\\|(I-\\Pi) e\_\\theta\\right\\|\_D^2}\_{\text{ignored by MSPBE}}$.
>
> Compared to MSBE, MSPBE only measures the **component of the Bellman residual that lies in the function class** (the projected part \\(\\Pi e\_\\theta\\)), and ignores the component \\((I - \\Pi)e\_\\theta\\) that is orthogonal to the approximation space.
>
> 2. As [Jiang and Xie (2025)](https://nanjiang.cs.illinois.edu/files/STS_Special_Issue_Offline_RL.pdf) discuss, approximate dynamic programming methods do not extend well to parametrization beyond linear parametrization; they diverge (Proposition 1 of [Jiang and Xie (2025)](https://nanjiang.cs.illinois.edu/files/STS_Special_Issue_Offline_RL.pdf)). In addition, marginal importance sampling-based methods do not guarantee Bellman consistency.
>
> 3. For these reasons, when the goal is Bellman consistency, no exact and stable method has been developed so far when parametrized with a neural network, other than BRM. Therefore, we don't provide existing risk bounds for different methods with neural network parametrization.

---

> ### Author Response · Authors · 2025-11-25
> **(Continued) Answer to comments in "Weaknesses" - 2**
>
> **Comment 2.**
> >The techniques used seem to be standard, e.g. PL for analyzing SGDA etc. It seems unclear from the manuscript what the technical challenges are and what techniques were developed in this paper that are independent of the developments from combining Kang et al. 2025 and Wang et al 2022. What is the motivation when defining the Lyapunov potential? Some discussions around lines 369-375 when introducing this object would greatly help the reader.
>
> **Author response to Comment 2:**
>
> Thank you for this comment. It is crucial to explain to readers what our contribution is clearly. We also added an intuitive explanation of the Lyapunov potential in our new draft.
>
> 1. **What Kang et al. give us, and what they do not :**  Kang et al. (2025) establish that, after a bi-conjugate transformation, both the population and empirical BRM objectives enjoy PL-strongly-concave structure in the parameterization of $Q$ and $\zeta$. They then prove global convergence of SGDA to the empirical minimizer, and use PL to translate optimization error into parameter error. However, they:
> - do not analyze algorithmic stability,
> - do not analyze generalization (population vs empirical BRM).
> In fact, in the latest version of Kang et al., Lemma 28 uses our present result for the BRM sample complexity. So from their side, we only inherit the PL and strong-concavity constants for BRM.
>
> 2. **What Wang et al. give us – and what they do not :** Wang et al. (2022) develop on-average argument stability for Markov chain SGD/SGDA and show that for convex-concave or strongly-convex-strongly-concave objectives the excess population risk scales as $O(1 / \sqrt{n})$, with explicit dependence on the mixing parameter of the Markov chain over indices. We reuse from Wang et al.:
> - The concept of on-average argument stability and its connection to generalization (Lemma 5 in our notation).
> - Some proof templates: two-run coupling on neighboring datasets and the idea of counting "hits" of the replaced data point via an indicator $\\mathbf{1}\\left\\{i\_t=i\\right\\}$.
> However, there are two crucial structural differences:
> - Geometry: Wang et al. assume convex-concave or $\rho$-strongly-convex-strongly-concave objectives. Our BRM objective is nonconvex in the primal parameters; only the value function $\Phi_D(w):=\max _v F_D(w, v)$ is PL (and QG ), and only in $w$. The algorithm, however, is run on the bilevel saddle objective $F_D(w, v)$, not on $\Phi_D$. This destroys the standard convexity-based distance contraction used in Wang et al.
> - Object of interest: Wang et al. work directly with the original minimax $F$ and directly analyze its risk/generalization. We instead are interested in the value function $\Phi_D$ (Bellman residual) and the primal-dual gap built on $\Phi_D$. The gradients used by SGDA at time $t$ are biased w.r.t. $\nabla \Phi_D\left(w_t\right)$ because the dual variable is not at its maximizer:
> $$
> \Delta_t:=\nabla_w F_D\left(w_t, v_t\right)-\nabla \Phi_D\left(w_t\right) \neq 0
> $$
> This mismatch term $\Delta_t$ is absent in Wang et al. and is precisely what drives the need for our Lyapunov potential.
>
> 3. **Why Lyapunov? What is its role?**  When we define $\\Psi\_{\\alpha, D}(w, v):=\\underbrace{\\Phi\_D(w)-\\Phi\_D^{\\star}}\_{A(w)}+\\alpha \\cdot \\underbrace{\\left(\\Phi\_D(w)-F\_D(w, v)\\right)}\_{\\Gamma(w, v)},\\quad \\alpha \\geq \\frac{4 L^2}{\\rho^2}$, we are trying to solve the following problem: *We want a single scalar quantity that (i) measures how far the primal iterate is from optimal in terms of the value function $\Phi_D$, and (ii) penalizes the dual mismatch $\Gamma(w, v)$ strongly enough that the gradient used in SGDA is effectively aligned with $\nabla \Phi_D(w)$.*. This type of potential does not appear in Kang et al. (who only need PL to control optimization error for a single run, not coupled runs) nor in Wang et al. (where convex-concavity avoids this mismatch altogether). It is specifically tailored to:
> - The PL–strongly-concave structure of BRM, and
> - The need to couple two SGDA runs on neighboring datasets in a way that keeps track of both the primal and dual errors simultaneously.

---

> ### Author Response · Authors · 2025-11-25
> **(Continued) Answer to comments in "Weaknesses" - 3**
>
> **Comment 3.**
> > The presentation of Theorem 6 and in general Section 3 can be improved. As far as I understand, this paper is considering the specific problem of learning the (action)-value function, and thus introducing 9 assumptions for a general function F and auxiliary results about general risks introduces additional notation while not clear to what extent they are helpful in elucidating the final result (Theorem 6). I would think a clearer explanation why value functions and lyapunov potential satisfy the assumptions needed to establish Theorem 6 and intuition of the result would be more helpful than the results about general F along with 9 additional assumptions (that will automatically be satisfied).
>
> **Author response to Comment 3**:
>
> We genuinely appreciate this comment, as this shows the reviewer's efforts in reading this paper. We indeed missed one explanation paragraph that motivates our abstraction to the general function f and F, and this must frustrate every reader who carefully follows the paper's flow. To address this, we added the following paragraph before we discuss f and F:
>
> (Start of added paragraph)
>
> *For the stability analysis, it is convenient to momentarily step back from the specific Bellman–residual objective and view our problem as a generic empirical PL--strong-concave saddle-point problem. In the BRM formulation, the primal variable $w$ parametrizes the action–value function $Q\_w$, the dual variable $v$ parametrizes the auxiliary function $\\zeta\_v$, and the empirical objective in equation 4 is an average over samples $z=(s,a,s')$ drawn from the offline dataset. In this section, we abstract this structure and write $f(w,v;z)$ for the per–sample saddle loss and $F\_D(w,v)$ for its empirical average over a dataset $D=\\{z\_i\\}\_{i=1}^n$. All assumptions (A1)–(A9) that we impose below are satisfied by the BRM objective under the conditions of Section 2 and the PL–strongly-concave properties established by Kang et al. 2025. Working in this slightly more abstract template keeps the proofs uncluttered; in Theorem 6, we then specialize the resulting stability and generalization bounds back to BRM.*
>
> (End of added paragraph)
>
> **Minor points**: We addressed all points the reviewer raised.
>
> Again, we appreciate your enormous effort in reviewing the paper and the opportunity to improve the representation issue that could have confused potential readers.

---

> > ### Comment · Area_Chair_16qU · 2025-11-26
> >
> > The evaluation of this submission has a large variance so we need to gather more information from both the authors and the reviewers. Please take a look at the author's response and see how it affects your evaluation.
> >
> > Best,
> > AC

---

### Official Review · Reviewer_MpYr · 2025-11-09

**Soundness:** 3
**Presentation:** 3
**Contribution:** 2
**Rating:** 2
**Confidence:** 4

**Summary:**

The paper considers the problem of minimizing the Bellman error in a TD (Temporal Difference) update and cast this as a minimax problem of optimizing an objective that involves two parameterized functions : 1) Q function as a function of state and action and 2) The other is a parameterized neural net that given current state and action approximates the value function of the future sampled state. This optimization objective is derived (from prior work) by characterizing the bias between the squared Bellman error with respected to the expected TD operator and sampled TD Bellman error.  Further the surprising fact about this parameterization is that the problem is concave with respect the second function and the objective after inner optimization satisfies the PL condition with respect to the first Q function when you consider the stochastically approximated variant under general parameterizations (specifically linear function approximation).

Motivated by this, the authors propose to perform a stability analysis that would bound the generalization error (in terms of the duality gap) between the mini max problems which sees the population version and the the sample version. Authors adopt the stability analysis (that is known to imply generalization in the sense of duality gap from prior work) where the mini max problem see two sets of sequence of samples (state transitions) where one of the samples is different and authors seek to bound the distance of between the primal and dual iterates of these two coupled minimax problems.

 Authors introduce two interesting ideas: 1) Ghost index which is an index independently sampled from the dataset which is independent of the Filtration and gradient with respect to this sample in expectation can approximate the population gradient 2) PL condition implies for the outer problem and strong concavity for the inner problem imply contraction for a Lyapunov function that is a combination of the primal gap and the dual gap in the expected function value.

Authors use this and existing results about stability to prove generalization of the primal and dual gap from sample to the population version.

**Strengths:**

The paper (to my knowledge) is the first to consider stability analysis exploiting the PL condition and strong concavity of the respective problem to show generalization errors in primal and dual gaps.  There are a lot of algebraic manipulations that deftly use the ghost index, contraction properties of the outer and inner problem to establish bounds on generalization error. The application to Bellman residual optimization is noteworthy although it borrows heavily from prior work.

**Weaknesses:**

1) My first concern is inadequate quoting of results from Kang et al 2025 that misleads reading this paper. Line 230 and 231 says that Kang et al. 2025 proved that PL condition is satisfied with respect to the parameters of the Q function (primal variables) when parameterized by a Neural Network. I read the prior paper. There are lots of caveats to the Neural Network result - it traces back to the result in https://arxiv.org/pdf/2003.00307 - where authors show that - wide and deep neural nets satisfy the PL condition over a radius around a random initialization if the width scales as radius^depth.  Further, the theorem is easily proven only for linear function approximation in Kang et.al. 2025.

2) Second concern is that ghost index trick works because, say for the inner problem, gradient is assumed to be uniformly bounded. This is rather a very strong assumption. However, the inner problem is strongly concave and *Page 2 of this ICML paper https://proceedings.mlr.press/v80/nguyen18c.html  shows that unless the ball of iterates is bounded explicitly, uniform gradient norm bound contradicts strong convexity (or concavity) !*

Authors can have uniform bound G on gradient norm only if the iterates stay within a ball of certain radius from where it starts at least for the inner concave problem. The algorithm described is unprojected SGDA and the problem needs to project itself on every update to some ball. In the RL context that would mean projecting the iterates of the parameters of the Q function to a ball that would encapsulate the optima - rather a very strong assumption. Even the Neural net satisfying bounds of gradient, Hessian and Jacobian operator (assumption 5 in Kang et al 2025  paper) is possibly within some small ball around the initialization for a network of given width.

**Questions:**

1) Can you answer the above 2 weakness points ? Question about the need for projected steps if gradient bound is assumed is rather concerning and could be a serious weakness as written

2) Paper quotes the deadly triad relating to convergence of Q learning. There is a recent paper on resolving it for linear function approximation (https://arxiv.org/abs/2203.02628) using truncation and target network. Discussing these alternative works is very important.

I think the gradient bound issue is more serious. Therefore, I have given rating of 2. I would wait for authors to respond to that and I can raise my score.

---

> ### Author Response · Authors · 2025-11-24
> **We thank the reviewer MpYr for letting us improve essential details of the paper.**
>
> We thank the reviewer MpYr for letting us improve essential details of the paper.
>
> ### **Answer to comments in "Weaknesses"**
> **Comment 1.**
> > "... quoting of results from Kang et al 2025 that misleads reading this paper. Line 230 and 231 says that Kang et al. 2025 proved that PL condition is satisfied with respect to the parameters of the Q function (primal variables) when parameterized by a Neural Network."
>
> **Author response to Comment 1**:
>
> Thank you for pointing out this representation issue in the draft. Indeed, what Kang et al. 2025 prove is that, "under a condition which Liu et al. proved to hold locally for Neural network parametrization (Assumption 5 in Kang et al. 2025), PL condition *locally* holds for the region that matters (a ball around the initialization), under the condition that *width scales as $\\text{radius}^\\text{depth}$ "*. We appreciate the opportunity to correct this.
>
> We change the Line 230 and 231 to:
>
> *Kang et al., 2025 proved that both equation 3 and equation 4 satisfy the Polyak-Łojasiewicz condition within a large enough ball around the initialization point for a Neural Network parametrization of $Q$ with a sufficient network width (width scaling with $\\text{radius}^\\text{depth}$), and therefore the SGDA algorithm finds the $(\\boldsymbol{\\theta}_1, \\boldsymbol{\\theta}_2)$ that are global minima of Equation 4.*
>
>
>
> **Comment 2.**
> > "... gradient is assumed to be uniformly bounded (A5). This is rather a very strong assumption... uniform gradient norm bound contradicts strong convexity (or concavity)"**
>
> **Author response to Comment 2:** Thank you for letting us know that on an unbounded domain $\\mathbb{R}^d$, a function cannot satisfy both global strong concavity/convexity and a global gradient bound (Assumption 5). Fortunately, we found that we can avoid this assumption, as we only use a weaker one in the proof procedure. We appreciate the opportunity to correct this.
>
> We have updated the new draft, substituting the Assumption A5 to "effective domain" bound. That is, the assumed gradient's boundedness is only "along the optimization trajectory". Specifically, we say:
> * **(A5 [updated]) Bounded gradients on the effective domain.**
> There exists a compact convex set $\Omega \subset \mathcal{W} \times \mathcal{V}$ such that the sequence of iterates $\\{(w\_t, v\_t)\\}\_{t=0}^{T}$ generated by the algorithm remains within $\Omega$ almost surely.  We define $G$ as the uniform gradient bound on this set:
> \\[
> G := \\sup_{(w,v) \\in \\Omega, z \\in \\mathcal{Z}} \\max \\left\\{ \\|\\nabla_w f(w,v; z)\\|, \\|\\nabla_v f(w,v; z)\\| \\right\\} < \\infty.
> \\]
>
> And then we add these phrases right after:
>
> *Assumption A5 is justified by the coercivity of the PL-Strongly Concave landscape, which ensures iterates remain in a bounded sublevel set [(Yang et al., 2020)](https://arxiv.org/pdf/2002.09621) While strong concavity implies unbounded gradients on $\\mathbb{R}^d$, the geometry of problem equation 4 induces a drift that keeps iterates bounded (coercivity). Thus, we only require the gradient to be bounded within the effective domain $\\Omega$ visited by the algorithm, consistent with the global convergence guarantees for unprojected SGDA in this setting [(Yang et al., 2020)](https://arxiv.org/pdf/2002.09621).}*
> .
>
> ### **Answer to comments in "Questions"**
> **Question 2.**
> > Paper quotes the deadly triad relating to convergence of Q learning. There is a recent paper on resolving it for linear function approximation (https://arxiv.org/abs/2203.02628) using truncation and target network. Discussing these alternative works is very important.
>
> **Author response to Question 2:**
>
> Thank you for pointing this out. We put the paper in the new draft.
>
> Again, we appreciate your enormous effort in reviewing the paper and the opportunity to correct essential errors in the details that could have confused potential readers.

---

> ### Comment · Area_Chair_16qU · 2025-11-26
>
> The evaluation of this submission has a large variance so we need to gather more information from both the authors and the reviewers. Please take a look at the author's response and see how it affects your evaluation.
>
> Best,
> AC

---

### Author Response · Authors · 2025-11-13
**Sincere gratitude to reviewers and ACs**

Dear AC and reviewers,

We would like to express our sincere gratitude to the area chair and the reviewers. We have not encountered such thoughtful, thorough, and high-quality reviews in many years, and we are deeply appreciative of the care and expertise that went into them.

In particular, we are especially grateful to the reviewers who invested substantial time and effort in evaluating our work. Your detailed comments and suggestions have given us many concrete ideas for improving the paper, and they will significantly strengthen the paper. We are grateful for this chance given to us to learn this area more.

Thank you again for your exceptional feedback and support. Although our score set is closer to rejection than acceptance, we would not trade this review set for a set of acceptance-level scores with less effort.

We will upload our detailed responses to your reviews as soon as possible, in the hope that they further clarify our contributions and address your concerns.

Authors

---

### Author Response · Authors · 2025-11-26
**Sincere gratitude to reviewers and ACs**

Again, we would like to express our sincere gratitude to the area chair and the reviewers for their efforts. We enjoyed high-quality, constructive reviews, and we are again deeply appreciative of the care and expertise that went into them.

We would particularly like to thank reviewer **MpYr** and **kFyc** for identifying that assuming bounded gradients is a strong and unrealistic assumption. This assumption was indeed unnecessarily strong, and indeed we can simply relax it to a justifiable level by leveraging the coercivity of the PL-Strongly Concave landscape. This was a simple but critical theoretical improvement that we really appreciated. We also thank the reviewer for giving us a new paper on the deadly triad that we can cite.

We want to thank other reviewers for their valuable suggestions for improving representation.
- We would particularly like to thank reviewer **yTNX** for clarifying what our contribution is. (Comparison with mean-squared projected Bellman error minimizing methods, divergence of approximate dynamic programming methods for neural networks (Proposition 1, [Jiang and Xie 2025](https://nanjiang.cs.illinois.edu/files/STS_Special_Issue_Offline_RL.pdf), comparison with Kang et al, motivation behind Lyapunov potential?...)
- We would like particularly to thank reviewer **uiVV** for pointing out that it would be nice if we can remove $\theta_1, \theta_2$ and just use $v, w$ throughout. This enabled us to significantly improve the paper's representation. We also thank you for allowing us to include the paragraph stating that Assumption A8 is not a strong assumption common in the literature, as well as the paragraph comparing it with previous related work.
- We would like to particularly thank **kFyc** for giving us lots of detailed insights on how to improve representation for general audiences. In particular, clearly separating stability/generalization error from optimization error improved the paper's presentation. We also thank you for allowing us to include the paragraph stating that Assumption A8 is not a strong assumption common in the literature.

---

### Author Response · Authors · 2025-12-03
**Summary of review responses (3)**

### 4. Remaining items and scope

Reviewer kFyc’s follow-up distinguishes between concerns that are now resolved and suggestions that remain as open opportunities. The resolved items include A5, A8, A9, positioning relative to convex–concave minimax work, minibatch scaling, and the prominence of the excess-risk decomposition.

The remaining points are explicitly acknowledged as future directions:

- A small-scale empirical sanity check that illustrates the $\mathcal{O}(1/n)$ slope in a toy BRM setting.
- Extensions of the Lyapunov + ghost-index framework to other PL–minimax or actor–critic problems.
- High-probability bounds and condition-number sensitivity.
- A brief remark on the precise way in which the hard-max Bellman operator breaks the PL–strongly-concave structure.
- More concrete, practically verifiable sufficient conditions for neural BRM (e.g., norm or spectral controls).

Reviewer kFyc notes that these are “unrealized opportunities” rather than correctness issues; they do not alter the validity of our current expectation-based theory in the entropy-regularized BRM setting.

---

In summary, the revised manuscript and our responses address all substantive technical and clarity-related concerns raised in the reviews: assumptions have been weakened and justified in line with modern PL–minimax analyses; the relationship to Kang et al. (2025), Wang et al. (2022), and classical approximate dynamic programming methods is now explicit; the Lyapunov potential and stability argument are motivated in detail; and the exposition has been significantly streamlined. At least one reviewer (kFyc) confirms in their final comment that their core technical concerns are resolved and maintains a positive overall assessment (score 6). We hope this clarifies that the remaining discussion points concern future extensions, rather than gaps in the main theoretical claims, and we respectfully ask you to consider the paper in this light.

---

### Author Response · Authors · 2025-12-03
**Summary of review responses (2)**

### 2. Positioning relative to prior work and clarity of contribution

Reviewer yTNX asked for a clearer comparison to approximate dynamic programming methods, MSPBE minimization, and other BRM alternatives, and for a sharper delineation of what is inherited from Kang et al. (2025) and Wang et al. (2022) versus what is new in our work.

**BRM vs. MSPBE and approximate dynamic programming.**
We added a detailed discussion explaining that MSPBE-based methods minimize the projected Bellman error, i.e., the Bellman residual after projection onto the approximation space, while BRM directly controls the full Bellman residual. Using the Pythagorean decomposition for orthogonal projection, we show that MSPBE measures only the component of the residual inside the function class and ignores the orthogonal component, which matters once the approximation space is misspecified. We also explain, following Jiang and Xie (2025), that approximate dynamic programming methods can diverge under neural-network parametrization, and that marginal importance sampling methods do not guarantee Bellman consistency. This motivates our focus on BRM as the only currently known exact and stable route to Bellman consistency under neural networks.

**What comes from Kang et al. vs. what is new here.**
We now explicitly state that Kang et al. (2025) provide the PL–strongly-concave geometry for BRM and a global optimization guarantee for SGDA, but do not analyze algorithmic stability or generalization. The latest version of their paper even cites our result for BRM sample complexity. Our work takes their PL constants as input and builds the stability and generalization theory from scratch.

**What comes from Wang et al. vs. what is new here.**
We reuse the conceptual framework of on-average argument stability and the shared-index coupling idea from Wang et al. (2022). The key differences are now clearly explained: Wang et al. work with convex–concave or strongly-convex–strongly-concave objectives and analyze the original minimax risk, whereas our BRM objective is nonconvex in the primal variables, PL only in the value function, and the algorithm operates on a bilevel saddle objective whose gradients are biased with respect to the maximized value function $\Phi_D$. This mismatch necessitates our new Lyapunov potential.

**Lyapunov potential: motivation and role.**
In response to questions from yTNX and kFyc, we added an intuitive explanation of the Lyapunov potential $\Psi_{\alpha,D}$, emphasizing that it simultaneously measures (i) how far the primal iterate is from optimality in terms of the value function and (ii) how far the dual variable is from its maximizer, so that the SGDA gradient is effectively aligned with $\nabla \Phi_D$. This type of potential is absent in both Kang et al. and Wang et al. and is specifically tailored to the PL–strongly-concave BRM setting and the coupled-run stability analysis.

Reviewer yTNX’s main concerns about contribution clarity, comparison to alternatives, and Lyapunov motivation are therefore addressed in the revised text.

---

### 3. Exposition and readability

Several reviewers requested improvements to the exposition, especially in Section 3.

**Abstraction to a generic PL–strongly-concave saddle problem.**
Reviewer yTNX found the jump to a general function $F$ with nine assumptions opaque. We added a bridging paragraph that explains why the stability analysis temporarily abstracts away from the specific BRM objective to a generic PL–strongly-concave empirical saddle problem, and that all assumptions (A1)–(A9) are satisfied by the BRM objective under the conditions of Section 2. We also state explicitly that Theorem 6 specializes the abstract stability and generalization bounds back to BRM.

**Introduction and notation clean-up.**
In response to multiple comments, we improved the introductory discussion of Bellman consistency and its role in offline RL, fixed minor grammatical issues, removed redundant “Equation” prefixes, and unified the notation for primal/dual variables (using a single pair consistently).

**Excess-risk decomposition and optimization vs. generalization error.**
Reviewer kFyc asked that the decomposition into optimization and statistical terms be more prominently presented, and that the algorithmic error be clearly separated from the stability-based generalization term. The revised version highlights this decomposition and, in our comparison table, aligns it with the rates obtained in Wang et al. (2022).

**Assumption A8 and comparison to prior stability work.**
As noted earlier, we added an explicit paragraph that connects A8 to standard Lipschitz/smoothness assumptions in stability papers, and clarified the dependence of constants across neighboring datasets.

Reviewer uiVV’s questions about assumptions, thresholds where optimization error falls below the statistical term, and notation unification are all addressed in the revised draft and appended clarifications.

---

### Author Response · Authors · 2025-12-03
**Summary of review responses**

Dear AC,

Thank you again for coordinating the discussion around our submission "Stability and Generalization for Bellman Residuals." We write this final note to summarize how the revised manuscript and our responses address the main concerns raised by the reviewers, and to clarify what remains as clearly flagged future work rather than unresolved issues. **Only Reviewer kFyc was able to provide a follow-up comment before the response-freeze caused by the reviewer-information-leak incident; the other reviewers did not have an opportunity to submit their post-rebuttal feedback.**

Our central contribution remains an $\mathcal{O}(1/n)$ stability and generalization guarantee for Bellman Residual Minimization (BRM) in offline RL/IRL, for Markov data and standard SGDA without independence assumptions. The analysis couples two SGDA trajectories via a single Lyapunov potential, and transfers on-average argument stability to an excess MSBE risk bound that cleanly separates optimization and statistical errors.

Below, we organize the resolution of reviewer concerns into four themes.

---

### 1. Core technical assumptions and correctness

**Assumption A5 (bounded gradients).**
Reviewers MpYr and kFyc highlighted that the original global bounded-gradient assumption conflicts with strong convexity/concavity on an unbounded domain. In the revised draft, we no longer assume global boundedness. Instead, A5 is now stated as “bounded gradients on the effective domain”: we assume only that SGDA iterates remain in a compact convex set $\Omega$ and that gradients are uniformly bounded on $\Omega$. This assumption is mathematically justified by the coercivity of the PL–strongly-concave landscape, which keeps iterates in a bounded sublevel set and aligns with known global convergence guarantees for unprojected SGDA (which was proved in [Yang et al. 2020](https://arxiv.org/abs/2002.09621)). *Reviewer kFyc explicitly notes that this revision resolves their concern about A5.*

**Assumption A8 (uniform constants across neighboring datasets).**
Reviewer uiVV asked how strong A8 is and requested an example. We added a paragraph explaining that A8 is the standard PL/QG analogue of the uniform Lipschitz and smoothness conditions assumed in stability analyses such as Hardt et al. (2016) and Wang et al. (2022). Under these usual uniform bounds on per-example losses, the corresponding constants automatically match for any dataset and its replace-one neighbor, and A8 is the direct counterpart in the PL setting. *Reviewer kFyc’s post-rebuttal comment acknowledges that the clarification of A8 is satisfactory.*

**Assumption A9 (uniqueness of the saddle).**
Reviewer kFyc asked that we explain the role of the unique saddle assumption and how it is justified. The revised draft now explicitly discusses the justification of A9 referring to [Yang et al. 2020](https://arxiv.org/abs/2002.09621)) and clarifies how it is used to control the Lyapunov potential and the primal–dual gap, and how it fits into the PL–strongly-concave setting. In their follow-up, kFyc lists “Clarification regarding uniqueness of the saddle (Assumption A9)” among the issues that are fully addressed.

**PL condition and neural networks.**
Reviewer MpYr pointed out that our original wording overstated what Kang et al. (2025) proved about neural networks. We corrected the text to make it precise that PL holds locally in a ball around the initialization, under the width–depth scaling conditions established via Liu et al. The revised wording states that both the population and empirical BRM objectives satisfy PL within a sufficiently large ball around the initialization for neural networks with appropriate width, and that SGDA then converges to global minima of the empirical objective.

**Markov vs. i.i.d. data and minibatch scaling.**
Reviewer kFyc requested clearer positioning relative to minimax stability for convex–concave problems and an explicit statement that our analysis targets Markov-sampled data. In the revised version, we added a comparison table contrasting geometry, setting, rates, and independence assumptions between our work and Wang et al. (2022), and we explicitly state that our goal is to handle Markovian dependence rather than i.i.d. samples. The same comment also notes that minibatch scaling, variance-reduction factors, and the previously ambiguous “one pass over $n$ samples” phrasing have been corrected.

Taken together, these changes resolve the main technical concerns around the strength, interpretation, and correctness of our assumptions. Reviewer kFyc summarizes this by stating that A5, A8, A9, and the connection to prior minimax stability work and minibatch scaling are now satisfactorily addressed.

---

### Meta-Review · Area_Chair_Tj23 · 2026-01-05

**Summary:**

This paper addresses the minimization of Bellman error in TD updates. While the reviewers acknowledged the importance of this direction, they raised significant technical concerns regarding the strength of the assumptions and the clarity of the proofs.

Although the authors attempted to address these issues during the rebuttal, the resulting revisions involve fundamental and extensive modifications. Given the limited timeframe of the rebuttal phase, it is not feasible to properly verify the correctness of these major changes. Consequently, I believe this work would benefit from a full review cycle where the revised theoretical contributions can be evaluated in depth.

**Reviewer Concerns:**

While the problem setting is important, critical concerns raised by reviewers MpYr and yTNX regarding the theoretical foundations remain unresolved.

Specifically, Reviewer MpYr questioned the strength of the assumptions. Although the authors proposed replacing Assumption A5 with a weaker variant in the rebuttal, they failed to discuss how this change propagates through the rest of the paper or verify that the subsequent proofs remain valid under this new condition.

Additionally, Reviewer yTNX raised concerns regarding proof novelty and the realizability of the assumptions on standard value functions. The authors' rebuttal clarified the differences between this work and prior art, but did not convincingly argue the significance of these differences. Furthermore, the rebuttal did not fully/detailed demonstrate that the assumptions actually hold for standard value functions (especially when there are errors in discussion on Kang et al 2025).

**Reviewer Scores:**

The two negative reviewers did not reply, yet I believe their concerns are not fully addressed.

---

### Decision · Program_Chairs · 2026-01-26

Reject